# MATRIX: MASK TRACK ALIGNMENT FOR INTERACTION-AWARE VIDEO GENERATION

**Siyoon Jin**     **Seongchan Kim**     **Dahyun Chung**     **Jaeho Lee**     **Hyunwook Choi**
**Jisu Nam**     **Jiyoung Kim**     **Seungryong Kim**
KAIST AI
https://cvlab-kaist.github.io/MATRIX

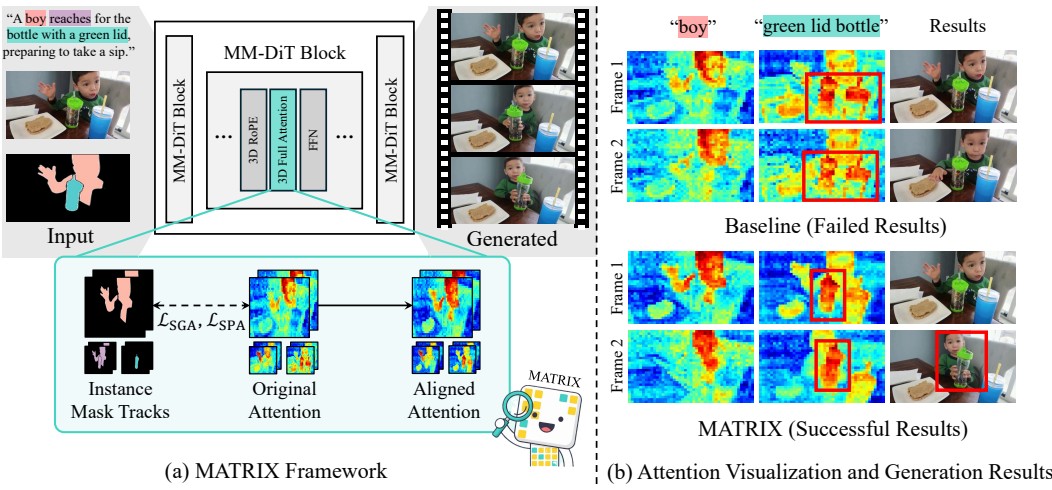

Figure 1: **Teaser:** We reveal how video diffusion transformers (DiTs) represent multi-instance or subject-object interactions during video generation by focusing on their internal attention mechanisms. Building on these, our MATRIX framework further enhances the interaction-awareness of video DiTs via the proposed Semantic Grounding Alignment (SGA, $\mathcal{L}_{\text{SGA}}$) and Semantic Propagation Alignment (SPA, $\mathcal{L}_{\text{SPA}}$) losses.

## ABSTRACT

Video DiTs have advanced video generation, yet they still struggle to model multi-instance or subject-object interactions. This raises a key question: *How do these models internally represent interactions*? To answer this, we curate MATRIX-11K, a video dataset with interaction-aware captions and multi-instance mask tracks. Using this dataset, we conduct a systematic analysis that formalizes two perspectives of video DiTs: *semantic grounding*, via video-to-text attention, which evaluates whether noun and verb tokens capture instances and their relations; and *semantic propagation*, via video-to-video attention, which assesses whether instance bindings persist across frames. We find both effects concentrate in a small subset of interaction-dominant layers. Motivated by this, we introduce **MATRIX**, a simple and effective regularization that aligns attention in specific layers of video DiTs with multi-instance mask tracks from the MATRIX-11K dataset, enhancing both grounding and propagation. We further propose **Inter-GenEval**, an evaluation protocol for interaction-aware video generation. In experiments, MATRIX improves both interaction fidelity and semantic alignment while reducing drift and hallucination. Extensive ablations validate our design choices. Codes and weights will be released.

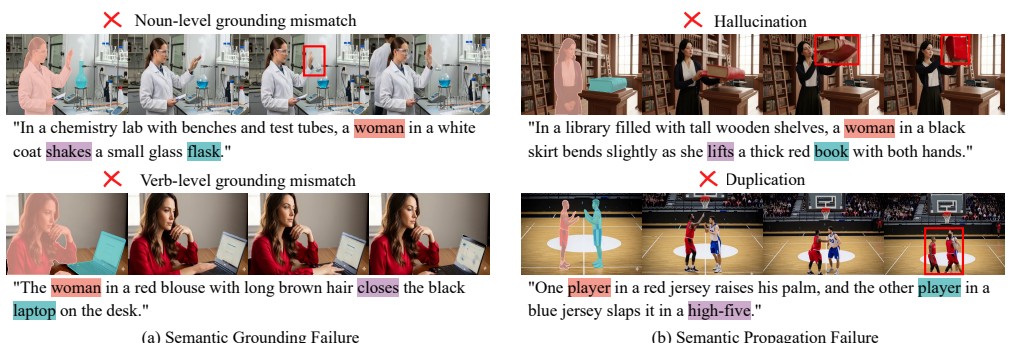

Figure 2: **Failure cases of existing video DiTs:** (a) semantic grounding failures, where subjects, objects, or their verb relations are mismatched, and (b) semantic propagation failures, where bindings break over time, leading to hallucinations or duplications. Overlays indicate the intended instances.

# 1    INTRODUCTION

Recent video diffusion transformers (DiT) have advanced text-to-video generation and manipulation of a single object or human, enabling applications in simulation (Soni et al., 2025), AR/VR (Zhou et al., 2025), robotics (Kim & Joo, 2025) and embodied reasoning (Feng et al., 2025b). Despite these advances, DiT-based models (Yang et al., 2024; Zheng et al., 2024) still struggle to generate multi-instance or subject-object interactions from text prompts (*e.g., who does what to whom*).

As illustrated in Fig. 1 and 2, two main failures emerge: (1) *semantic grounding failure*, where they fail to localize subject or object specified by prompt nouns or to bind verb-specified subject-object interaction, resulting in text-video mismatch; and (2) *semantic propagation failure*, where this noun/verb grounding does not persist over time, causing drift, duplication, or hallucination. These observations raise key questions, *How do video DiTs semantically bind text and video, and how is this binding propagated to support interactions?*, which motivates us to analyze and strengthen this to improve interaction-aware video generation.

Fig. 3 and Fig. 4 motivate our analysis. In 3D full attention of video DiTs Yang et al. (2024), video-to-text attention aligns noun tokens with subject and object regions and verb tokens with their interaction region, which is the union of subject and object. Video-to-video attention propagates this binding across frames, by linking interaction regions in one frame to the corresponding regions in other frames. Especially, in successful generations, these alignments concentrate in a few layers and persists across frames. We regard these alignments as the binding to analyze, assessing where it emerges and whether it persists across frames. To quantify this binding, the reference must provide spatial precision to verify grounding and temporal continuity to test persistence, and instance separability to disambiguate same-class instances. We therefore adopt *multi-instance mask tracks* as the reference, since for each instance, a per-frame mask is linked by a persistent ID across the video, and the union of the subject and object masks defines the interaction region.

Since no existing dataset pairs such tracks with interaction-aware captions, we curate **MATRIX-11K**, 11K videos with interaction-rich captions and instance masks tracks. With MATRIX-11K, we conduct the first systematic study of how subject-object interactions are internally represented in video DiTs (Yang et al., 2024; Peebles & Xie, 2023; Esser et al., 2024). We analyze 3D full attention where text and video tokens interact, and study two core perspectives: *semantic grounding*, via video-to-text attention, measuring whether noun tokens localize to subject or object regions and verb tokens attend to their union; and *semantic propagation*, via video-to-video attention, measuring whether these noun/verb groundings are preserved so identities and their interaction persists across frames. We observe both effects emerge strongly in a small subset of layers, which we term **interaction-dominant layers**, and the alignment in these layers is consistently stronger in successful generations and weaker in failures, yielding a clear success-failure contrast.

Based on these insights, we propose **MATRIX (Mask Track Alignment for Interaction-Aware Video Generation)**, a simple yet effective regularization that aligns attention in interaction-dominant layers with multi-instance mask tracks. We finetune the image-to-video model (Yang et al., 2024) with LoRA (Hu et al., 2021), condition on multi-instance mask and supervise only

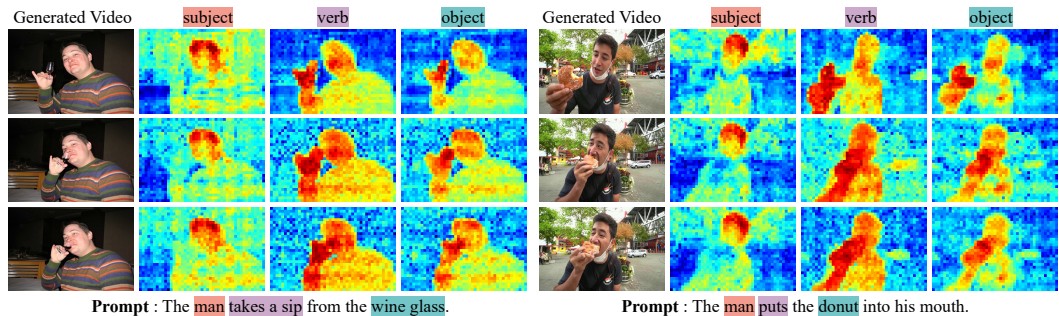

Figure 3: **Attention maps per token type.** Noun tokens (subject, object) align with their respective regions (*e.g., layer 11*); verb tokens aligns with the union of subject–object regions (*e.g., layer 7*).

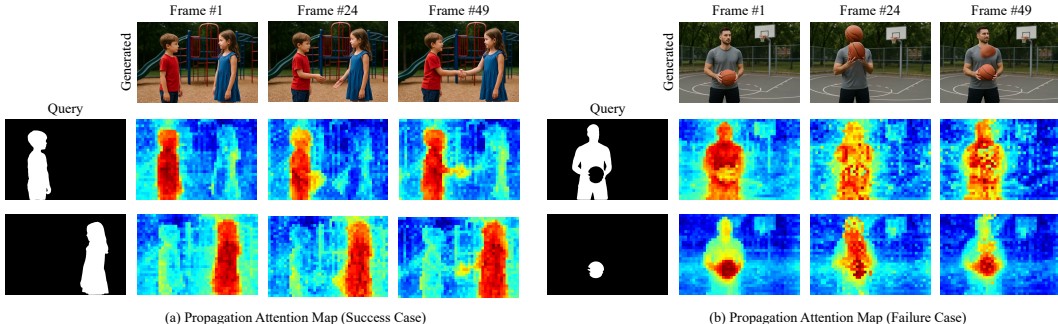

Figure 4: **Visualization of attention maps.** Propagation maps (*e.g., layer 12*) directly affect whether interaction propagation succeeds or fails.

those layers via two terms: Semantic Grounding Alignment (SGA) loss, which aligns noun tokens with subject/object regions and verb tokens with unions in video-to-text attention, and Semantic Propagation Alignment (SPA) loss, which enforces video-to-video attention to preserve consistent instance tracks across frames. To align attention space and pixel space, we introduce a lightweight causal decoder that maps attention to frame-level mask tracks. Our approach applies to any Video DiTs that employ 3D full attention.

In addition, existing metrics (Huang et al., 2023; Zheng et al., 2025a; Gu et al., 2025a) capture only global alignment and cannot localize subjects, verbs, or objects, making interaction-aware evaluation unreliable. We introduce **InterGenEval**, an interaction-aware evaluation protocol. Specifically, key interaction semantic alignment (KISA) checks the pre, during, and post conditions of key interaction. Semantic grounding integrity (SGI) measures whether the subject, object and verb are correctly grounded. Semantic propagation integrity (SPI) assess the temporal persistence of bindings and is applied alongside KISA and SGI. Interaction fidelity (IF) is reported as the mean of KISA and SGI.

In summary, our contributions are:

- We construct MATRIX-11K, 11K videos with multi-instance mask tracks and interaction-aware captions for both analysis and training.
- We introduce the first systematic analysis of semantic grounding and semantic propagation in video DiTs, revealing how subject-object interactions emerge.
- Motivated by our analysis, we propose MATRIX, an effective regularization composed of SGA and SPA, applied to interaction-dominant layers, and conditioned on multi-instance mask tracks via lightweight LoRA, improving grounding accuracy and consistency.
- We design InterGenEval, a protocol for evaluating interaction-awareness of the video.

## 2 RELATED WORK

**Interaction Representations in Video DiTs.** Prior works have examined internal representations in UNet image diffusion (Hedlin et al., 2023; Jin et al., 2025; Nam et al., 2024; Tang et al., 2023), image

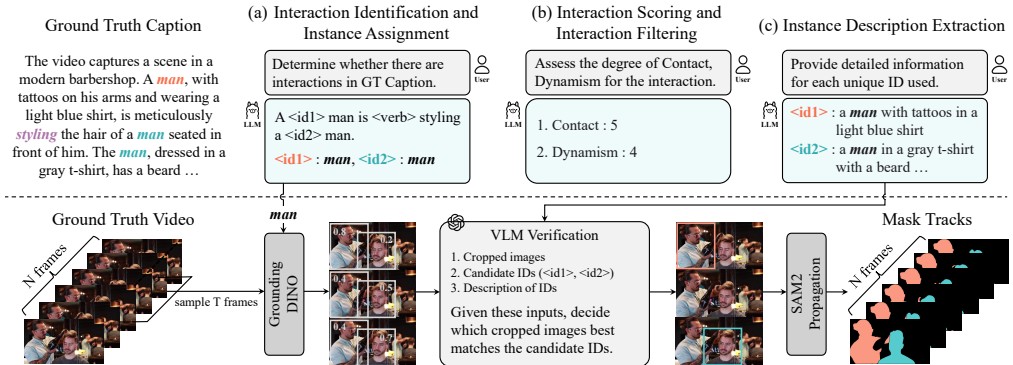

Figure 5: **Our dataset curation pipeline.**

DiTs (Yu et al., 2025; Lee et al., 2025), and video DiTs (Nam et al., 2025; Zhang et al., 2025), but none formalize interaction representations. Since pixel-level reconstruction give little supervision for binding *"who does what to whom"* or maintaining it over time, an analysis of interactions in Video DiTs remains absent. We therefore define interactions as semantic grounding (token level binding) and semantic propagation (temporal binding), and analyze through attention.

**Human-Object Interaction (HOI) Synthesis.** Research in HOI synthesis has explored generating human motions conditioned on interaction prompts. Early works (Chao et al., 2018; Gkioxari et al., 2018) focused on recognizing and localizing HOIs in 2D, while more recent studies (Pi et al., 2023; Soni et al., 2025) synthesize 3D motions of a single human or multiple humans under verb conditioning. These methods demonstrate that interactions can be generated when instances are explicitly parameterized, but they remain restricted to motion-level synthesis. Importantly, they have not been integrated into video diffusion, where interaction modeling must directly govern pixel generation. **Relation Customization.** Recent methods (Wei et al., 2025; Tan et al., 2025) customize specific relations (e.g., *pick up*) via relation-specific adapters or motion priors. While effective in narrow cases, they rely on a closed verb set, require per-relation tuning, decouple control from text grounding, and struggle with multiple instance pairs, limiting generalization to open-vocabulary verb set. **Controllable Video Diffusion Models.** Controllable video generation (Esser et al., 2023; Zhang et al., 2023; Cai et al., 2025; Li et al., 2025; Gu et al., 2025b; Geng et al., 2025; Feng et al., 2025a) introduces guidance signals such as depth, bounding boxes, optical flow, or trajectories to constrain scene geometry and motion. While such controls improve temporal consistency and enable user-defined dynamics, they remain agnostic to interaction semantics. Even multi-instance controls using bounding boxes or mask sequences operate independently of text, leaving subject-action-object relations under-specified. As a result, controllable methods support single-instance manipulation but fall short on multi-instance interactions, which require explicit alignment with textual descriptions.

# 3 MATRIX-11K DATASET

To systematically analyze and enhance semantic binding in 3D full attention of video DiTs, we introduce **MATRIX-11K**, a dataset of videos $V$ paired with interaction-aware captions $P$ and instance mask tracks $M$ for each instance ID $k$. Prior datasets (Goyal et al., 2017; Ravi et al., 2024; Li et al., 2021; Zhang et al., 2020) often suffer from low video fidelity, or semantically weak or misaligned captions and mask. MATRIX-11K addresses this by aligning instance mask tracks with interaction-aware captions, via an interaction-aware curation pipeline. We will release the curation pipeline and this dataset publicly. Sec. 3.1 describes LLM-based caption processing for interaction and ID extraction, while Sec. 3.2 details mask track construction with GroundingDINO, VLM verification and SAM2 propagation.

## 3.1 INTERACTION-AWARE CAPTIONING

We employ an off-the-shelf LLM (et al., 2024) to process caption $P$ in three steps. First, the LLM identifies whether an interaction verb is present (*e.g.,* hold, throw) and assigns an instance ID $k$ to every participating noun while recording its base-noun class (*e.g.,* man, cup). This yields interaction

triplets $\langle k_{\mathrm{sub}}, \mathrm{verb}, k_{\mathrm{obj}}\rangle$, where $k_{\mathrm{sub}}$ and $k_{\mathrm{obj}}$ denote the IDs bound to the subject and object nouns, and will later be tied to an instance mask track $M_k$. Second, to focus on physically grounded and temporally meaningful interactions, the LLM scores each interaction for *Dynamism* (degree of motion or temporal change) and *Contactness* (physical contact or spatial proximity). Only interactions exceeding predefined thresholds are retained, and any ID $k$ not linked to a retained interaction is also removed. Third, for every retained $k$, the LLM extracts an appearance description (*e.g., a man in a gray shirt*) to disambiguate same-class instances, which we later use for VLM verification.

## 3.2 MULTI-INSTANCE & INTERACTION MASK TRACKS

For each video and its instance set, we uniformly sample frames and use GroundingDINO (Liu et al., 2024) to generate multiple bounding box candidates per instance ID $k$, each with a confidence score. We begin with the highest-confidence candidate; if it fails VLM verification, we move to the next highest and continue until one verifies or all fail. A VLM (OpenAI & et al., 2024) inspects each candidate as visual prompt together with the class label and the appearance description of $k$ from Sec. 3.1 and decides whether it matches the target instance. The first verified candidate becomes the anchor frame and box. From the anchor, we initialize SAM2 (Ravi et al., 2024) and propagate masks through the clip to obtain a per-ID instance track $M_k$. If all candidates fail we remove $k$ and drop any interaction that is linked to it. Videos with no remaining valid interactions are discarded.

Finally, human annotators manually inspect and filter residual errors, such as mask drift, missing frames, or misaligned clips. Fig. 30, Fig. 31, Fig. 32 and Fig. 33 provide examples of the final dataset we curated. More details including data statistics are provided in Appendix A.

## 4 INTERACTION-AWARENESS ANALYSIS IN VIDEO DiTs

We present, to our knowledge, the first systematic analysis of how Video DiTs (Yang et al., 2024; Wan et al., 2025; Kong et al., 2024) internally represent text-based interactions during generation. We ask whether DiTs encode (i) *semantic grounding*, where textual tokens (nouns, verbs) localize to the correct visual regions, and (ii) *semantic propagation*, where these bindings remain spatially coherent over time so that instance identities and relations persist. These analyses determine whether models capture interactions *end-to-end*, both grounding roles ("who does what to whom") and propagating them throughout the sequence. This analysis motivates our regularization.

### 4.1 PRELIMINARIES- VIDEO DIFFUSION TRANSFORMERS

A MM-DiT (Esser et al., 2024), the basic block of video DiT, stacks layers of 3D full attention that jointly process spatiotemporal and textual information. This design allows the model to integrate text and video tokens during generation. In the $l$-th layer, attention is:

$$\mathrm{Attn}(\mathbf{Q}_l, \mathbf{K}_l, \mathbf{V}_l) = \mathbf{A}_l \mathbf{V}_l, \quad \text{where } \mathbf{A}_l = \mathrm{Softmax}\left(\frac{\mathbf{Q}_l \mathbf{K}_l^{\mathrm{T}}}{\sqrt{d}}\right),$$

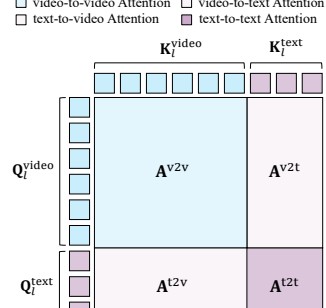

Figure 6: **Illustration of full 3D attention in video DiTs.**

Here, $\mathbf{Q}_l, \mathbf{K}_l, \mathbf{V}_l$ are query, key, value matrices of the $l$-th layer, and $d$ is dimension of key. 3D full attention in DiTs operates on a unified sequence concatenating video latents and text embeddings:

$$\mathbf{Q}_l = \mathrm{Concat}(\mathbf{Q}_l^{\mathrm{video}}, \mathbf{Q}_l^{\mathrm{text}}), \quad \mathbf{K}_l = \mathrm{Concat}(\mathbf{K}_l^{\mathrm{video}}, \mathbf{K}_l^{\mathrm{text}})$$

where $\mathrm{Concat}(\cdot)$ indicates the concatenation operation along the token dimension. To summarize, the attention matrix of a DiT can be divided into four distinct regions: video-to-video $\mathbf{A}^{\mathrm{v2v}}$, video-to-text $\mathbf{A}^{\mathrm{v2t}}$, text-to-video $\mathbf{A}^{\mathrm{t2v}}$ and text-to-text $\mathbf{A}^{\mathrm{t2t}}$, as shown in Figure 6. This unified formulation supports analysis of how Video DiTs bind visual and textual modalities into a coherent generative process and propagate across frames. In this work, we focus on $\mathbf{A}^{\mathrm{v2t}}$ and $\mathbf{A}^{\mathrm{v2v}}$.

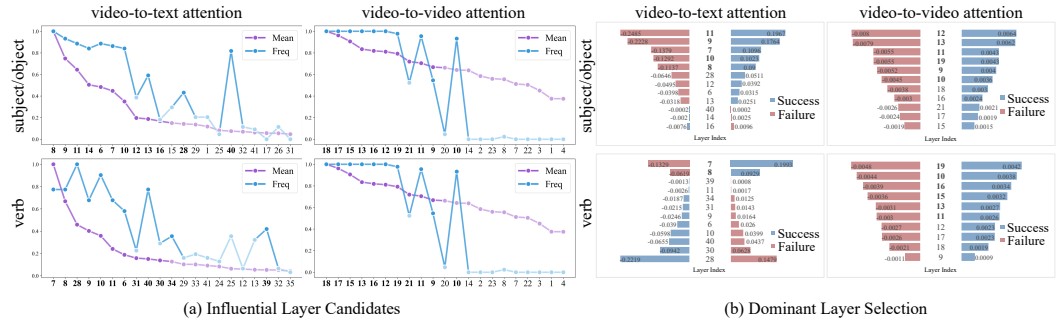

Figure 7: **Layer analysis.** (a) **Influential layers** : layers with high AAS that rank in the Top-10 for many videos. (b) **Dominant layers** : the influential layers whose mean AAS clearly separates successes from failures (Best viewed in zoom).

## 4.2 SEMANTIC GROUNDING

We ask whether video DiTs ground textual tokens to visual regions. We read $\mathbf{A}^{\text{v2t}}$ as a per-token heatmap: for text token $t$ (noun or verb), let $\mathbf{A}^{\text{v2t}}(t) \in \mathbb{R}^{F \times H \times W}$ denotes attention over $F$ frames and $H \times W$ latent grid. We consider: (i) *nouns*, which align with subject/object spatial regions, and (ii) *verbs*, which capture *interaction* via joint attention to subject and object.

**Noun Grounding.** Nouns cover the roles of the instance, *subject* and *object*. For each role $e \in \{\text{sub}, \text{obj}\}$, we form a token set $\mathcal{T}_e$, containing the role's head noun and its modifiers. Since modifiers tend to attend to the instance region as the head noun, we aggregate heatmaps by mean:

$$\mathbf{A}_e^{\text{v2t}} = \frac{1}{|\mathcal{T}_e|} \sum_{t \in \mathcal{T}_e} \mathbf{A}^{\text{v2t}}(t), \quad e \in \{\text{sub}, \text{obj}\}.$$

Concretely, the sequence $\mathbf{A}_e^{\text{v2t}} \in \mathbb{R}^{F \times H \times W}$ indicates where the subject/object is grounded.

**Verb Grounding.** Verbs express the interaction between the grounded subject and object. We obtain the verb map by averaging over the verb token set:

$$\mathbf{A}_{\text{verb}}^{\text{v2t}} = \frac{1}{|\mathcal{T}_{\text{verb}}|} \sum_{t \in \mathcal{T}_{\text{verb}}} \mathbf{A}^{\text{v2t}}(t),$$

where $\mathcal{T}_{\text{verb}}$ contains the head verb and auxiliaries/particles (*e.g.*, "is", "up" in "is lifting up"). For evaluation, $\mathbf{A}_{\text{sub}}^{\text{v2t}}$ and $\mathbf{A}_{\text{obj}}^{\text{v2t}}$ are compared to their respective instance mask tracks and $\mathbf{A}_{\text{verb}}^{\text{v2t}}$ is compared to their interaction region, which is the per-frame union of subject and object mask tracks.

## 4.3 SEMANTIC PROPAGATION

Semantic propagation asks whether properly grounded bindings *remain spatially coherent* over time. Specifically, the attention originating form a subject, or object region in the first frame should concentrate on the same instance over time, and the interaction region should remain clustered without drift or duplication. To this end, we study $\mathbf{A}^{\text{v2v}}$, which maps each video token to all others and reuse mask tracks $M_k$ (Sec. 3). For subject/object IDs $k_{\text{sub}}, k_{\text{obj}}$, we take first-frame masks $M_{\text{sub}}^0, M_{\text{obj}}^0$, downsample to latent grid $H \times W$ and denote the resulting masks as $m_{\text{sub}}^0, m_{\text{obj}}^0 \in \{0, 1\}^{H \times W}$ (we drop the frame superscript hereafter). The query sets are the latent locations where masks are one:

$$Q_e = \{(h, w) \mid m_e^0(h, w) = 1\} \quad e \in \{\text{sub}, \text{obj}\}, \quad Q_{\text{verb}} = Q_{\text{sub}} \cup Q_{\text{obj}}.$$

For any $q \in Q_e$ ($e \in \{\text{sub}, \text{obj}, \text{verb}\}$), let $\mathbf{A}^{\text{v2v}}(q) \in \mathbb{R}^{F \times H \times W}$ be the attention from $q$ to all spatiotemporal tokens. The propagation map is :

$$\mathbf{A}_e^{\text{v2v}} = \frac{1}{|Q_e|} \sum_{q \in Q_e} \mathbf{A}^{\text{v2v}}(q) \in \mathbb{R}^{F \times H \times W}, \quad e \in \{\text{sub}, \text{obj}, \text{verb}\}$$

producing per-frame heatmaps that trace identity-consistent attention for subjects, objects and a stable interaction region for verb. This produces the same canonical form as the grounding maps in Sec. 4.2, but shifts the focus from token alignment to temporal consistency.

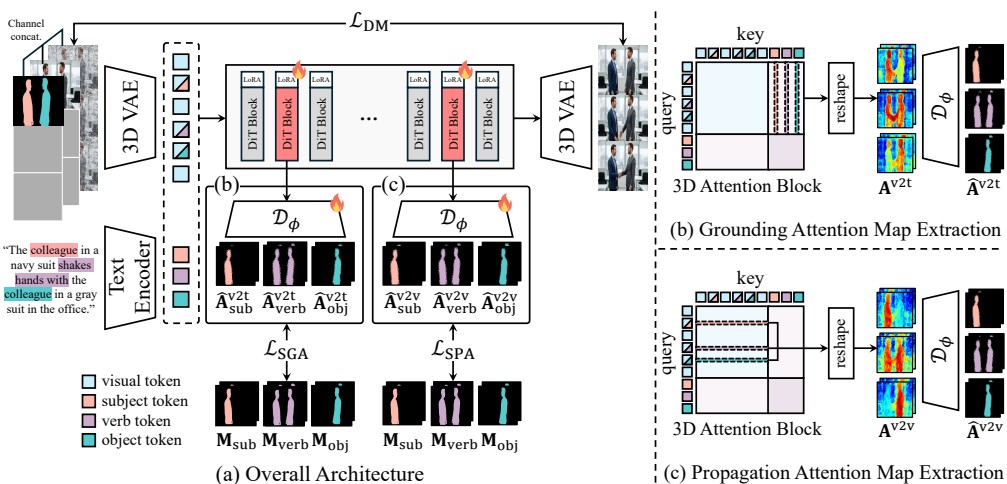

Figure 8: **Main architecture of MATRIX.**

## 4.4 EVALUATION METRIC: ATTENTION ALIGNMENT SCORE (AAS)

Each $\mathbf{A}_e^{\text{v2t}}$ or $\mathbf{A}_e^{\text{v2v}}$ is per-frame heatmaps, where larger values indicate more attention mass at that location. Using the mask tracks $M_{\text{sub}}, M_{\text{obj}}$ (Sec. 3), we downsample to latent grid to obtain $m_{\text{sub}}, m_{\text{obj}} \in \{0,1\}^{F \times H \times W}$ and define the verb mask tracks $m_{\text{verb}}$ by element-wise OR as $m_{\text{sub}} \vee m_{\text{obj}}$. Given $\mathbf{A}_e^{\text{v2t}}, \mathbf{A}_e^{\text{v2v}}$ with $e \in \{\text{sub}, \text{obj}, \text{verb}\}$, we score alignment by the sum of attention maps on mask-1 locations, called attention alignment score (AAS):

$$\text{AAS}_e^{\text{v2t}} = \sum_{f,h,w} (\mathbf{A}_e^{\text{v2t}} \odot m_e)(f,h,w), \quad \text{AAS}_e^{\text{v2v}} = \sum_{f,h,w} (\mathbf{A}_e^{\text{v2v}} \odot m_e)(f,h,w),$$

where $\odot$ indicates the element-wise multiplication.

## 4.5 ANALYSIS

We analyze CogVideoX-5B-I2V (Yang et al., 2024) for semantic grounding and propagation of both nouns and verbs. For all analyses, we compute the Attention Alignment Scores (AAS) defined in Sec. 4 from 3D full attention across 42 layers and 50 denoising timesteps. We consider four variants: noun grounding (v2t), verb grounding (v2t), noun propagation (v2v) and verb propagation (v2v). Additional analyses on other video DiTs (Wan et al., 2025; Kong et al., 2024) are in Appendix C.

**Layer Influence.** For each video, we rank layers by their step-averaged AAS and mark the top-10. Aggregating across videos, each layer receives two statistics. *Frequency* counts in how many videos the layer appears in the top-10. *Magnitude* is the mean AAS of that layer. As shown in Fig. 7 (a), we combine the two by a simple rank-sum and select the top-10 layers as **influential** for each variant. We find that the influence concentrates in a few layers that repeatedly achieve high alignment across videos, indicating that alignment is governed by specific layers rather than by outliers.

**Layer Dominance.** Among influential layers, we identify the dominant layer that directly govern outcomes. We split the generated video set into equal-sized success and failure sets by human verification. For each influential layer, we compute its mean AAS on the success set, the failure set, and full set. The success gap is the difference between the success mean and the overall mean, and the failure gap is the difference between the failure mean and the overall mean. We call a layer **interaction-dominant** when the success gap is large and positive while the failure gap is large and negative relative to the overall mean; we rank layers by this separation, as depicted in Figure 7 (b).

**Relevance to Interaction-Awareness in Generated Videos.** Fig. 1 and 4 show when attention concentrates on the corresponding instance/union regions, generations are correct and preferred; when diffused or mislocalized, failures are common. These observations support the defined AAS as a reliable proxy for interaction fidelity. As a sanity check, we apply perturbation guidance (Ahn et al., 2025) to these layers. As shown in Fig. 19, attention sharpens toward instance regions and interaction fidelity improves slightly. Detailed protocol and results are in Appendix B and E.

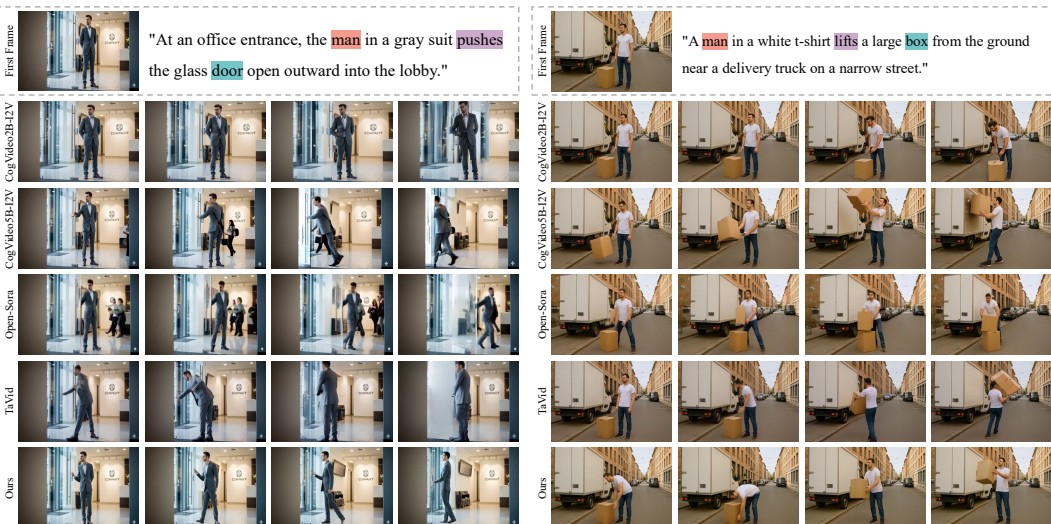

Figure 9: **Qualitative comparison.**

## 5 MATRIX FRAMEWORK

Sec. 4 identifies a small set of interaction-dominant layers whose attentions correlate strongly with semantic grounding and propagation. This analysis motivates MATRIX, which introduces Semantic Grounding Alignment (SGA) and Semantic Propagation Alignment (SPA) losses that directly align attention with ground-truth instance mask tracks.

**Baseline Architecture.** Building on CogVideoX-5B-I2V (Yang et al., 2024) with LoRA (Hu et al., 2021), the model conditions on noise latent $z_t$, the first RGB frame $x_0$, a first-frame multi-instance ID map $I_0$ with stable IDs, and prompt $P$ whose tokens mark subject, verb and object. We extend the input projection to ingest $x_0$ and $I_0$ by channel-wise concatenation with the latent. Here $I_0$ is the palette-indexed aggregation of per-instance binary masks $\{M_0^k\}$, so each ID $k$ keeps a fixed color across the clip. This grounds identities at generation start, and gives users explicit control over targets at inference, since $I_0$ can be obtained by off-the-shelf segmentors (Ravi et al., 2024).

**Attention Alignment.** We supervise attention directly with ground-truth instance mask tracks. We aggregate attentions at latent resolution, $\mathbf{A}^{\text{v2t}}, \mathbf{A}^{\text{v2v}} \in [0,1]^{F \times H \times W}$, and compare them to pixel-space mask tracks $M_e \in \{0,1\}^{F_{\text{pix}} \times H_{\text{pix}} \times W_{\text{pix}}}$ for $e \in \{\text{sub}, \text{obj}, \text{verb}\}$, where $F_{\text{pix}}, H_{\text{pix}}, W_{\text{pix}}$ denote the decoded video length and pixel resolution. To align scales, a lightweight causal decoder $\mathcal{D}_\phi(\cdot)$ that mirrors 3D VAE (Yang et al., 2024) upsampling schedule maps attention to RGB-space mask tracks at the correct spatiotemporal scale. Specifically, it expands time and space with the same strides as the 3D VAE with causal alignment of the first frame, so supervision is applied at the correct spatiotemporal scale. Specifically, let $\hat{\mathbf{A}}_e^{\text{v2t}} = \mathcal{D}_\phi(\mathbf{A}_e^{\text{v2t}})$ and $\hat{\mathbf{A}}_e^{\text{v2v}} = \mathcal{D}_\phi(\mathbf{A}_e^{\text{v2v}})$ denote decoder outputs at the pixel grid for $e \in \{\text{sub}, \text{verb}, \text{obj}\}$. We compare the decoder outputs to the target mask tracks $M_e$. Both SGA and SPA use the same composite loss $\ell$, a weighted sum of BCE, soft DICE and $L_2$ regression to the mask track. For prediction $X$ and target $Y$, $\ell$ is formulated as :

$$\ell(X, Y) = \beta_{\text{bce}}\text{BCE}(X, Y) + \beta_{\text{dice}}(1 - \text{Dice}(X, Y)) + \beta_2||X - Y||_2^2,$$

where $\beta_{\text{bce}}, \beta_{\text{dice}}$ and $\beta_2$ are coefficients, respectively. The SGA and SPA losses are defined as:

$$\mathcal{L}_{\text{SGA}} = \sum_{e \in \{\text{sub}, \text{obj}, \text{verb}\}} \ell(\hat{\mathbf{A}}_e^{\text{v2t}}, M_e), \quad \mathcal{L}_{\text{SPA}} = \sum_{e \in \{\text{sub}, \text{obj}\}} \ell(\hat{\mathbf{A}}_e^{\text{v2v}}, M_e),$$

We apply these losses only to the interaction-dominant layer found in the analysis, routing alignment where it is most effective, while leaving the other layer to preserve general video quality. Training minimizes a simple objective that adds these losses to the denoising loss:

$$\mathcal{L}_{\text{total}} = \mathcal{L}_{\text{DM}} + \lambda_{\text{SGA}}\mathcal{L}_{\text{SGA}} + \lambda_{\text{SPA}}\mathcal{L}_{\text{SPA}},$$

updating the LoRA parameters, the input projection layer and the lightweight decoder $\mathcal{D}_\phi$ while keeping the remaining backbone frozen. Here $\mathcal{L}_{\text{DM}}$ is the denoising loss, and $\lambda_{\text{SGA}}, \lambda_{\text{SPA}}$ are scalar weights of grounding and propagation, respectively. Additional details are provided in Appendix D.

Table 1: **Quantitative comparison.**

| Methods | InterGenEval | | | Human Fidelity | Video Quality | |
| --- | --- | --- | --- | --- | --- | --- |
| | KISA (↑) | SGI (↑) | IF (↑) | HA (↑) | MS (↑) | IQ (↑) |
| CogVideoX-2B-I2V Yang et al. (2024) | 0.420 | 0.470 | 0.445 | 0.937 | 0.993 | 69.69 |
| CogVideoX-5B-I2V (Yang et al., 2024) | 0.406 | 0.491 | 0.449 | 0.936 | 0.987 | 69.66 |
| Open-Sora-11B-I2V (Zheng et al., 2024) | 0.453 | 0.508 | 0.480 | 0.891 | 0.992 | 63.32 |
| TaVid (Kim & Joo, 2025) | 0.465 | 0.522 | 0.494 | 0.917 | 0.991 | 68.90 |
| **MATRIX (Ours)** | **0.546** | **0.641** | **0.593** | **0.954** | **0.994** | **69.73** |

Table 2: **Ablation studies.** All finetuning experiments were conducted on the MATRIX-11K.

| | Methods | InterGenEval | | | Human Fidelity | Video Quality | |
| --- | --- | --- | --- | --- | --- | --- | --- |
| | | KISA (↑) | SGI (↑) | IF (↑) | HA (↑) | MS (↑) | IQ (↑) |
| **(I)** | Baseline (CogVideoX-5B-I2V) (Yang et al., 2024) | 0.406 | 0.491 | 0.449 | 0.936 | 0.987 | 69.66 |
| **(II)** | TaVid (Kim & Joo, 2025) | 0.465 | 0.522 | 0.494 | 0.917 | 0.991 | 68.90 |
| **(III)** | **(I)** + LoRA w/ MATRIX-11K dataset | 0.445 | 0.526 | 0.486 | 0.940 | 0.994 | 69.77 |
| **(IV)** | **(III)** + SPA loss | 0.451 | 0.540 | 0.496 | 0.937 | **0.995** | **70.26** |
| **(V)** | **(III)** + SGA loss in $\mathbf{A}^{t2v}$ | 0.486 | 0.578 | 0.531 | 0.935 | 0.993 | 70.03 |
| **(VI)** | **(III)** + SGA loss in $\mathbf{A}^{v2t}$ | 0.509 | 0.592 | 0.550 | 0.952 | 0.994 | 69.62 |
| **(VII)** | **(III)** + SPA loss + SGA loss (**Ours**) | **0.546** | **0.641** | **0.593** | **0.954** | 0.994 | 69.73 |

# 6 EXPERIMENTS

## 6.1 EXPERIMENTAL SETUP

**Dataset.** We construct two evaluation sets, covering synthetic and real domains. The synthetic comprises 60 (image, prompt) pairs generated using (OpenAI & et al., 2024) where each prompt describes interactions among distinct instances, corresponding images are generated to match. For real domain, we curate 58 (image, prompt) pairs from open-source dataset (Nan et al., 2025; Chao et al., 2018), selecting examples using our curation pipeline. Additional details are in Appendix A.

**InterGenEval.** We evaluate interaction-aware semantics with a structured QA protocol. For each key interaction we auto-generate 10 questions: six stage-wise checks (KISA) of the pre, during, and post states, and four grounding checks (SGI) of the subject, object, verb-conditioned union, all phrased with appearance cues and bounding boxes. We report KISA and SGI, each reweighted by the temporal-consistency factor SPI, which penalizes emergence and disappearance across frames. The overall score, IF, is the mean of KISA and SGI. Details are in the Appendix F.

**Additional Metric.** Hallucination is measured by HA (Human Anatomy) from VBench2.0 (Zheng et al., 2025a). For video quality, we adopt metrics from VBench (Huang et al., 2023), including MS (Motion Smoothness) and IQ (Image Quality). Further details and results are in Appendix G.

## 6.2 COMPARISON AND ANALYSIS

Fig. 9 and Tab. 1 compare our method with open-source models (Yang et al., 2024; Zheng et al., 2024; Kim & Joo, 2025). The 2B model (Yang et al., 2024) rarely completes the action (fails to open the door or lift the box; Fig. 9), yielding low KISA, SGI and IF, yet its conservative motion produces clean frames with higher IQ and MS and fewer human anomalies, reflected by higher HA. The 5B model (Yang et al., 2024) attempts actions more often and slightly raises interaction scores, but identity drift and contact violations (twisted torso, floating box; Fig. 9) reduce SGI and HA. Open-Sora-I2V (Zheng et al., 2024) follows prompt strongly and lifts KISA, while unstable grounding and propagation lower SGI and HA and degrade overall quality via extra or missing instances. TaVid (Kim & Joo, 2025) benefits from an explicit target cue and improves grounding of one instance, but the lack of propagation signal limits temporal consistency and HA. In contrast, our method maintains correct bindings and tracks via SGA and SPA on the interaction-dominant layers, achieving the strongest interaction fidelity in KISA, SGI and IF, highest HA, IQ and MS. Additional qualitative results in various scenarios, including non-contact and non-human scenarios, are provided in Fig. 24 of Appendix.

"The woman holding the blue cup and the woman holding the red cup clink their cups together while laughing."    "The player in orange jersey closely defends the player in white jersey as they battle for the puck."

Figure 10: **Qualitative results of Wan2.1-14B-I2V (Wan et al., 2025) with our MATRIX.** While the baseline often ignores motion and produces nearly static videos, MATRIX helps preserve motion dynamics and enhances interaction fidelity.

### 6.3 ABLATION STUDIES

Tab. 2 aligns with our analysis and isolates the effects of our curated data, layer selection, and each interaction-aware loss. (I) Vanilla CogVideoX-5B-I2V (Yang et al., 2024), without any finetuning, performs worst on interaction metrics since it lacks any interaction signal. (II) LoRA tuning with single-object conditioning improves over (I) but fails to enforce propagation and degrades overall quality. (III) Naive LoRA finetuning on our curated dataset without layer selection or auxiliary losses yields balanced yet middling performance and corresponds to the baseline that simply finetunes on our data. In particular, the improvement from (I) to (III) captures the benefit of finetuning on MATRIX-11K train set with minimal architectural changes. (IV) Adding SPA on selected layers further enhances propagation, however, without explicit grounding it trades off noun/verb alignment, leading to higher smoothness and quality but lower grounding. (V) and (VI) correspond to two SGA variants: (V) applies SGA on selected layers of $\mathbf{A}^{\text{t2v}}$, while (VI) applies SGA on selected layers of $\mathbf{A}^{\text{v2t}}$. In practice, since a single text token can correspond to multiple locations, constraining text-to-video attention in (V) leads to unstable grounding. In contrast, (VI) supervises video-token queries directly, so the signal is applied to the spatial regions being generated and leads to more consistent gains. Therefore, adding SGA to (III) significantly boosts grounding (KISA, SGI, IF) by aligning noun/verb attentions, while keeping propagation. (VII) Combining SGA and SPA to (III) yields the best overall balance: strongest interaction fidelity (KISA, SGI, IF), the best human fidelity (HA) and improved video quality over the baselines, indicating that grounding first and then enforcing propagation offers complementary gains.

### 6.4 RESULTS WITH OTHER DiT BACKBONES

We also evaluate the MATRIX framework on another DiT-based backbone, Wan2.1-14B-I2V (Wan et al., 2025), making only minimal architectural modifications and finetuning solely with our SGA and SPA objectives. The procedures used to identify the dominant layers are described in Appendix C. As shown in Fig. 10, although Wan2.1-14B-I2V is already a strong backbone, simply applying MATRIX consistently improves interaction fidelity while preserving overall visual quality. These results demonstrate that MATRIX functions as a plug-and-play adapter: when applied on top of various DiT-based video models, it reliably enhances interaction fidelity without requiring extensive architectural redesign or sacrificing video quality. Additional qualitative results are presented in Fig. 42 and Fig. 43.

## 7 CONCLUSION

We study how video DiTs represent multi-instance interactions. To answer this, we firstly curate MATRIX-11K, video dataset pairing interaction-aware captions with multi-instance mask tracks. Using these tracks, we analyze 3D full attention and observe that semantic grounding and propagation concentrate in a small set of interaction-dominant layers. Guided by this, we introduced **MATRIX**, a lightweight regularization that aligns attention in those layers to mask tracks via SGA and SPA loss. On InterGenEval (KISA, SGI, IF), MATRIX improves interaction fidelity, strengthens noun and verb grounding, and reduces drift and duplication. Ablations confirm the critical role of layer selection and the complementary contributions of SGA and SPA.

ACKNOWLEDGMENTS

This research was supported by Institute of Information & communications Technology Planning & Evaluation (IITP) grant funded by the Korea government (MSIT) (RS-2019-II190075, RS-2024-00509279, RS-2025-II212068, RS-2023-00227592, RS-2025-02214479, RS-2024-00457882, RS-2025-25441838, RS-2025-25441838, RS-2025-02214479, RS-2025-02217259) and the Culture, Sports, and Tourism R&D Program through the Korea Creative Content Agency grant funded by the Ministry of Culture, Sports and Tourism (RS-2024-00345025, RS-2024-00333068, RS-2023-00222280, RS-2023-00266509), and National Research Foundation of Korea (RS-2024-00346597).

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

# APPENDIX

In this material, Sec. A provides details of our MATRIX-11K dataset curation pipeline. Sec. B expands our analysis with additional visualizations and discussions and Sec. C extend our analysis to additional DiT-based video models, including HunyuanVideo-I2V (Kong et al., 2024) and Wan2.1-14B-I2V (Wan et al., 2025). Sec. D and Sec. E describe details of our proposed model and guidance strategy. Sec. F introduces details of our novel interaction-aware evaluation protocol, while Sec. G reports evaluation results across metrics, datasets, and human evaluation studies. Sec. H presents additional qualitative and quantitative results with analysis. Finally, Sec. I mentions the limitations of our work Sec. J discusses the future direction of our work.

## A    DATASET CURATION DETAILS

As illustrated in Fig. 5 of the main paper, our MATRIX-11K dataset was curated and filtered through a step-wise process. We describe the detailed prompt design used when leveraging a large language model (LLM) (et al., 2024) and Vision-language model (VLM) (OpenAI & et al., 2024), along with the examples of the resulting filtered data.

### A.1    DETAILS FOR INTERACTION-AWARE CAPTION PROCESSING DETAILS

**Interaction Identification and Instance Assignment.**    Fig. 26 illustrates prompt design for interaction identification and instance assignment. Given a natural-language prompt $P$ for a video $V$, the goal of this stage is to extract the ID set $\mathcal{K}$ and interaction triplets $\mathcal{R}$. The first turn is a validator that counts only active interaction linking a living subject to a distinct object via an explicit action verb, rejecting self-directed actions, vague verbs, and internal states. Then it outputs $\mathcal{A}(P)$, which is the valid actions, or *null* if none exist. Second turn then enumerates all instance mentions that participate in some $a \in \mathcal{A}(P)$, assigns a stable instance index $k$ and semantic class $\mathrm{cls}_k$, forming the ID set $\mathcal{K} = \{(k, \mathrm{cls}_k) \mid k \text{ participates in some } a\}$, and record role-type relation as $\mathcal{R} = \{(a, k_{\mathrm{sub}}, k_{\mathrm{obj}}) \mid a \in \mathcal{A}(P), (k_{\mathrm{sub}}, \cdot), (k_{\mathrm{obj}}, \cdot) \in \mathcal{K}\}$. The figure also shows the normalized output used in practice: one JSON object per interaction containing the surface form $\langle \mathrm{idX\ verb\ idY} \rangle$, subject and object IDs, an interaction-type label (multi-subject relation or functional action), and the exact source sentence span. These outputs $\mathcal{K}$ and $\mathcal{R}$ serve as the formal supervision for all subsequent interaction-aware curation and evaluation steps.

**Interaction Scoring and Filtering.**    Fig. 27 presents the prompt design for interaction scoring and filtering. For each extracted interaction triplet $(a, k_{\mathrm{sub}}, k_{\mathrm{obj}}) \in \mathcal{R}$ from the prompt of video $V$, a LLM rater (et al., 2024) consumes the full textual context including prompt $P$, $\langle \mathrm{idX\ verb\ idY} \rangle$, and noun descriptors of each IDs, and returns two integer scores $\in \{1, ..., 5\}$. *Contactness* quantifies physical contact or tight spatial coupling implied by the action (1 = no contact, 3 = indirect/uncertain, 5 = direct/certain contact). *Dynamism* measures degree of motion or temporal change (1 = static relation, 3 = low/moderate movement or readiness, 5 = strong action/state change). In addition, the rater also outputs a concise natural-language justification for its judgment and a self-reported confidence, which we use for auditability and to discard uncertain cases. Interactions judged to exhibit sufficient contact and motion are retained, and instances not appearing in any retained triplet are pruned.

**Instance Description Extraction.**    Fig. 28 shows the prompt design for instance description extraction. Given the prompt $P$, a selected interaction triplet $(a, k_{\mathrm{sub}}, k_{\mathrm{obj}})$, and the base nouns $\mathrm{cls}_k$ for the participating IDs, the LLM rater (et al., 2024) produces, for every referenced instance $k \in \mathcal{K}$, a compact descriptor $\mathrm{desc}_k = (\mathrm{noun}, \mathrm{app}, \mathrm{spatial})$. Here "noun" is a short, visually discriminable noun phrase (*e.g.,* "a man in a blue shirt"), "app" is a one-sentence summary of salient appearance or physical attributes, and "spatial" is a one-sentence statement of location or role in the scene. Descriptors are canonicalized, coverage-complete (one per ID), and linked to $(k, \mathrm{cls}_k)$, redefining ID set as $\mathcal{K} = \{(k, \mathrm{cls}_k, \mathrm{desc}_k)\}$. We use this set to support grounding and to verify detected bounding boxes or masks by matching appearance and spatial cues, improving disambiguation among same-class entities.

## A.2 Details for Interaction-aware Multi-instance Mask Tracks with Verification

**Why multi-instance mask tracks?** While recent video generation methods leverage additional modalities such as optical flow (Chefer et al., 2025) and depth (Yang et al., 2025), we adopt instance-level segmentation and tracking (mask tracks) as our core interaction signal. In our setting, the reference modality must (i) provide instance-level semantic precision to verify whether a specific object is correctly grounded, (ii) exhibit temporal continuity so that the grounding can be tracked across frames, and (iii) disambiguate multiple instances that share the same class (e.g., two people). Among common cues such as optical flow, depth, and segmentation, only instance mask tracks naturally satisfy all three conditions: optical flow offers dense motion but no instance IDs, and depth encodes geometry but neither instance grouping nor same-class disambiguation. Mask tracks, in contrast, yield clear instance regions and persistent IDs over time, which is exactly what we need to analyze and supervise "who does what to whom" in subject–object–verb interaction structures.

**Interaction-aware multi-instance tracks with verification.** Fig. 29 illustrates the prompt design for vision-language verification, which checks the consistency between bounding-box visual prompts and instance appearance. We generate tracks in four steps.

*(1) Class-only proposals.* Given a video $V$, its prompt $P$ and its ID set $\mathcal{K} = \{(k, \text{cls}_k, \text{desc}_k)\}$, we uniformly sample $T$ frames and run GroundingDINO (Liu et al., 2024) with $\text{cls}_k$ only. For each frame $i$, it returns up to $J$ candidates boxes $(b_k^{i,j}, c_k^{i,j})$, where $b_k^{i,j}$ is box coordinate and $c_k^{i,j} \in [0, 1]$ is the class-conditioned confidence for the given class $\text{cls}_k$. Thus, for each id $k$, the video yields at most $JT$ candidates across the $T$ sampled frames. This class-only setting provides high recall but cannot disambiguate same-class instances and may still miss the intended target on difficult frames.

*(2) Anchor selection and VLM verification.* For each noun ID $k$, we collect at most $J \times T$ candidates $\{(b_k^{i,j}, c_k^{i,j})\}$ over the $T$ sampled frames. We sort them by confidence and define the *anchor* as the highest-scoring pair as: $(i^\star, j^\star) = \arg\max_{i,j} c_k^{i,j}$ with $b_k^\star = b_k^{i^\star, j^\star}$. We then query a vision-language model (VLM) (OpenAI & et al., 2024) with for inputs, including frame $i^\star$, the crop from $b_k^\star$, the class name $\text{cls}_k$ and the descriptor $\text{desc}_k$, and ask whether the crop matches the description of ID $k$. If the VLM verifies the match, we accept $b_k^\star$ as the *final* box for ID $k$ and initialize SAM2 (Ravi et al., 2024) propagation from that frame to obtain the mask track of the ID. If not, we move to the next candidate in descending $c_k^{i,j}$ and repeat the aforementioned process. When no candidate is verified, the ID is dropped; if both subject and object are dropped, the clip is excluded.

When multiple IDs share the same class (*e.g., two persons*), verification is one-to-one: once a candidate box is accepted for an ID, it is removed from the pools of the other IDs of the same class. This mutual-exclusion pruning prevents duplicate assignments and reduces verification cost from a naive $O(|\mathcal{K}|JT)$ scan to a much smaller set of checks in practice, while keeping recall high and disambiguation accurate.

*(3) Human verification.* As a final check, we run a lightweight but explicit quality-control pass on the verified tracks. For each clip, annotators review 10 frames, including the first verified frame, the last valid frame and eight uniformly spaced interior frames. They view the RGB, instance mask tracks and boxes, the union mask tracks and the triplet descriptor. Each track is labeled *Accept* (clean and consistent), *Fix* (minor boundary/jitter; quick snap/smooth), or *Drop* (identity drift/swap, duplication, hallucination, or clear temporal gaps). A clip is used for supervision only if both subject and object are *Accept* after any minor fixes; otherwise it is excluded. For same-class IDs we enforce one-to-one assignment by dropping the worse of any substantially overlapping tracks. About 5% subset of accepted clips is spot-checked by a second annotator.

**Effect of the proposed VLM verification.** As described in Sec. 3 of the main paper, we employ a VLM (OpenAI & et al., 2024) to verify and refine the error of GroundingDINO (Liu et al., 2024). Fig. 11 illustrates why this step is necessary. With only a class name (*e.g., person, man, cake*), GroundingDINO frequently returns multiple same-class instances over pre-defined threshold and cannot single out the intended target. In Fig. 11, (a) captures both the person inside and outside the shop. (b) captures the photographer, the person being photographed and even a reflection in a phone. (c) captures every cake in view and (d) captures both the stylist and the client. A straightforward solution is to add appearance phrases (*e.g.,* the man outside the shop) to figure out the intended

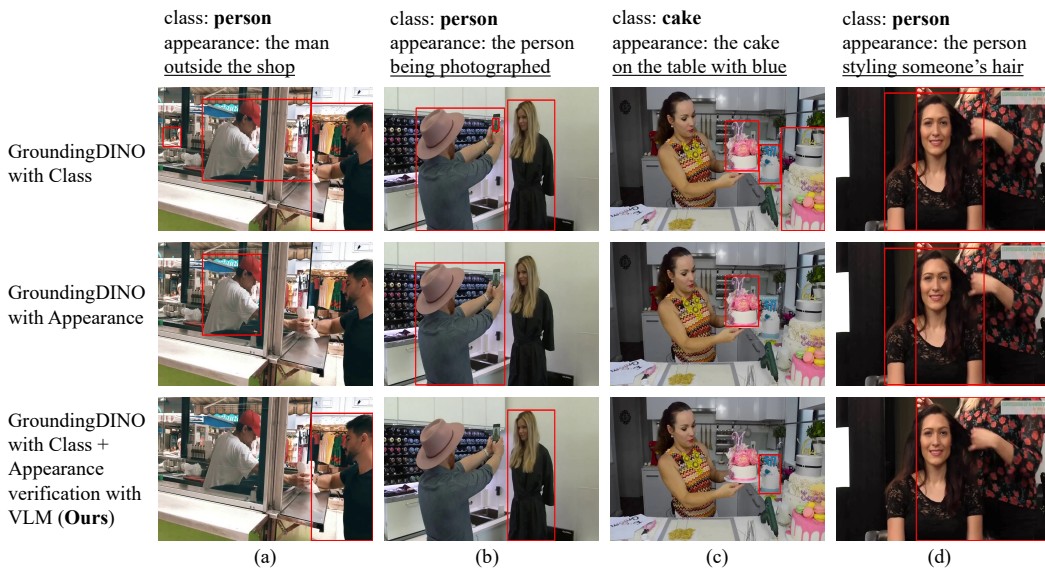

Figure 11: **Effects of Our VLM Verification.** The first row (GroundingDINO with class) and the second row (GroundingDINO with appearance) often pick a wrong same-class instance. The third row (ours) verifies candidates with a VLM and keeps exactly one box per noun, resolving (a) to (d). Best viewed in zoom.

target. However, it is unreliable since GroundingDINO often latches onto partial tokens and ignores modifiers. For instance, in (a) it selects the man *inside* the shop by focusing on "man" and "shop" while missing "outside", and in (b) it selects the person *taking* the photo instead of the intended person being photographed. In (c) it selects wrong cake rather than the blue cake on the table, and in (d) it still captures both people, failing to disambiguate the stylist from the client.

Rather than injecting appearance phrases into GroundingDINO, we use it purely as a class-consistent proposal generator, since with class names alone, it reliably enumerates candidate boxes but cannot disambiguate same-class instances. Motivated by recent results (Cheng et al., 2025; Jia et al., 2025; Chen et al., 2025) showing that VLMs (Bai et al., 2023; Li et al., 2024; Ye et al., 2024; OpenAI & et al., 2024; Dai et al., 2023) excel at image and multi-image reasoning, we introduce a VLM verification stage that cross-checks each candidate against descriptors, including appearance cues, and selects exactly one box per noun. If no candidate satisfies the verifier, we drop that instance and we remove the clip from the supervision. This preserves high recall from GroundingDINO while delegating fine-grained disambiguation to the VLM, yielding cleaner per-instance tracks. As presented in the last row of the Fig. 11, VLM evaluates candidates against the provided appearance descriptor and selects the final bounding box that matches the cue.

## A.3 DATASET EXAMPLES AND STATISTICS

We provide more dataset examples in Fig. 30, Fig. 31, Fig. 32 and Fig. 33. Additionally, Fig. 12 shows the overall statistics of our curated dataset.

In Fig. 12, (a) summarizes the distribution of video-text sources we used in our study. Our primary source is HOIGen (Liu et al., 2025), whose captions explicitly describe humans, human-object interaction, human action and scene descriptions. Therefore, the text is strongly interaction-aware and provides dense cues for extracting interactions. Since HOIGen collects videos from diverse sources, it spans everyday to highly specific scenarios and offers abundant interaction instances. To improve generalization and ensure data quality, we further incorporate PE-Video (Bolya et al., 2025), a high-quality, carefully annotated collection that covers a wide range of categories. (b) reports the joint distribution of contact and dynamism score in our curated corpus. We score contact on a 1-5 scale (none - contact-rich) and dynamism on a 1-5 scale (static - highly dynamic). While the corpus includes static or non-contact cases, it is enriched for dynamic, contact-rich interactions. Crucially, within each contact level(from 1 to 5), dynamism spans a broad range, ensuring diverse motion intensities conditioned on contact level. Additionally, (c) summarizes the distribution of per-video

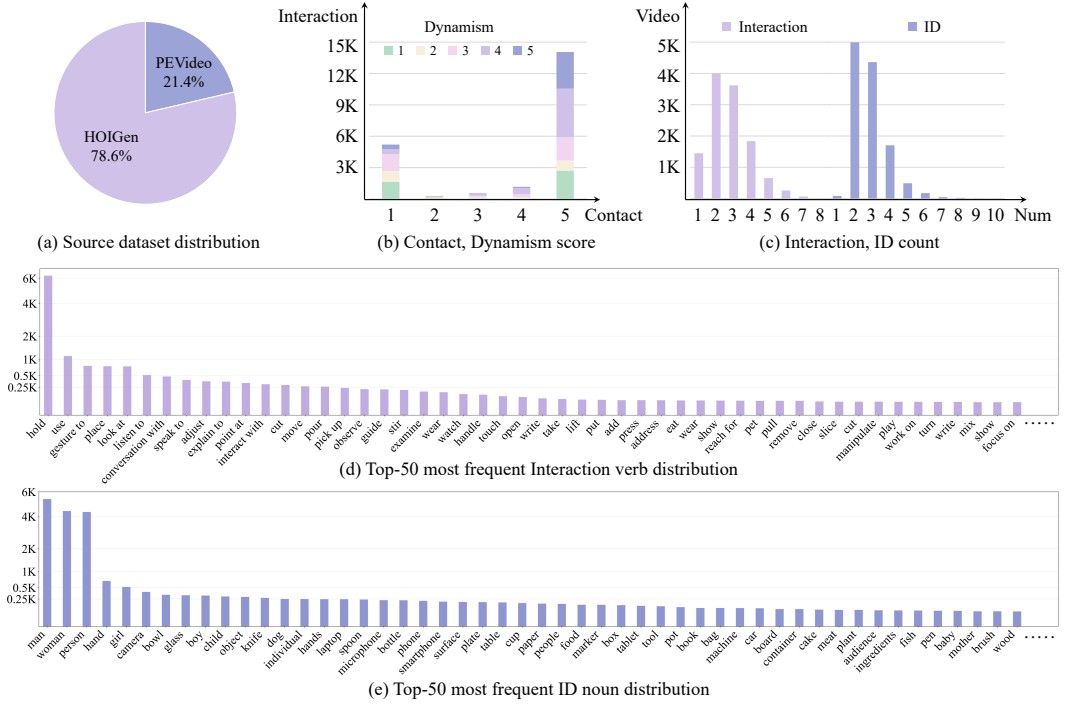

Figure 12: **Dataset Statistics.**

counts of interactions (1-8) and identities (1-10). The mass concentrates in the 1-5 range for both, with clear modes at two interactions and two identities, indicating that pairwise subject-object settings dominate. Motivated by this distribution, we cap instance identities at $|\mathcal{K}| = 5$ per clip: the annotator collects up to five tracks and the model predicts up to five instance mask tracks. This choice balances coverage and compute while remaining extensible, raising $|\mathcal{K}|$ only increases the number of track slots without altering the rest of the pipeline. Moreover, clips with more than five interactions or instances are empirical outliers in Fig. 12 (c), providing evidence that such highly crowded cases are rare. When they do occur, we either split the video into shorter sub-clips or retain the top-$k$ salient instances and aggregate metrics at the original-video level. Considering with (a) and (b), these statistics indicate an interaction-dense yet not overcrowded corpus, aligned with our modeling in Sec. 5 and evaluation design in Sec. 4.

Finally, the dataset exhibits strong linguistic coverage. In Fig. 12 (d), we plot the top-50 interaction verbs. Since contact frequently entails "hold", that verb dominates. Excluding "hold", the remaining verbs follow a comparatively balanced distribution, indicating broad action diversity rather than reliance on a handful of predicates. Fig. 12 (e) shows the top-50 identity nouns. As interaction typically involves at least one human subject, nouns such as "man", "person", and "woman" are frequent. Nevertheless, object nouns are broadly distributed, reflecting diverse targets and scenes. Together, (d) and (e) indicate wide linguistic coverage over actions and instances, supporting robust training and evaluation of interaction-aware models.

# B  ANALYSIS DETAILS

## B.1  ANALYSIS EVALUATION DATASET

To faithfully evaluate interaction-aware video generation, we curate a dedicated analysis evaluation dataset rather than relying on real-world videos. Using real videos for reconstruction is problematic due to inversion errors (Song et al., 2022), imperfect prompt-video alignment, and distributional drifts, making it difficult to isolate model behavior. To circumvent these issues, we curate a controlled analysis evaluation dataset designed to simulate the generation process itself. By fixing random seeds during synthesis, we approximate near-perfect reconstruction conditions. Human annotators further verify the outputs, ensuring that only videos with high overall fidelity and consistent

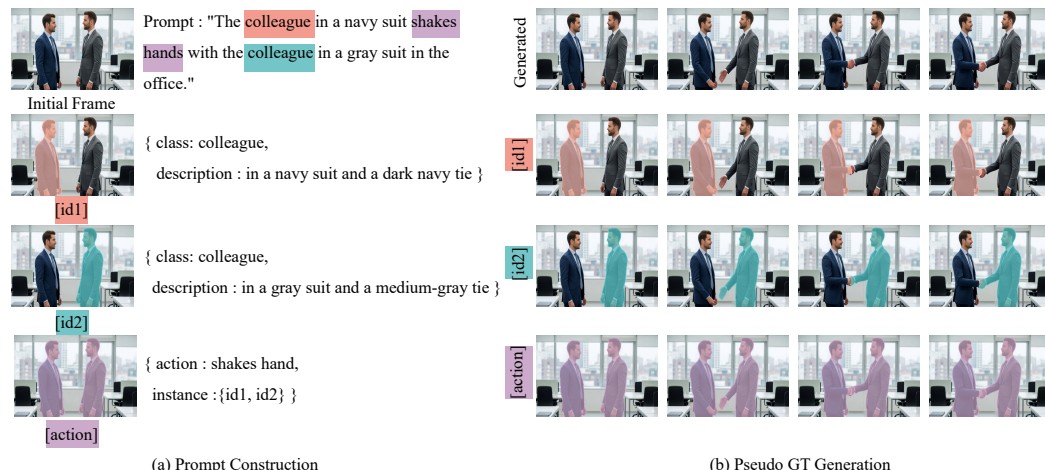

Figure 13: **Analysis Dataset Pairs Example.**

interactions are retained. Each video in the benchmark has a resolution of $480 \times 720$, contains 49 frames, and the final dataset consists of 50 carefully validated prompt-video pairs.

**Scenario Design.** The curation process begins with scenario design proposed by (OpenAI & et al., 2024), where we systematically specify the conditions of interaction to ensure diversity and coverage. Specifically, we distinguish between unidirectional interactions, where a subject acts upon an object (*e.g.,* a person pushing a box), and bidirectional interactions, where both subject and object mutually influence each other (*e.g.,* two people shaking hands). We then vary the number of participating instances, ranging from simple subject-object pairs to multi-party settings with three or more instances, which introduce additional ambiguity in role assignment. Interactions are further categorized into contact (*e.g.,* touching, holding), force (*e.g.,* pushing, pulling), transport (*e.g.,* handling over, carrying), manipulation (*e.g.,* cutting, opening) and social (*e.g.,* hugging, waving), thereby covering a broad spectrum of physical and social dynamics. Finally, we ensure class diversity by including human-object, human-human, human-animal, human-nature interactions, encouraging generalization beyond human-centric scenarios. Together, these design choices allow us to construct structured prompts that specify the instances, their roles, and their relations, ultimately yielding a balanced set of interaction scenarios for evaluation.

**Prompt Construction.** Given a scenario, we then construct prompts that specify instance identities (IDs), class labels, and concise descriptors, along with the intended interaction, following the same principles as our dataset curation process described in Sec. 3.1. We first compose an image prompt that captures the static scene and instance attributes. Next, we derive a motion-aware video prompt by adding action and relation clauses (subject-verb-object) with temporal qualifiers (*e.g.,* contact). To improve synthesis stability and phrasing consistency, we apply VLM (OpenAI & et al., 2024)-based prompt enhancement while preserving instance IDs and interaction roles. For controlled synthesis and verification, we generate videos with fixed random seeds and standardized rendering settings, holding resolution and length constant. Human annotators review each prompt-video pair for overall visual quality, semantic fidelity to the prompt, and interaction plausibility. Only pairs passing all criteria are retained in the analysis dataset. Fig. 13 (a) presents an example produced by our prompt-construction procedure.

**Pseudo Ground-truth Generation.** Finally, to quantitatively evaluate semantic grounding and semantic propagation, we produce pseudo ground-truth mask tracks for each instance, since synthesized videos do not contain ground-truth supervision. Following the same grounding-and-verification procedure used in dataset curation as Sec. 3.2 in the main paper, we first extract candidate bounding boxes using GroundingDINO (Liu et al., 2024), verify them with a vision-language model (OpenAI & et al., 2024) to eliminate irrelevant detections, and propagate the validated boxes using SAM (Ravi et al., 2024) to obtain per-instance mask tracks. A final human verification step ensures correctness of both instance identity and mask track quality, yielding high-quality mask

tracks that serve as supervision for interaction analysis. Fig. 13 (b) shows an example constructed by our pseudo ground-truth generation.

As a result of the above systematic and precise procedure, we obtain images, prompts and per-instance mask tracks for each instance ID. We use this analysis evaluation dataset to evaluate semantic grounding and semantic propagation, as presented in Sec. 4.

**Real-domain Analysis Evaluation Set.** To validate whether our findings hold beyond controlled synthesis, we additionally curate a real-domain set using PE-Video (Bolya et al., 2025) and Open-Vid (Nan et al., 2025). As discussed in Sec. B, reconstructing real videos via prompt inversion is prone to inversion errors (Song et al., 2022) since accurate text prompts are difficult to recover and the real-video distribution differs from the training distribution. Accordingly, we select video-text pairs whose captions instantiate our interaction schema, extract the captions to our ID, role, and action format, and reconstruct each clip with an image-to-video (Yang et al., 2024) using the paired caption. Human annotators then verify that the generated clip preserves the intended interaction, roles, and overall appearance; only verified pairs are retained. The same rater used in scoring and filtering provides contactness, dynamism and brief justification with confidence, and pseudo mask tracks are produced with the grounding and verification pipeline (Sec. 3) and checked for instance identity and mask track quality.

## B.2 ADDITIONAL ANALYSIS

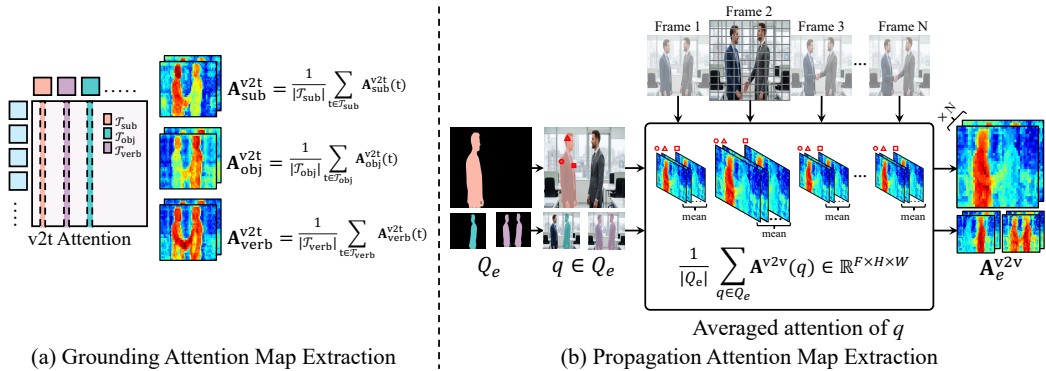

(a) Grounding Attention Map Extraction     (b) Propagation Attention Map Extraction

Figure 14: **Attention Map Details for Grounding and Propagation.**

Fig. 14 visualizes the procedures used in the main paper to extract grounding attention maps from video-to-text attention and propagation attention maps from video-to-video attention.

**Metric Choice.** In Sec. 4 in the main paper, we introduce the Attention Alignment Score (AAS) as the primary analysis metric. Our goal is to test whether the model's attention encodes *"who does what to whom"* at the level of targeted instances and whether the targeted spatial region for each instance is preserved consistently over time. In other words, attention should be concentrated on the instances (*e.g.,* subject, object) along their mask tracks, providing spatial binding within frames and temporal persistence across frames. We define AAS as the spatio-temporal inner product of $\mathbf{A}_e^{\text{v2t}}, \mathbf{A}_e^{\text{v2v}}$ $e \in \{\text{sub}, \text{obj}, \text{verb}\}$ and mask track, which measures how much attention mass is placed on the exact support of the instance over space and time.

This formulation is driven by the evaluation goal and by the normalization behavior of 3D full attention. Queries attend over the union of visual and text keys, and the softmax normalization is taken across that union. In our setting, the text stream contributes roughly 226 tokens, whereas the video stream contributes about $1350 \times 13 = 17550$ visual tokens (1350 spatial locations per latent frame across 13 latent frames). Even when video-to-text attention is correctly localized, the relative scale of attention to text tokens might be compressed by this large cardinality imbalance. Preserving raw magnitude is therefore informative since it quantifies how much attention mass is allocated on the instance's track versus how much is allocated to non-target tokens and regions, rather than merely indicating whether there is any overlap with the binary mask track. AAS integrates raw attention

on the mask track without thresholding or calibration, so it remains comparable across layers and is robust to this token-count imbalance.

A straightforward alternative is to treat $\mathbf{A}_e^{\mathrm{v2t}}, \mathbf{A}_e^{\mathrm{v2v}}$   $e \in \{\mathrm{sub}, \mathrm{obj}, \mathrm{verb}\}$ as a soft segmentation sequence and measure overlap with its corresponding mask track using standard segmentation scores (Rezatofighi et al., 2019; Lin et al., 2018; Sudre et al., 2017). For example, one can threshold to obtain a binary sequence and compute IoU (Rezatofighi et al., 2019) against mask track, or use threshold-free scores such as BCE or Dice. However, these options either introduce sensitivity to an arbitrary threshold or discard absolute magnitude and retain only shape overlap. The loss of magnitude is particularly limiting under 3D full attention, where cross-modal competition suppresses text-side scales. At the opposite extreme, simply aggregating raw attention over the whole scene preserves magnitude but no longer tests whether attention lies on the intended instance trajectory.

AAS provides a direct measure of what we seek to evaluate. Accordingly, we use AAS in Sec. 4 to locate interaction-dominant layers and to link attention concentration with semantic grounding and semantic propagation.

**Qualitative Link between Attention Alignment and Generation Quality.**   We provide qualitative evidence that attention-mask alignment relates to interaction fidelity. In the Fig. 1 (b), video-to-text (v2t) grounding improves generation when noun and verb attentions align with the subject, object ad union regions, whereas misalignment coincides with failures. Fig. 4 visualizes video-to-video (v2v) propagation maps. For each instance, (*e.g.,* boy, girl), first frame mask pixels serve as query points. For each query, we read its v2v attention to all spatial tokens across frames, reshape the result into an $F \times H \times W$ map, and overlay it on the video. In successful examples, attention initialized within the instance mask remains compact, follows the same instance through time, and yields clean, consistent clips. In failure cases, even with accurate first-frame grounding, propagation diffuses within the mask, leaks outside, or jumps to other regions, producing identity drift and hallucinated parts. These observations indicate that both v2t grounding and v2v propagation alignment matter for generation quality, which motivates supervising them explicitly with SGA and SPA.

**Layer-wise Analysis.**   Fig. 34 compares, for each noun and verb token, the attention maps from all 42 layers of naive CogVideoX-5B-I2V (Yang et al., 2024). Two patterns emerge. First, a small subset of layers shows strong alignment with the instance mask region for nouns and with the subject-object union for verbs. Second, many other layers exhibit grid-like responses consistent with positional embedding effects rather than semantic binding. This heterogeneity implies that layers play distinct roles, so finetuning every layer can dilute or damage the layers that carry useful semantics. Notably, even vanilla CogVideoX already displays alignment in several layers highlighted by our analysis (*e.g.,* layers 7, 8, 9, 10 and 11), further motivating our focus on interaction-dominant layers.

## C   MATRIX ON OTHER VIDEO DiTs

MATRIX is designed to be model-agnostic and can be plugged in any off-the-shelf video DiT architecture. To demonstrate this, we extend our analysis to two additional DiT-based image-to-video models, including HunyuanVideo-I2V (Kong et al., 2024) and Wan14B-I2V (Wan et al., 2025), and analyze their behavior under our framework.

**MATRIX on HunyuanVideo-I2V.**   In Fig. 15, we present additional analysis of HunyuanVideo-I2V using MATRIX framework. The backbone model consists of 60 layers, and for this experiment, we use 50 sampling steps. We observe a clear pattern that a small number of layers carry most of the interaction signals. For HunyuanVideo-I2V, the dominant video-to-text layers are concentrated in the early stage (layers 6,7,9,10), whereas the dominant video-to-video layers appear in middle layers (layers 11, 26, 27, 29). This behavior is consistent with our findings on CogVideoX-5B-I2V, where video-to-text dominant layers are located at 7 and 11 and video-to-video dominant layers at 12 and 19.

**MATRIX on Wan2.1-14B-I2V.**   We present further analysis of Wan2.1-14B-I2V using our framework. The backbone consists of 40 layers and we use 50 sampling steps in this experiment. However, the architecture of Wan slightly differs from the other DiT-based models we analyze. Instead

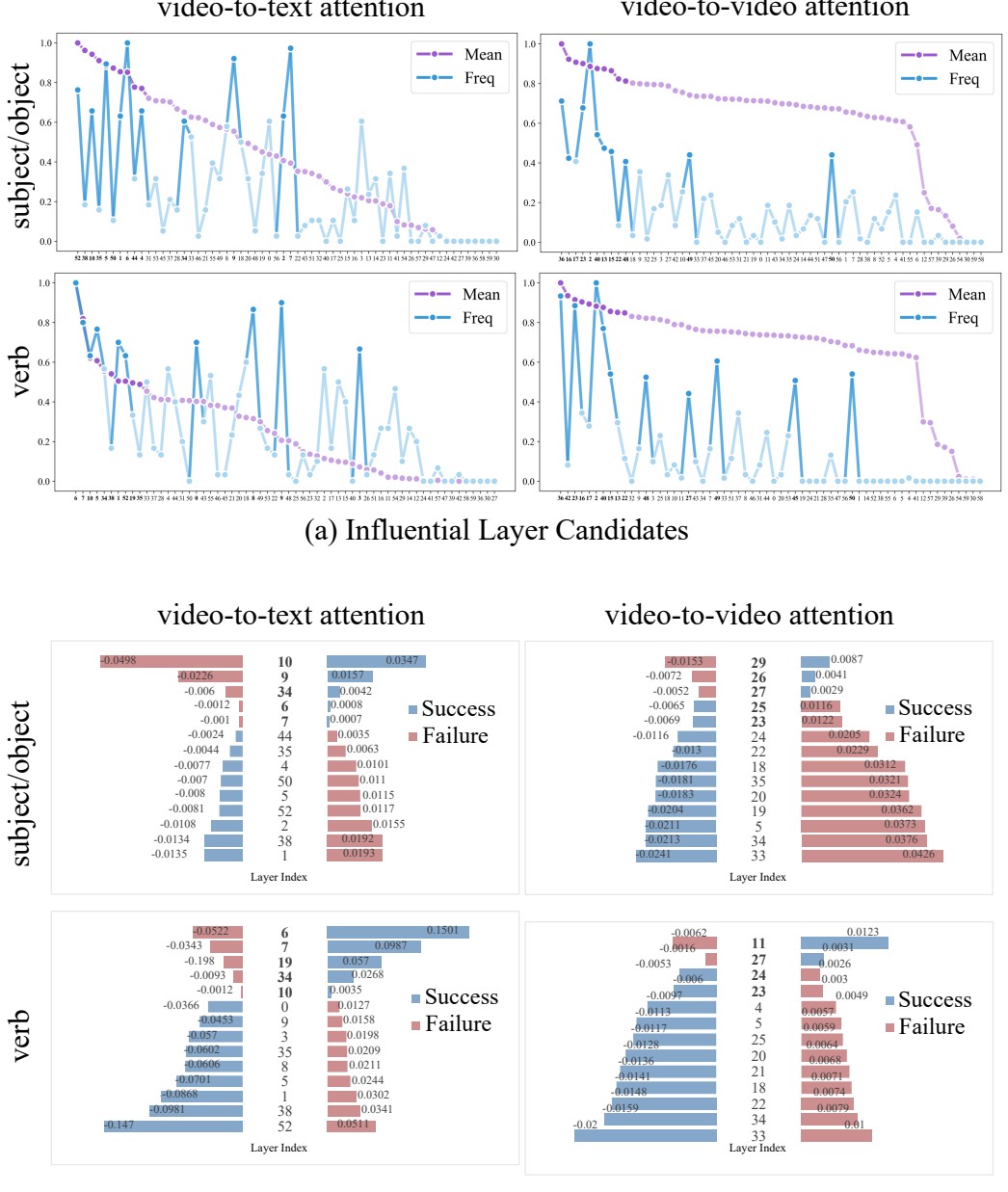

(a) Influential Layer Candidates

(b) Dominant Layer Selection

Figure 15: **Layer Analysis of HunyuanVideo-I2V.** (a) Influential layers, (b) Dominant layers.

of a single 3D full-attention block that jointly processes visual and textual tokens, it employs self-attention to process visual tokens and separate cross-attention modules to inject textual information. Due to this architectural separation, the distribution of our interaction-aware evaluation differs from the 3D full-attention cases. In CogVideoX-5B-I2V (Yang et al., 2024) and HunyuanVideo-I2V (Kong et al., 2024), textual tokens occupy only a small fraction of the joint token space ($T_{text}$ out of $F \times H \times W + T_{text}$), whereas in Wan2.1-14B-I2V the video-to-text branch operates purely on the $T_{text}$ tokens. As a result, we observe layers whose video-to-text scores are almost constant across different prompts and success/failure cases. In other words, they are effectively insensitive to the interaction content. To quantify this behavior, we analyze the variance of each layers' score across prompts and identify a subset of near-constant, prompt-insensitive layers, which we treat as "dummy" layers). As shown in Fig. 16, these dummy layers exhibit extremely low variance, espe-

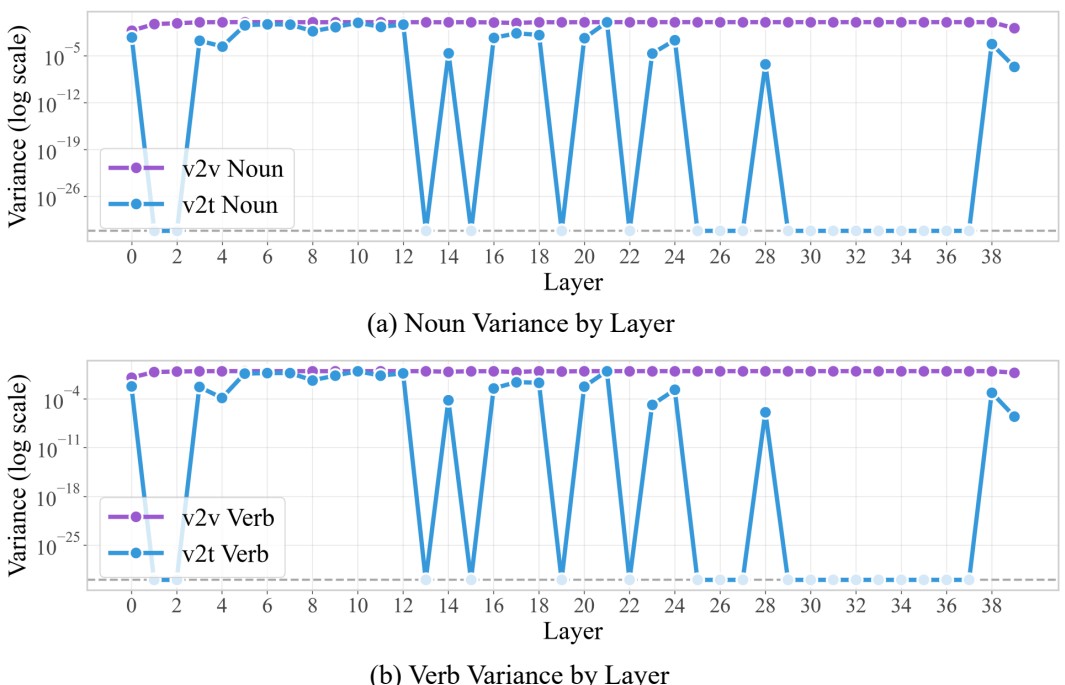

(a) Noun Variance by Layer

(b) Verb Variance by Layer

Figure 16: **Filtering Dummy Layers of Wan2.1-14B-I2V (Wan et al., 2025).** (a) Variance of noun-token cases across layers for video-to-text versus video-to-video attention. (b) The same analysis for verb-token cases. In both plots, several video-to-text layers exhibit almost zero variance ($10^{-29}$), meaning their attention responses remain nearly constant regardless of prompt. These layers are represented as gray points and are treated as *dummy* layers. In contrast, no such dummy layers are observed in video-to-video attention, so dummy layer filtering is applied only on the video-to-text before selecting dominant layers.

cially in the video-to-text branch. We therefore filter them out before selecting influential layers, focusing our analysis on layers that genuinely respond to interaction signals.

After removing these dummy layers, Wan2.1-14B-I2V still shows a clear dominance structure. Video-to-video dominant layers are concentrated in the early part of the network (layers 6–8), while video-to-text dominant layers emerge much later (layers 20–24). Compared to CogVideoX-5B-I2V or HunyuanVideo-I2V, this shift is consistent with Wan's self-then-cross design, where earlier self-attention layers over visual tokens mainly handle propagation across frames, while later cross-attention layers bind these interactions to the text. Fig. 42 and Fig. 43 provide additional qualitative comparisons between baseline (Wan et al., 2025) and MATRIX.

# D  MATRIX FRAMEWORK DETAILS

## D.1  ARCHITECTURAL DETAILS

We build on the pretrained CogVideoX-5B-I2V (Yang et al., 2024) and retain its transformer blocks and 3D VAE except for the input pathway and a small set of parameter-efficient adapters. Our network requires (i) the noise latent, (ii) a first RGB frame and (iii) instance masks that supervise attention alignment. These signals are concatenated along the channel dimension and projected by the input projection layer of the backbone. To preserve the pretrained capability at initialization, we copy the original weights into the slice that corresponds to the original channels and zero-initialize the newly added channel kernels. This keeps the base behavior unchanged at step 0, while allowing the added channels to learn during finetuning. We attach adapters to the query, key, value and output projections inside the corresponding attention modules while leaving all other weights frozen.

To manipulate internal attentions without overfitting, we adopt LoRA (Hu et al., 2021) on a minimal set of layers identified by our analysis. *Layer 7 and layer 11* are used for semantic grounding based

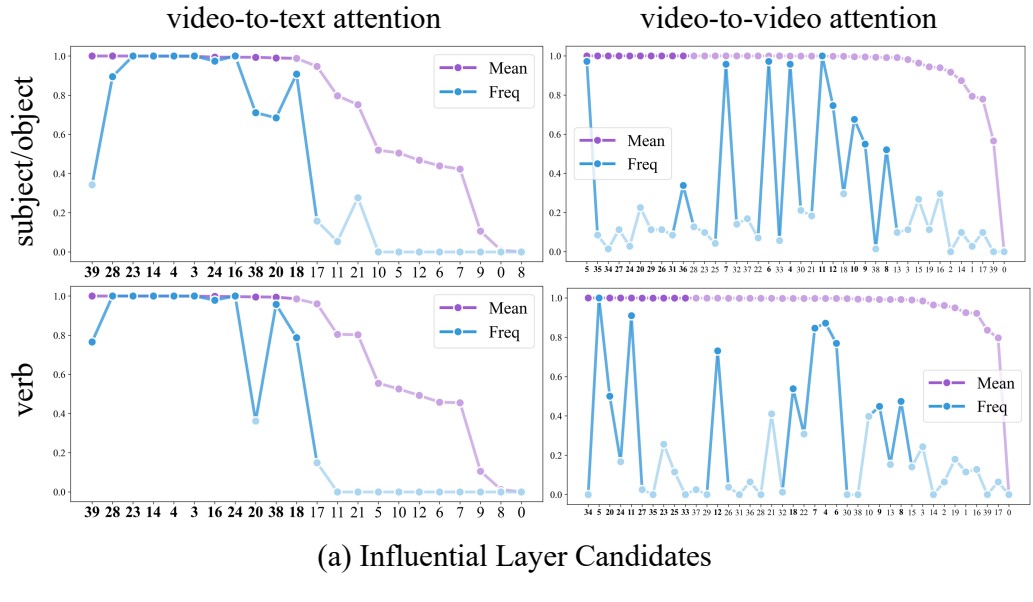

(a) Influential Layer Candidates

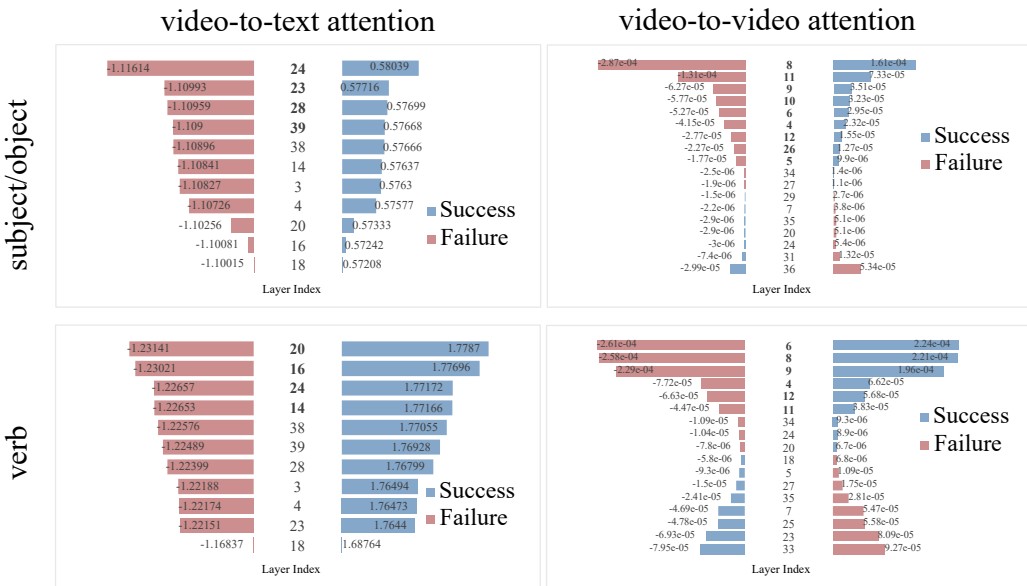

(b) Dominant Layer Selection

Figure 17: **Layer Analysis of Wan14B-I2V.** (a) Influential layers, (b) Dominant layers.

on video-to-text attentions and *layer 12* is used for semantic propagation based on video-to-video attentions. These are the only transformer weights that receive trainable LoRA parameters and the rest of the backbone remains fixed.

With these adapters, we supervise attention directly rather than supervising proxy features. We add two lightweight decoders, a grounding head and a propagation head that read the query-key product scores from the targeted layers, such as layer 7, 11 and 12, and convert them into alignment scores trained against binary ground truth mask tracks in RGB space while the generator remains unchanged.

For semantic grounding, it uses the video-to-text attention where video tokens act as queries and instance token in the text act as keys. For semantic propagation, it uses the video-to-video attention

that links each location in one frame to matching locations in the next few frames and checks whether the same instance persists over time. After computing the query-key product we reshape the result to the backbone spatiotemporal token grid so that each value aligns with a patch and a frame. We then take a simple mean across attention heads and feed the resulting map to a lightweight decoder. The decoder serves as a supervised readout that turns token space attention into dense alignment scores against binary mask tracks. This separation lets the alignment loss update only the query and key projections in the adapted layers preserves the pretrained behavior at initialization and allows the grounding and propagation heads to be removed at inference when only generation is needed.

Both heads follow the upsampling strategy and the time causality used in a 3D VAE (Yang et al., 2024) while remaining lightweight. A standard 3D VAE temporally compresses several frames into one latent which places attention on a shorter temporal lattice than the ground truth instance mask tracks. In CogVideoX, the VAE temporally compresses frames from $1 + 4F$ to $1 + F$, which reduces the effective frame rate of the latent sequence by a factor of $4$. This places attention on a shorter temporal lattice than the ground truth binary mask tracks. In our setup, the latent attention sequence spans 13 steps, whereas the ground-truth instance mask tracks span 49 frames. The most straightforward solution to address this gap is to compress supervision by taking an element-wise OR over every 4 consecutive frames so that each group maps to one latent step. However, this ignores temporal ordering and inflates foreground regions which weakens alignment under motion and degrades identity precision. Instead, we upsample the attention to the mask frame rate. The lightweight decoder mirrors the VAE temporal up path and causally expands the 13 step attention sequence to 49 frames without using future frames. Supervision is then applied at the original frame rate against the binary instance mask tracks. This preserves temporal ordering and sharp instance boundaries, avoids foreground inflation, and leaves the generator unchanged while confining updates to the query and key projections in the adapted layers.

### D.2 IMPLEMENTATION DETAILS

We use the CogVideoX-5B-I2V (Yang et al., 2024) as our base image-to-video diffusion model, and generate output videos at a resolution of $480 \times 720$ with a total of $49$ frames. The trainable parameters are limited to the selected LoRA (Hu et al., 2021) layers (*layer 7, 11, 12*), the input projection layer and lightweight decoder heads for grounding and propagation heads. For model finetuning, we adopt LoRA (Hu et al., 2021) with a rank of $128$ and $\alpha = 64$. We optimized only the selected LoRA layers, input projection layer and lightweight decoders while keeping the other parts of the model frozen. Training was conducted on our curate dataset, MATRIX-11K, using an AdamW (Loshchilov & Hutter, 2019) optimizer with a cosine learning rate decay schedule. The model is trained for $4,000$ steps, which takes approximately 32 hours on a single NVIDIA A6000 GPU.

We apply Semantic Grounding Alignment (SGA) loss and Semantic Propagation Alignment (SPA) loss selectively. The SGA loss supervises video-to-text attention in blocks 7 and 11. The SPA loss supervises video-to-video attention in block 12. This selective strategy concentrates updates on the query and key projections of the adapted layers, stabilizes optimization under motion and preserves the pretrained generator at initialization and at inference.

## E TRAINING-FREE CROSS-MODAL GUIDANCE DETAILS

Our analysis shows that semantic grounding (video-to-text) and semantic propagation (video-to-video) are concentrated in a small subset of *interaction-dominant layers*. To validate whether these layers provide effective handles for improving interaction fidelity, we design a zero-shot guidance strategy applied only at the identified layers. Specifically, we introduce Cross-Modal Guidance (CMG), our novel approach for enhancing grounding, and adopt Cross-Attention Guidance (CAG) (Nam et al., 2025) for propagation. CMG perturbs token-to-entity attention maps at dominant video-to-text layers to simulate degraded grounding and then guides the model away from these perturbed predictions, reinforcing semantic alignment. In parallel, CAG applies the same perturb-and-guide principle to cross-frame attention, reinforcing temporal consistency without additional training. Fig. 18 shows the architectural details of CAG and CMG.

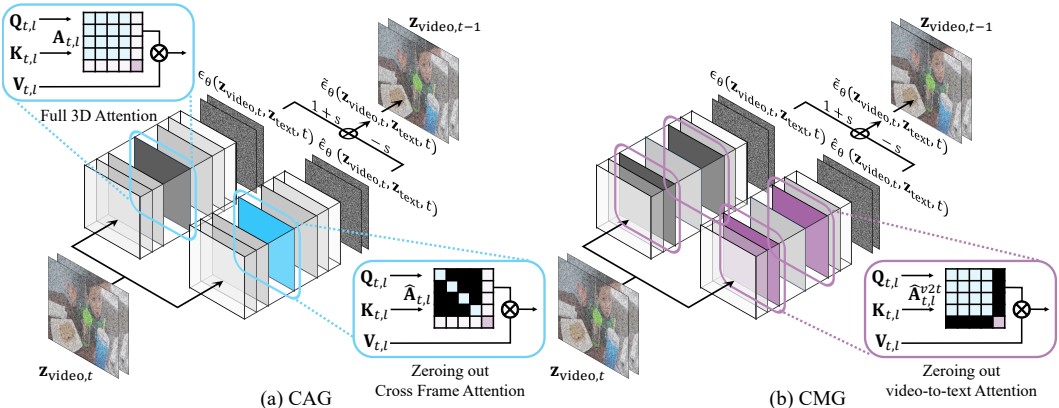

Figure 18: **Guidance Details.**

## E.1 ARCHITECTURAL DETAILS

**Cross-Attention Guidance (CAG).** Inspired by PAG (Ahn et al., 2025), which enhances image fidelity by transforming selected self-attention maps into identity matrices, we extend this idea to the video DiT architecture. In PAG, identity matrices are created by multiplying a diagonal mask into the attention map before the softmax operation, setting diagonal elements to 0 and off-diagonal to $-\infty$, which yields an identity matrix after softmax. A naive extension to video assigns $-\infty$ to cross-frame positions, but this undesirably suppresses self-frame and text-frame scales. To address this, DiffTrack (Nam et al., 2025) zero out only the cross-frame values after softmax in $\mathbf{A}_{t,l}^{\text{v2v}}$, producing modified maps $\hat{\mathbf{A}}_{t,l}^{\text{v2v}}$ that preserve other interactions.

**Cross-Modal Guidance (CMG).** Analogous to CAG, CMG applies the perturb-and-guide strategy to video-to-text attention. At interaction-dominant layers, we simulate degraded grounding by zeroing out token–instance alignments after softmax. For noun tokens, attention weights to instance regions are suppressed; for verb tokens, attentions capturing subject–object unions are removed. This produces modified maps $\hat{\mathbf{A}}_{t,l}^{\text{v2t}}$ where semantic grounding is intentionally weakened, while other attentions remain intact. The diffusion model is then guided away from these degraded predictions, reinforcing correct grounding without retraining or auxiliary conditions.

Both can be formulated as:

$$\tilde{\epsilon}_\theta(\mathbf{z}_{\text{video},t}, \mathbf{z}_{\text{text}}, t) = \epsilon_\theta(\mathbf{z}_{\text{video},t}, \mathbf{z}_{\text{text}}, t) + s \cdot (\epsilon_\theta(\mathbf{z}_{\text{video},t}, \mathbf{z}_{\text{text}}, t) - \hat{\epsilon}_\theta(\mathbf{z}_{\text{video},t}, \mathbf{z}_{\text{text}}, t)),$$

where $\epsilon_\theta(\cdot)$ is the noise prediction from a standard pass at timestep $t$, conditioned on the text, and $\hat{\epsilon}_\theta(\cdot)$ indicates the noise prediction from a perturbed forward pass. $s$ is the perturbation guidance scale and the final prediction $\tilde{\epsilon}_\theta(\cdot)$ is guided away from the degraded predictions.

## E.2 IMPLEMENTATION DETAILS

For CAG, we adopt the 1 interaction-dominant video-to-video layers (*e.g., layer 12* in CogVideoX-5B-I2V) identified by our analysis, and apply guidance across all sampling steps.

For CMG, we similarly select the 2 interaction-dominant video-to-text layers (*e.g., layer 7 and 11* in CogVideoX-5B-I2V) and apply zero-shot guidance at every timestep. Both guidance scales are set following PAG (Ahn et al., 2025), and no additional parameters or training are introduced.

## E.3 EXPERIMENTAL RESULTS

In Fig. 19, we diagnose the failures through attention. In the first row of (a), for "woman cuts cake", noun attention for *woman* leaks onto the man and the verb *cut* focuses on him rather than the union of the *woman-cake* region, so the action is assigned to the wrong agent. In the second row of (a), noun attention to the subject *man* is weak and diffuse and video-to-video attention does not carry a

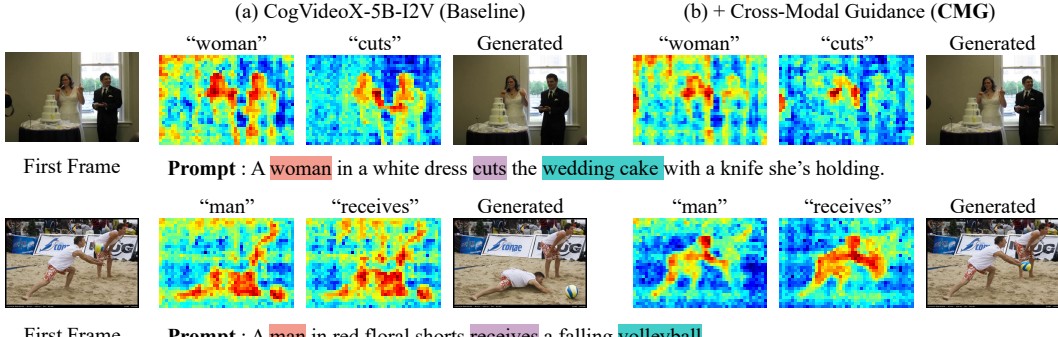

Figure 19: **Cross-Modal Guidance Visualization.**

Table 3: **Comparison of evaluation protocols.** Existing benchmarks assess quality, compositionality, or physics, but only InterGenEval targets *interaction-level semantic alignment*.

| Protocol | Target | Semantic Granularity | Temporal Semantics | Semantic Alignment |
|---|---|---|---|---|
| VBench | Visual Quality | Global (frame/clip) | × | Global appearance |
| VBench-2.0 | Faithfulness | Global / Semantic | ✓ | Human, controllability, physics |
| EvalCrafter | Quality & Alignment | Global (entity cues) | ✓ | Basic visual-text alignment |
| FETV | Attributes | Entity (attributes) | × | Attribute-level alignment |
| T2V-CompBench | Compositionality | Relation (multi-object) | Partial | Multi-object relations |
| PhyGenBench | Physics | Event (physics) | ✓ | Physical plausibility |
| PhyWorldBench | Physics | Event (physics) | ✓ | Physical plausibility |
| **InterGenEval (ours)** | **Interaction Fidelity** | **Interaction-level** | ✓ | **Interaction-level alignment** |

stable subject track forward, so the motion does not start. These cases show that when grounding is weak, propagation also breaks.

We then apply perturbation guidance only to the interaction-dominant layers identified by our analysis and leave all other layers unchanged. The guidance biases video-to-text attention $\mathbf{A}^{v2t}$ toward the intended subject, object and their union and stabilizes the carry-over in video-to-video attention with a small weight to avoid appearance drift. Under this setting, many borderline cases flip from failure to success. In the first row of (b), this sharpening results in the woman executing the cut with contact maintained across frames and in the second row of (b), the man is cleanly localized from the first frame and the motion initiates and proceeds without drift. The fact that a lightweight in-layer perturbation cleans up video-to-text and video-to-video attention and improves plausibility, frequently turning failures into successes, shows that these layers are the dominant handles for attention sharpening as well as for grounding and propagation.

However,the critical limitations remain. CMG is zero-shot guidance that amplifies existing attentions at selected layers, but it does not inject region-level or ID-level supervision. When the initial noun map is severely ambiguous, when the verb is not well approximated by the subject-object union, or under heavy occlusion, sharpening may be insufficient or may over-concentrate attention and subtly degrade appearance. Moreover, increasing the guidance scale to compensate often saturates attention and collapse diversity. Therefore, these observations motivate our mask-track alignment losses that provide explicit grounding and propagation signals, as depicted in the Sec. 5 in the main paper.

# F  EVALUATION PROTOCOL DETAILS

## F.1  RELATED WORKS

Early evaluations of video generation primarily relied on Inception Score (IS) (Salimans et al., 2016), Fréchet Inception Distance (FID) (Heusel et al., 2017), and Fréchet Video Distance (FVD) (Unterthiner et al., 2018), which measure distributional fidelity and diversity but fail to capture semantic correctness. To address this, recent benchmarks introduced multi-dimensional protocols. VBench (Huang et al., 2024) decomposes generation quality into 16 dimensions, including frame

| First frame | MATRIX(ours) | CogVideoX-5B-I2V | | First frame | MATRIX(ours) 30th frame | MATRIX(ours) 40th frame |
|---|---|---|---|---|---|---|

| CLIPScore | 24.39 | **29.21** |
| BLIP-BLEU score | 0.019 | **0.0599** |

**Prompt** : The girl pushes another girl on a tire swing.

(a) Score misalignment between models

| CLIPScore | 30.94 | **32.55** |
| BLIP-BLEU score | 0.0001 | **0.0016** |

**Prompt** : Woman in a black shirt lifts the red book in a library

(b) Score misalignment between frames

Figure 20: **Limitations of Existing Semantic Alignment Metrics using BLEU and CLIP.** (a) Cross-model comparison: despite clear human preference for one model, BLEU and CLIP favor the other, assigning high scores to implausible or semantically misaligned results. (b) Within-model frames: frames preferred by humans receive lower scores than other frames from the same clip, showing insensitivity to instance-level grounding and temporal consistency. This gap motivates InterGenEval, an interaction-aware evaluation.

aesthetics, temporal consistency, and prompt adherence, and validates alignment with human judgments. EvalCrafter (Liu et al., 2023a) further integrates a large prompt suite and combines multiple automatic metrics with human preference weighting. FETV (Liu et al., 2023b) emphasizes attribute-level evaluation, scoring static and temporal quality as well as fine-grained alignment. These works broaden coverage beyond single-number scores but remain global or attribute-focused. Many of these benchmarks rely heavily on CLIP-based models to assess semantic similarity. However, as shown in Fig. 20, CLIP score and the BLIP-BLEU score from EvalCrafter fail to capture interaction level granularity, highlighting their limitations in evaluating the interaction modeling capabilities of generated videos.

Other benchmarks target narrower capabilities. T2V-CompBench (Sun et al., 2024) measures compositionality over relations, attributes, and actions through VLM-based and detection-based metrics. PhyGenBench (Meng et al., 2024) and PhyWorldBench (Gu et al., 2025a) evaluate physical commonsense and causal plausibility with structured protocols, while VBench-2.0 (Zheng et al., 2025b) expands toward "intrinsic faithfulness", covering human fidelity, controllability, and physics. These efforts highlight compositional and physical reasoning, but still do not capture whether models realize prompt-specified interactions.

In particular, as compared in Tab. 3, existing protocols assess global semantics or object attributes but lack *interaction aware semantic alignment*: whether the correct subject acts on the correct object, contact occurs, and causal unfolding matches the prompt. Our proposed **InterGenEval** addresses this gap by treating interactions as the evaluation unit and introducing grounded criteria for role- and time-sensitive alignment.

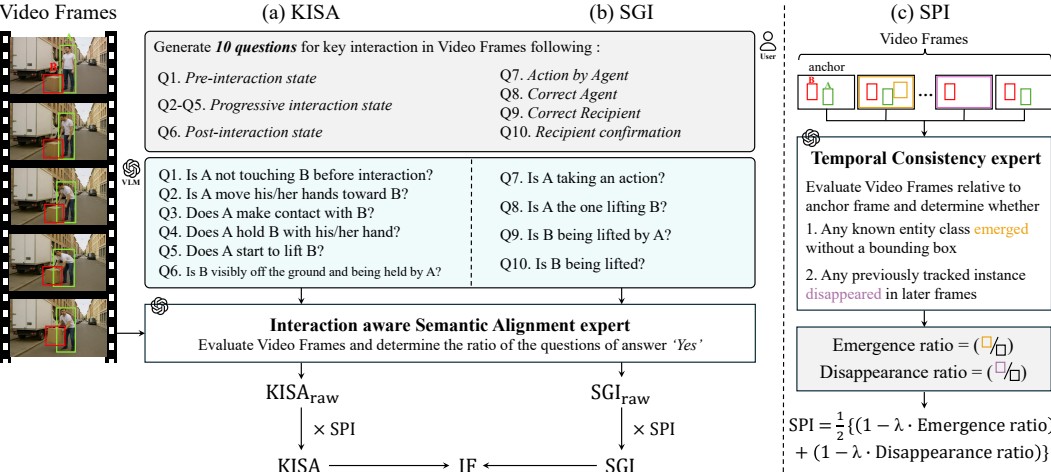

Figure 21: **Evaluation Protocol Pipeline.**

## F.2 OVERVIEW

InterGenEval focuses on *interaction aware semantic alignment* between the video and the prompt, measured by two metrics: Key Interaction Semantic Alignment (KISA) and Semantic Grounding Integrity (SGI). Specifically, after extracting key interactions from the prompt, KISA verifies step-by-step whether the subject actually performs the specified action on the object, while SGI assesses the grounding accuracy of the subject and object. When multiple key interactions are present in a prompt, each key interaction is evaluated to obtain its corresponding KISA and SGI, which are then averaged across all interactions to produce the final KISA and SGI. This enables evaluation in multi-interaction and multi-instance scenarios. Fig. 21 provides the details of the evaulation protocol pipeline.

Meanwhile, maintaining temporal consistency of interaction and grounding is also crucial. To account for this, we introduce Semantic Propagation Integrity(SPI) as a sub-metric. SPI captures whether any instance suddenly appears or disappears throughout the video, providing a measure of temporal consistency. SPI is then applied to KISA and SGI, injecting temporal consistency into both metrics by penalizing inconsistent instance propagation overtime. The mean of KISA and SGI is then defined as the final Interaction Fidelity (IF) score. For clarity, we denote the unadjusted scores as $\text{KISA}_{\text{raw}}$ and $\text{SGI}_{\text{raw}}$, and SPI applied scores as KISA and SGI.

**Setup**. InterGenEval leverages multimodal foundation model GPT-5 (OpenAI, 2025), utilizing its strong visual understanding and reasoning capabilities throughout the evaluation process. KISA and SGI are computed through a question-answering framework, which verifies whether the subject actually performs the intended action and whether the subject and object are correctly grounded. Additionally, SPI is derived through an instruction-based evaluation procedure, where GPT-5 is used to detect the emergence and disappearnace of each instance across frames assessing temporal consistency.

InterGenEval uses a sequence of frames where each instance involved in the interaction is visually annotated with a bounding box in a distinct color. We generate these annotated frames using SAM2 (Ravi et al., 2024), which allows us to extract precise bounding boxes for each instance. This visual representation enables GPT-5 to clearly identify each instance and focus on fine-grained interaction details. Each evaluation frame sequence includes the first and last frames of a video, while intermediate frames are uniformly sampled using a fixed stride. In this paper, the stride is set to 5.

## F.3 EVALUATION METRICS

**Question Generation.** Since KISA and SGI are derived from a question-answering framework, it is important to construct a well-structured set of questions that reflects whether each step of the interaction is performed and whether all instances are correctly grounded. To this end, we use GPT-5 to automatically generate 10 yes/no questions per key interaction, guided by a task-specific instruction. As input, GPT-5 receives the text prompt, a list of instances, their corresponding appearance descriptions, and assigned bounding box colors. Based on this input, GPT-5 first identifies key interactions described in the prompt. Then for each key interaction, it generates 10 questions that are aligned with the evaluation goals of KISA and SGI. Each question explicitly refers to instances using both appearance and bounding box color(*e.g., woman in a green jacket (red bbox)*). The first six questions (Q1-Q6) are used to compute $\text{KISA}_{\text{raw}}$, as they assess whether the interaction progresses through its expected stages. The remaining four questions (Q7-Q10) focus on verifying instance grounding and are used to compute $\text{SGI}_{\text{raw}}$. Further details on the structure of these questions and the computation of KISA and SGI are provided in the following section.

**Key Interaction Semantic Alignment (KISA).** KISA evaluates an interaction by decomposing it into three temporal stages: pre, during, and post interaction. Question 1 corresponds to the pre-interaction stage and checks whether the subject and object are in the expected initial state prior to any engagement. Question 2 through 5 cover the during-interaction stage, where the model verifies the progression of the action across multiple steps. Finally, Question 6 focuses on the post-interaction stage, assessing whether the expected outcome of the interaction has been visibly achieved. For example, in the case of the interaction "A lifts B", the six questions will be constructed as follows. *Q1. Is A not touching B before interaction? , Q2. Is A move his/her hands toward B?, Q3. Does A make contact with B? Q4. Does A hold B with his/her hand? Q5. Does A start to lift B?*

*Q6. Is B visibly off the ground and being held by A?* $\text{KISA}_{\text{raw}}$ is then computed as the proportion of "Yes" responses among these six questions, indicating how successfully the interaction is executed across all expected stages.

**Semantic Grounding Integrity (SGI).**    SGI evaluates whether the subject and object are correctly grounded within the interaction. To this end, it comprises four questions. Question 7 verifies whether the subject is correctly identified as the actor of the interaction. Question 8 checks whether the subject performs the specified action on the intended object. Question 9 evaluates whether the object is being acted upon by the specified subject. Question 10 assesses whether the object is indeed the correct recipient of the action. For example, in the case of the interaction "A lifts B", the four grounding questions will be constructed as follows. *Q7. Is A taking an action? Q8. Is A the one lifting B? Q9. Is B being lifted by A? Q10. Is B being lifted?* $\text{SGI}_{\text{raw}}$ is then computed as the proportion of "Yes" responses among these four questions, capturing the accuracy of instance level semantic grounding within the interaction.

**Semantic Propagation Integrity (SPI).**    SPI measures the temporal consistency of each instance throughout the video. The first frame is used as an anchor, and the remaining frames are compared against it to detect any changes. We provide GPT-5 with a list of instances, their bounding box colors, and the bounding box visualized frame sequence as input. GPT-5 then outputs the detection results for emergence and disappearance for each frame. Specifically, emergence is defined as the appearance of a new instance that does not appear in the anchor frame but emerges in later frames. Disappearance occurs when an instance annotated with a bounding box in the anchor frame is no longer visible in subsequent frames. To compute the SPI score, we first calculate the ratio of frames in which emergence or disappearance is detected. Each of these ratio is multiplied by a penalty weight $\lambda$, and the result is subtracted from 1 to obtain the emergence score and disappearance score, respectively. The final SPI score is defined as the average of these two scores. In this paper, we set $\lambda = 5$.

**Overall Scoring.**    As previously mentioned, we use SPI to incorporate temporal consistency into KISA and SGI. SPI ranges within (-4,1], with higher values indicating better temporal consistency of instances. To strongly penalize videos with poor temporal consistency, we multiply SPI with both $\text{KISA}_{\text{raw}}$ and $\text{SGI}_{\text{raw}}$ to obtain their reweighted final values.

$$\text{KISA} = \text{KISA}_{\text{raw}} \times \text{SPI}, \text{SGI} = \text{SGI}_{\text{raw}} \times \text{SPI}.$$

The final Interaction Fidelity (IF) score is then computed as the average of the reweighted KISA and SGI.

$$\text{IF} = \frac{\text{KISA} + \text{SGI}}{2}.$$

IF combines KISA, SGI, and SPI to provide a quantitative score that reflects interaction aware semantic alignment with temporal consistency. This formulation offers an interpretable and consistent metric for assessing interaction quality. As a result, InterGenEval functions as a practical evaluation framework that gives precise feedback on the quality of interaction-aware video generation.

# G  EVALUATION

## G.1  COMPARISON MODELS

We compare our approach against several recent open-source image-to-video diffusion models including CogVideoX-2B-I2V (Yang et al., 2024), CogVideoX-5B-I2V (Yang et al., 2024), TaVid (Kim & Joo, 2025)and Open-Sora (Zheng et al., 2024). CogVideoX-2B-I2V is a lightweight version with approximately 2 billion parameters, designed for efficient video synthesis. In contrast, CogVideoX-5B-I2V scales to 5B parameters and offers stronger generative capacity through larger model size and broader training coverage. Finally, we include Open-Sora (11B) as a fully open-source alternative, widely adopted as a community benchmark. Collectively, these comparison models span a spectrum of scales, training regimes, and accessibility levels, enabling us to evaluate both the absolute quality of our method and its relative efficiency against existing models.

## G.2 ADDITIONAL METRICS

In addition to our proposed protocol, we adopt several metrics from VBench (Huang et al., 2024) and VBench-2.0 (Zheng et al., 2025b) to provide a broader evaluation of video quality. VBench decomposes video quality into temporal and frame-wise aspects.

For temporal quality, *Subject Consistency* measures whether the main subject maintains a stable appearance across frames, computed via DINO (Caron et al., 2021) feature similarity. *Background Consistency* evaluates the stability of the background using CLIP (Radford et al., 2021) feature similarity. *Motion Smoothness* quantifies whether motion is physically plausible and continuous, using motion priors derived from a video interpolation model (Li et al., 2023). *Dynamic Degree* measures the amplitude of motion in the generated video, estimated with RAFT (Teed & Deng, 2020)-based optical flow.

For frame-wise quality, *Aesthetic Quality* captures perceptual attractiveness such as composition and color harmony, evaluated with the LAION aesthetic predictor (Beaumont & Schuhmann, 2022). *Imaging Quality* assesses low-level fidelity by detecting distortions such as blur, noise, or over-exposure using MUSIQ (Ke et al., 2021) trained on the SPAQ (Fang et al., 2020) dataset.

From VBench-2.0, we additionally include *Human Anatomy*, which evaluates whether human instances are consistently maintained without abnormal merging, splitting, or deformation across frames. This is achieved by detecting humans, hands, and faces with YOLO-World (Cheng et al., 2024), and applying anomaly detectors trained on a large-scale dataset of real and generated human samples. The final score is defined as the proportion of frames not flagged as abnormal.

## G.3 EVALUATION DATASET

Fig. 37 illustrates the benchmark we used for evaluated, consisting of 118 image-prompt pairs. These pairs were constructed by selecting images with varying number of instance IDs (2, 3 or 4), and by categorizing motions from simple to complex based on levels of contact and dynamism, such as "walking along the street" (low contact and low dynamism) or "hands over the cup" (hight contact and hight dynamism). Each prompt was designed to include (1) main subjects and objects involved in the interaction, (2) the interaction or motion descriptions between the main subjects and objects, and (3) a scene description specifying the appearance of the main instances. For all images, we used a large language model (LLM) to generate prompts that satisfy these conditions, following the same guidelines used during our dataset curation process in Sec. 3 of the main paper and analysis evaluation dataset curation process in Sec B of Appendix.

## G.4 ADDITIONAL ANALYSIS

To further validate the reliability of our evaluation, we visualize the full per-clip distributions of the normalized KISA, SGA, and IF scores for all models on our interaction-aware evaluation set, as presented in Fig. 22. Each violin is computed from all clips in the benchmark, and summarizes the distribution of scores for a given model-metric pair.

We observe that the distributions for MATRIX are consistently skewed toward higher scores with less mass in the low-score region across all three metrics, while baseline and other comparison models exhibit longer tails toward low values. This indicates that our model not only improves the mean performance reported in the main paper but also reduces variance and the frequency of severe failures. In other words, MATRIX achieves more reliable interaction-aware generation, leading to produce high-quality, interaction-consistent videos, rather than relying on a few outlier successes.

## G.5 HUMAN EVALUATION

**Human evaluation details.** We adopt a Two-Alternative Forced Choice (2AFC) protocol (Blattmann et al., 2023; Chefer et al., 2025), where raters compare two videos side-by-side and select the better one. Two models are uniformly sampled from {CogVideoX-5B-I2V (Yang et al., 2024), Open-Sora-I2V (Zheng et al., 2024), TaVid (Kim & Joo, 2025), Ours}, yielding all six model pairs. For each sampled pair, we randomly select a text–image prompt from the InterGenEval

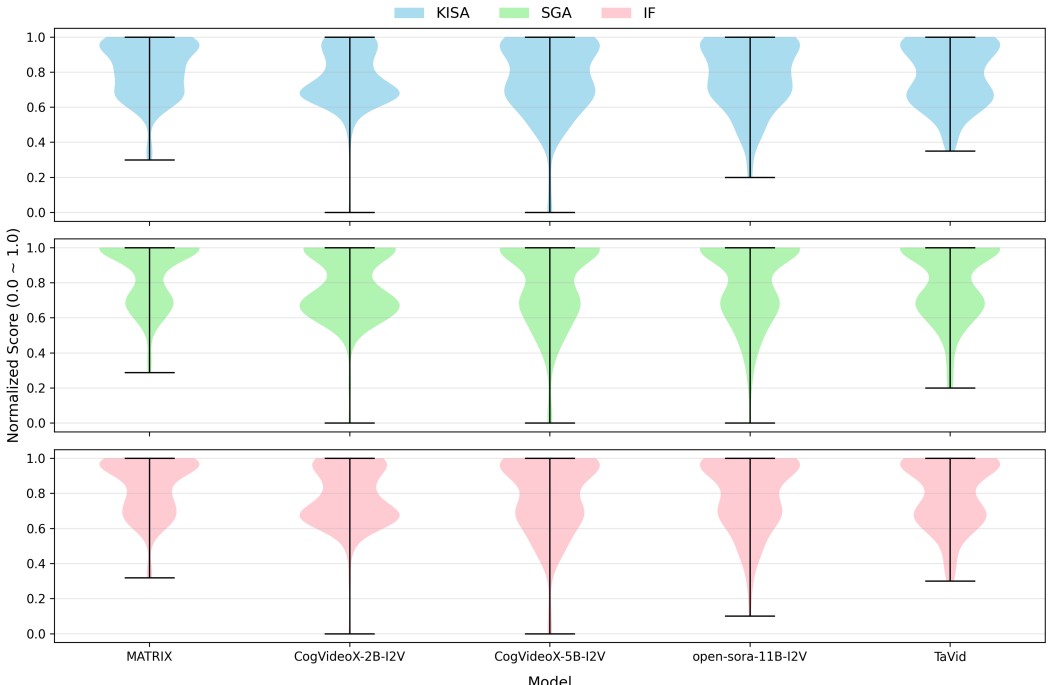

Figure 22: **Per-Clip Distributions of Interaction-aware Metrics.** Violin plots of normalized KISA, SGA, and IF scores for MATRIX and four comparison models( Yang et al. (2024), Zheng et al. (2024), Kim & Joo (2025)). Each violin summarizes the distribution of per-clip scores over all evaluation videos, and the black vertical line indicates the overall range observed for that model-metric pair.

evaluation set and generate one video per model using the same prompt. The left/right presentation order is randomized to avoid positional bias, and raters are not allowed to skip or assign ties.

Each trial consists of five evaluation questions: (1) *Interaction Accuracy* – correctness of the specified interaction (who interacts with whom and what they are doing); (2) *Semantic Grounding* – inclusion of objects indicated in the image prompt as instructed by the text prompt; (3) *Semantic Propagation* – temporal consistency and absence of hallucinated objects; (4) *Semantic Alignment* – overall fidelity and naturalness of the interaction; (5) *Overall Quality* – perceptual realism and visual plausibility.

Each participant evaluated all six model pairs with two prompts per pair, resulting in $6 \times 2 = 12$ video comparisons (12 pairs) per participant. This design ensured equal comparison frequency across models, providing a balanced and fair evaluation protocol.

**Participants.** We recruited 31 participants, each responding to multiple trials to cover all pairwise comparisons under diverse prompts. Results are aggregated using the *win rate* across pairwise comparisons and criteria, following standard practice in perceptual evaluation. Fig. 39 illustrates the 2AFC setup.

**Human evaluation results.** Fig. 23 summarizes the win rates. Our model (MATRIX) consistently exceeds 0.9 across all criteria, while its backbone CogVideoX-5B-I2V remains around 0.36–0.44. This demonstrates substantial improvements in *interaction-aware*

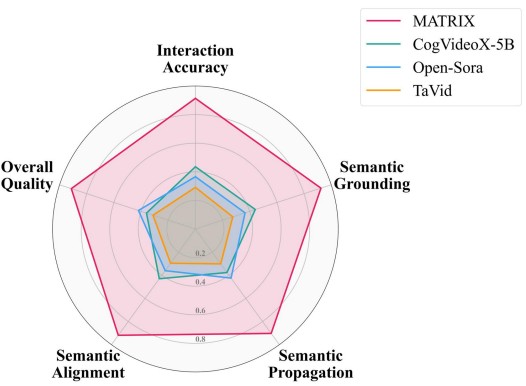

Figure 23: **Human Evaluation Results.**

Table 4: **Additional Quantitative Comparison.**

| Methods | Human Fidelity | Video Quality | | | | | |
| --- | --- | --- | --- | --- | --- | --- | --- |
| | HA (↑) | SC (↑) | BC (↑) | MS (↑) | DD (↑) | AQ (↑) | IQ (↑) |
| CogVideoX-2B-I2V (Yang et al., 2024) | 0.937 | **0.969** | **0.962** | 0.993 | 0.152 | **0.602** | 69.69 |
| CogVideoX-5B-I2V (Yang et al., 2024) | 0.938 | 0.946 | 0.942 | 0.986 | 0.556 | 0.582 | 69.66 |
| Open-Sora-I2V (Zheng et al., 2024) | 0.893 | 0.926 | 0.937 | 0.992 | **0.762** | 0.495 | 63.32 |
| TaVid (Kim & Joo, 2025) | 0.919 | 0.942 | 0.939 | 0.991 | 0.727 | 0.568 | 68.90 |
| **MATRIX (Ours)** | **0.954** | 0.962 | 0.956 | **0.994** | 0.492 | 0.587 | **69.73** |

*semantic alignment*, covering interaction accuracy, grounding, propagation, and alignment as well as perceptual quality. Other baselines, such as Open-Sora and TaVid, show even lower performance. Overall, MATRIX not only inherits the strengths of CogVideoX but also delivers robust interaction fidelity and perceptual realism, validating the core contribution of our approach.

# H  ADDITIONAL RESULTS

## H.1  ADDITIONAL QUALITATIVE RESULTS

Fig. 40 and Fig. 41 present the additional qualitative results comparing our method with others, including CogVideoX-5B-I2V (Yang et al., 2024), Open-Sora-I2V (Zheng et al., 2024) and TaVid (Kim & Joo, 2025). To further verify that our improvements are not limited to human-object contact, Fig. 24 reports results on a dedicated evaluation subset consisting solely on non-human and non-contact interaction scenarios.

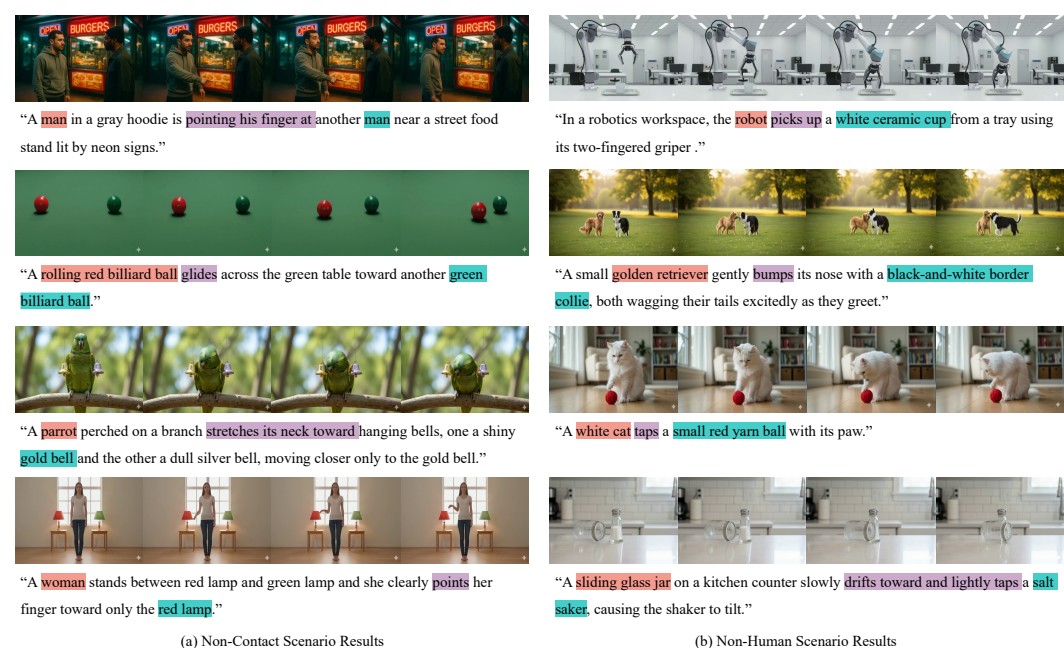

(a) Non-Contact Scenario Results          (b) Non-Human Scenario Results

Figure 24: **Generalization to non-contact and non-human interactions.**

## H.2  ADDITIONAL QUANTITATIVE RESULTS AND ANALYSIS

Tab. 4 reports additional quantitative comparisons across CogVideoX-2B-I2V (Yang et al., 2024), CogVideoX-5B-I2V (Yang et al., 2024), Open-Sora-I2V (Zheng et al., 2024), TaVid (Kim & Joo, 2025) and MATRIX (Ours), across the standard metrics of VBench (Huang et al., 2024).

SC (Subject Consistency) and BC (Background Consistency) are highest for CogVideoX-2B-I2V, but this stems from its near-static outputs rather than stronger modeling. As shown in Fig. 25, little

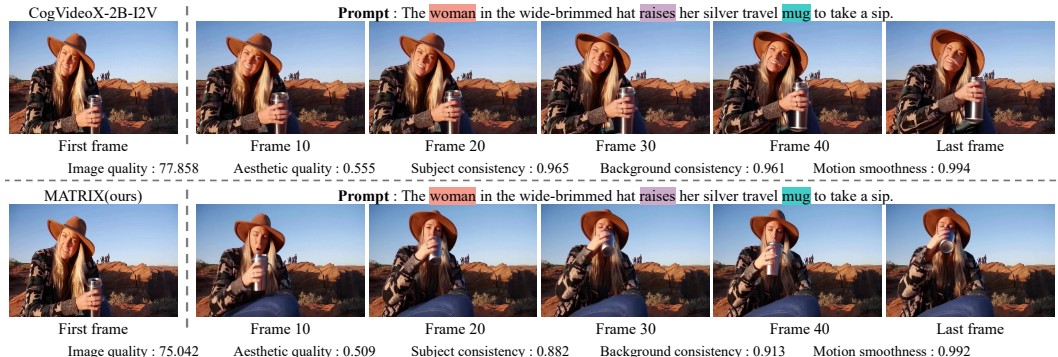

Figure 25: **Limitations of VBench metrics.**

Table 5: **Training Cost and Efficiency Comparison.**

| | Methods | Time/epoch | InterGenEval KISA (↑) | SGI (↑) | IF (↑) | Human Fidelity HA (↑) | Video Quality MS (↑) | IQ (↑) |
|---|---|---|---|---|---|---|---|---|
| (I) | Baseline (Yang et al., 2024) | - | 0.406 | 0.491 | 0.449 | 0.936 | 0.987 | 69.66 |
| (II) | LoRA w/o layer selection | 24.43 hr | 0.445 (+9.6%) | 0.526 (+7.1%) | 0.486 (+8.2%) | 0.940 (+0.4%) | 0.994 (+0.7%) | 69.77 (+1.6%) |
| (III) | **Ours** | 31.09 hr | **0.546 (+34.5%)** | **0.641 (+30.5%)** | **0.593 (+32.1%)** | **0.954 (+1.9%)** | 0.994 (+0.7%) | 69.73 (+1.0%) |

or no motion inflates inter-frame consistency and keeps AQ and IQ high because there is minimal motion-induced degradation. The motion metric, Dynamic Degree (DD), confirms this with very low values. Very high DD is not always desirable either, since excessive motion increases off-track drift and hallucination risk. In Fig. 25, when comparable motion is introduced, SC, BC and AQ drop sharply, while human raters still prefer results that express the intended motion with correct bindings. Thus, SC, BC, AQ and IQ do not reliably track human preference in these settings. These metrics should be integrated alongside motion-aware and interaction-aware measures such as DD, KISA, SGI and IF. Our method maintains moderate DD and higher interaction fidelity, which aligns better with human judgments.

## H.3 ADDITIONAL ABLATION RESULTS

We compare the training cost of different variants under a near-iso-FLOPs setting. All fine-tuning experiments are conducted on a single A6000 GPU at a resolution of $480 \times 720$ with the same batch size, so the wall-clock time per epoch serves as a practical proxy for the total FLOPs used. We consider three variants, (I) the naive CogVideoX-5B-I2V model, which is the original pretrained model without any additional finetuning, (II) a LoRA-only baseline, which we term "LoRA w/o layer selection", and (III) our full method. All three variants are evaluated under the same configuration, including resolution, number of sampling steps and guidance setting. For variants (II) and (III), Tab. 5 reports the per-epoch training time. Specifically, the LoRA-only baseline requires about 24.43 hours per epoch, whereas our full method requires 31.09 hours per epoch, corresponding to an additional 6.7 hours of training time. Despite this moderate overhead, the improvements on interaction-centric metrics, including KISA, SGI and IF, are substantially larger. Compared to the naive CogVideoX-5B-I2V model, our method improves KISA from $0.406$ to $0.546$ (+34.5%), SGI from $0.491$ to $0.641$ (+30.5%), and IF from $0.449$ to $0.593$ (+32.1%). Even when compared directly to the LoRA-only baseline, our method still yields relative gains of +22.7% (KISA), +21.9% (SGI), and +22.0% (IF), while holistic appearance metrics (HA/MS/IQ) remain essentially unchanged. This efficiency is particularly notable when contrasted with recent video generation approaches (Chefer et al., 2025) that jointly train a video backbone from scratch, which typically incur substantially higher training cost. In our case, a lightweight LoRA-based fine-tuning recipe on top of CogVideoX-5B-I2V yields 30–35% relative improvements on core interaction metrics at essentially identical inference cost and only a modest increase in training time, resulting in a favorable trade-off.

## I LIMITATIONS

**Instance Scalability.** Our current framework supports up to five instance mask tracks per scene. While this upper bound appear restrictive, it is well aligned with the distribution observed in our

dataset Appendix A. As illustrated in Fig. 12 in Appendix, scenes containing more than five interacting instances are rare, and most examples contain two to four distinct objects involved in interaction. This design choice thus reflects a practical tradeoff between generality and simplicity, allowing the model to remain effective without introducing unnecessary architectural complexity. Nevertheless, extending support to a larger number of instances remains a feasible direction for future work.

**Small Mask Sensitivity.** Another limitation arises when the instance mask occupies a very small spatial region. In such cases, the visual grounding signal may become weak or even ambiguous, potentially degrading the model's ability to generate accurate motion. Future improvements could involve strategies such as hierarchical mask encoding, spatially adaptive attention or resolution-aware learning to enhance robustness against object size variations. We leave these directions for future exploration.

## J   Discussion and Future Work

Although we focus on instance mask tracks as the core supervision and analysis modality, our formulation is modality-agnostic on the analysis side, and naturally extends to multi-modal settings. Additional cues such as optical flow and depth are highly complementary. Optical flow can provide stronger supervision for fine-grained motion continuity and temporal consistency, while depth can help disambiguate occlusions and 3D proximity in contact-rich interactions. In fact, the same semantic grounding and propagation framework can incorporate these signals by adding flow- or depth-based consistency terms, or by conditioning the video DiT on flow or depth features during training. A systematic integration of such multi-modal cues (*e.g., mask tracks + flow + depth*) is therefore a promising direction for future work.

Beyond flow and depth, point tracking can provide another complementary modality. Mask tracks offer instance-level grouping and are well suited for supervising semantic grounding and propagation at the level of each instance ID, but they inherently provide relatively coarse supervision for fine-grained local motion (*i.e.,* exactly where each local detail moves in the next frame). Point tracks are almost the opposite: they capture very fine local motion but do not induce clear instance-level grouping or role semantics (*e.g., subject vs. object*). This suggests a natural trade-off and complementarity. A compelling extension is to use mask tracks to supervise instance-level roles and interaction structure (subject–object–verb), while using point tracks to refine fine motion and local detail propagation across frames. We leave such multi-modal extensions as valuable future directions for interaction-aware video generation.

## K   The Use of Large Language Models (LLMs)

In accordance with the ICLR 2026 submission policy, we disclose that Large Language Models were used to assist in grammar correction, polishing of the writing in this paper and caption processing in our dataset curation pipeline.

# First turn
Given the following video caption, determine whether there are any active and meaningful interactions involving a living subject and another distinct entity (another person, object, or animal).
Video caption: *[caption]*
Valid interactions must involve:
    - A living subject
    - A separate target entity (another person, an object)
    - A clear relationship or action connecting them
Do NOT count:
    - Self-directed actions (e.g., 'a man gesturing', 'a person walking', 'someone raising their hand')
    - Vague verbs with no target (e.g., 'a woman moves', 'a person acts')
    - Emotional or internal states with no external relation (e.g., 'a boy thinking', 'a girl smiling')
Only count interactions that involve:
    - Clear action verbs between two entities (e.g., 'hugging', 'pointing at', 'talking to', 'giving something')
Respond with exactly one of the following:
    - null → if no such interaction exists
    - an integer (e.g., 1, 2, 3, 4, ...) representing the exact count of interactions

# Second turn
You are an AI that extracts valid and meaningful interactions from a video caption.
Video caption: *[caption]*
Follow these rules:
    - First, identify all unique, living subjects and distinct entities mentioned in the caption. Assign a consistent ID (<id1>, <id2>, etc.) to each unique entity. A single entity must have only one ID, even if it is part of multiple interactions.
    - If the caption describes multiple entities of the same type (e.g., 'two men'), you must use descriptive details (like 'on the left', 'on the right', or clothing) to assign them distinct IDs. Do not use a single ID for multiple individuals.
    - Extract all active and meaningful interactions described in the caption. Do not omit any valid interactions, even if they seem less dynamic than others.
    - A valid interaction must meet all of the following conditions: a living subject (src1), a separate target (tgt1 ≠ src1), and a clear action verb. Valid examples include both highly active actions like '<id1> gives <id2>' and less dynamic actions like '<id1> holds <id2>'.
    - Second, classify each interaction by its type: 'multi subject relation' or 'functional action based interaction'.
    - Third, for each interaction, provide the exact sentence from the original caption where it was found.
    - Do NOT include self-directed actions, vague verbs, or internal states.
    - Your output must be a JSON array of interaction objects, with no extra text or explanation.
Output format:
    { "interaction": "<idX> <action verb> <idY>",
       "src1": "<idX>",
       "tgt1": "<idY>",
       "type": "...",
       "source_sentence": "..."}

Figure 26: **Prompt Design for Interaction Identification and Instance Assignment.**

You are a strict rater that evaluates an interaction triplet itself (e.g., '<id1> is holding <id2>').
Use the full available context(caption, ids) to determine CONTACT and DYNAMISM.
Scoring rules (integers 1–5):
    - Contact: 1=no contact; 3=uncertain/indirect; 5=direct/firm contact implied by the interaction
    - Dynamism: 1=static relation; 3=low/moderate movement/readiness; 5=strong action/state change
Video caption: *[caption]*
Interaction: *[interaction triplet]*
Noun of <id>: *[base nouns of <id>s in interaction]*
Detailed information of noun: *[detailed information of base nouns]*

Output Format:
{"Contact": <int 1-5>, "Dynamism": <int 1-5>, "Explanation": <short reason>}

Figure 27: **Prompt Design for Interaction Scoring and Filtering.**

Provide detailed information for each unique ID used above.
Make sure to include a detailed entry for every ID (e.g., <id1>, <id2>, <id3>) mentioned earlier.
For each ID, include:
  - "noun": a short and visually distinguishable noun phrase (e.g., "a man in a blue shirt", "a dog with brown fur")
This should be specific and concise to help an object detection model localize the entity.
  - "app": appearance or physical description (1 sentence)
  - "spatial": their spatial location or role in the scene (1 sentence)
Video caption: *[caption]*
Interaction: *[interaction triplet]*
Noun of <id>: *[base nouns of <id> in interaction]*

Output Format:
  { "<id1>": {"noun": ..., "app": ..., "spatial": ...},
    "<id2>": {...}, }

Figure 28: **Prompt Design for Instance Description Extraction.**

You are given one image crop (JPEG) of a detected object and a list of candidate IDs.
Each candidate has fields: id, noun, appearance.
Decide which ID best matches the crop.
If none of the candidate IDs clearly match, or if the object appears to be something else not described in the candidates, then you MUST return null.
Be conservative when uncertain.

Return STRICT JSON only:
{ "assigned_id": string|null, "confidence": number, "rationale": string }
- The detection label for this crop is: *[bbox_label]* (may help disambiguation)."

Candidate IDs (JSON array):
*[id_descriptions]*

Image to classify: *[image]*

Figure 29: **Prompt Design for Vision-Language Verification.**

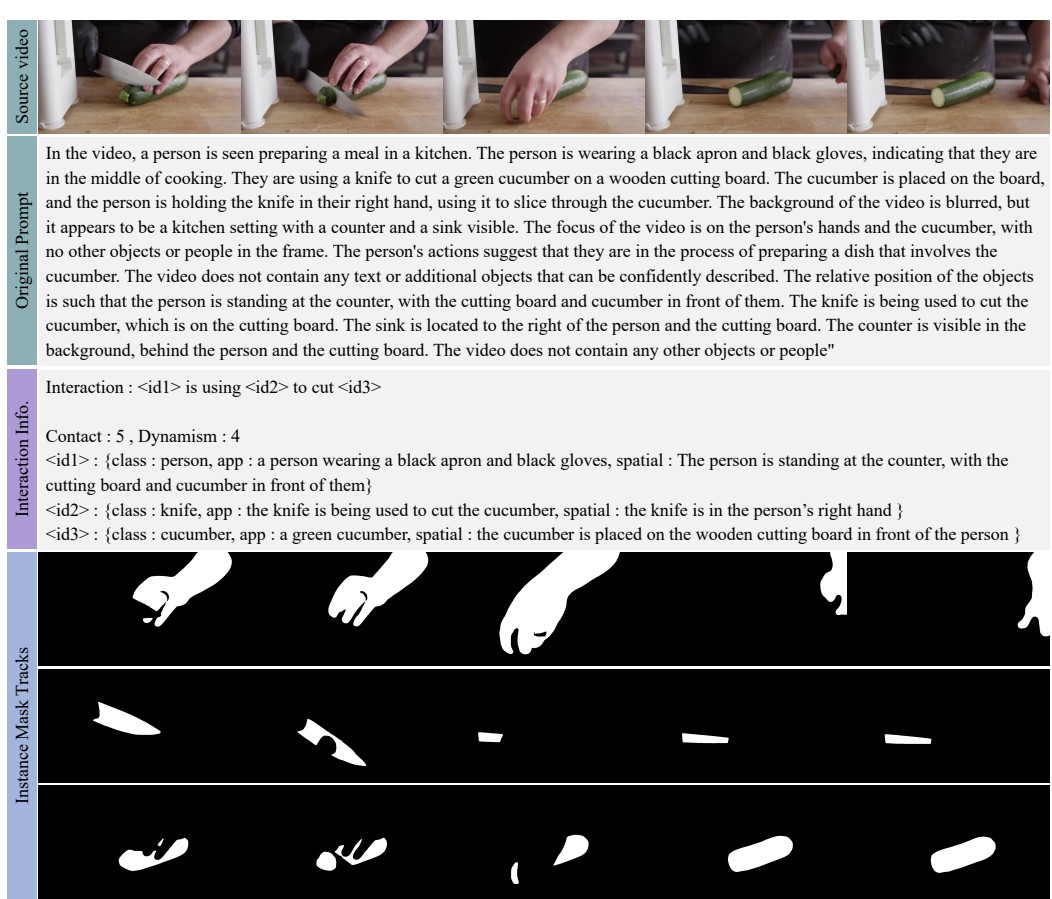

Figure 30: **Dataset Examples.**

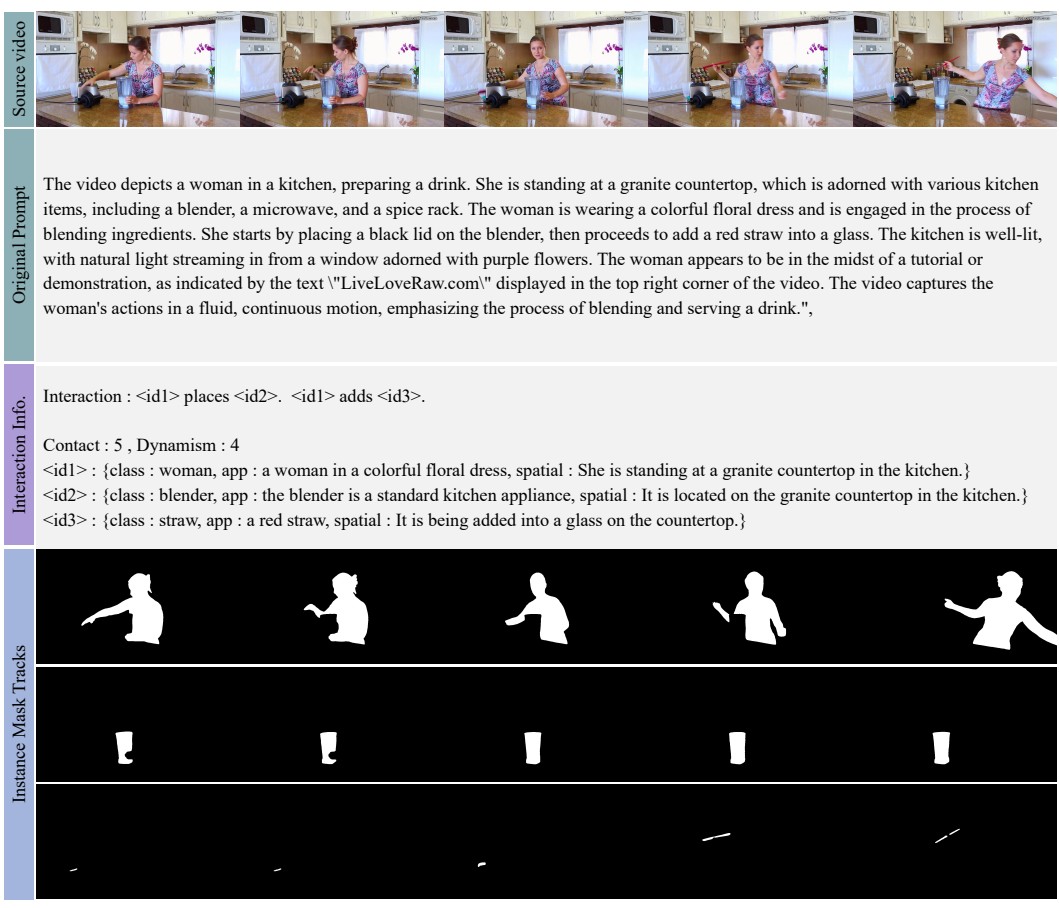

Figure 31: **Additional Dataset Example.**

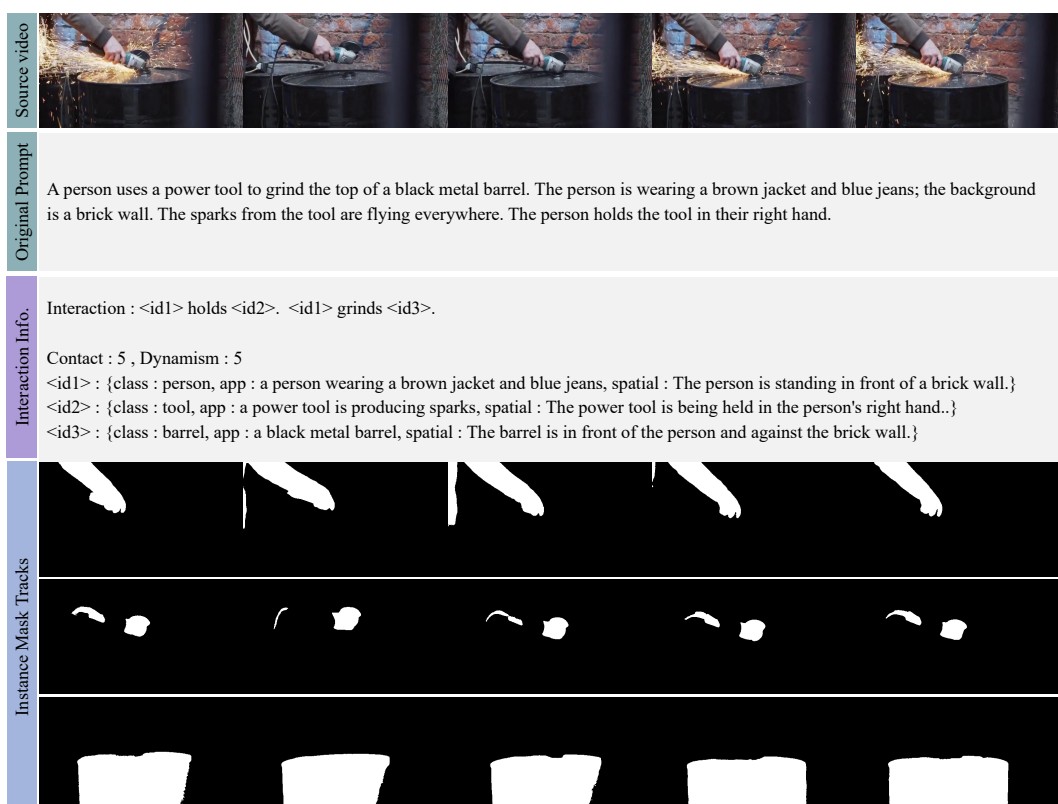

Figure 32: **Dataset Examples.**

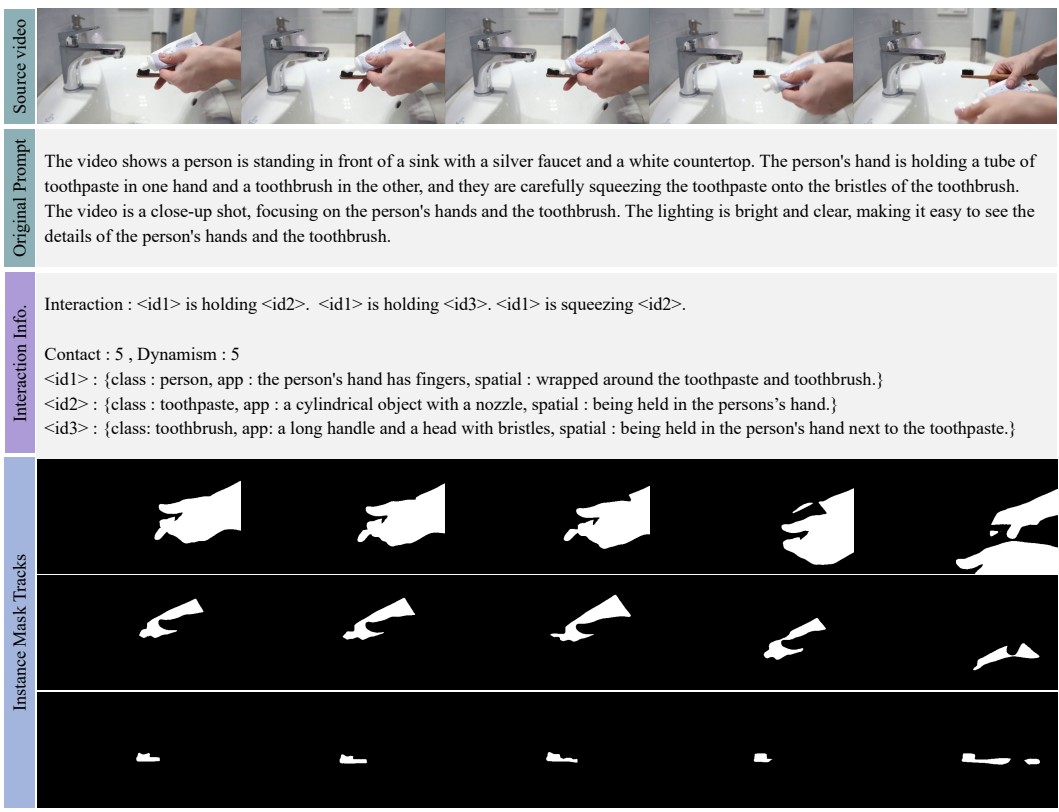

Figure 33: **Dataset Examples.**

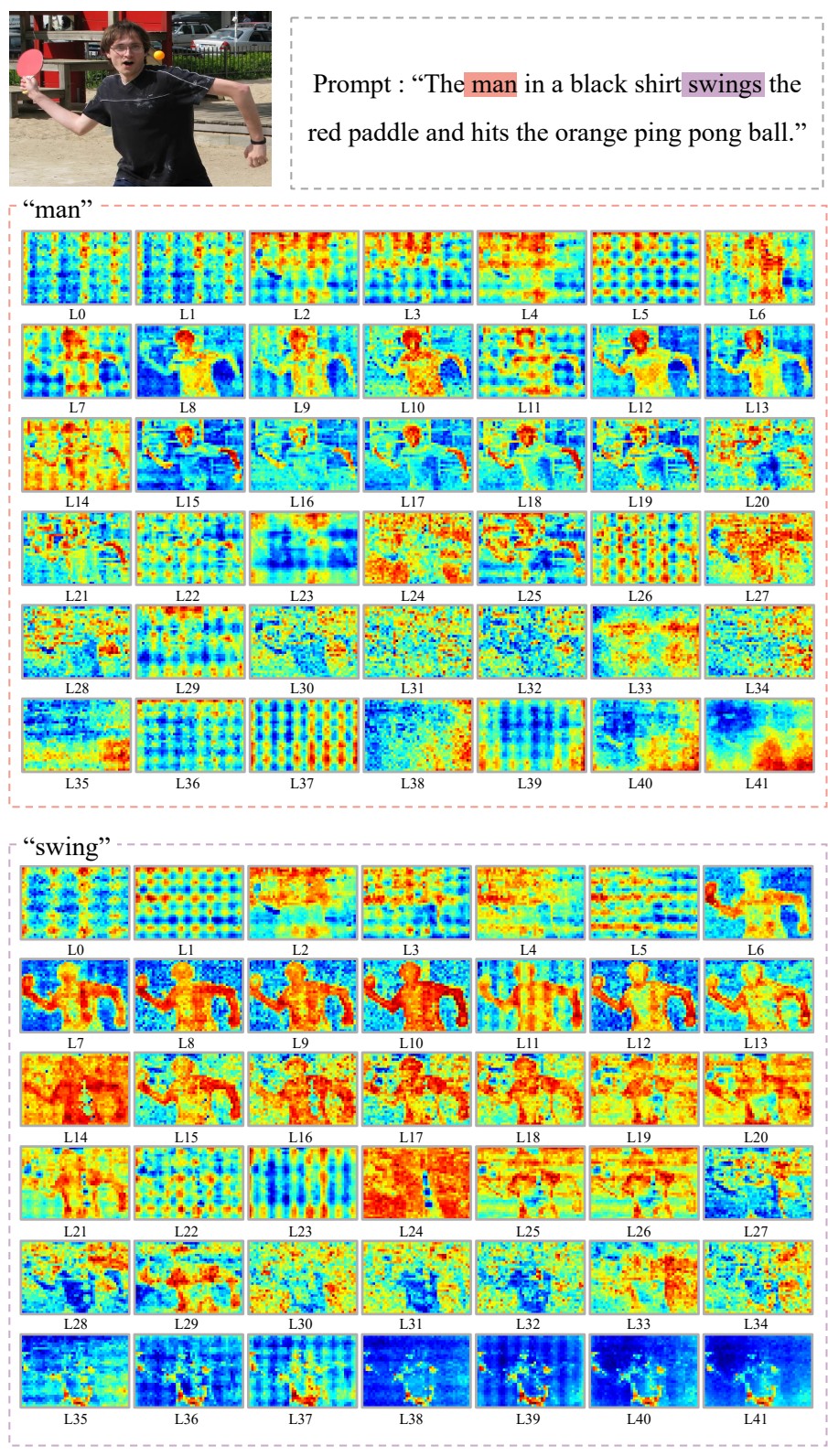

Figure 34: **Layer-wise Visualization Comparison.**

Task — answer yes/no questions about video frames
Inputs
An ordered list of frames (indexed from 0).
A list of yes/no questions. Each question specifies entities with their colored bboxes (e.g., "a man (yellow bbox) touches a cup (blue bbox)").

Rules
Judge by visible evidence only. Do not infer beyond what is clearly seen in the frames.
Color disambiguation. Because text alone may not uniquely identify an instance in a frame, use the specified colored bbox as the reference to pinpoint the intended entity, and base your judgment on that entity's visible evidence.

Per-Question Procedure
Select the decisive frame.
Scan frames and choose the single frame that gives the clearest evidence for "yes" or "no".
If multiple frames are equally decisive, pick the earliest index.
If no frame provides clear evidence, answer "no" and set frame_index to null.

Answer (yes/no).
Based solely on what is visible in the decisive frame (and color-tagged entities), answer "yes" or "no".

Visual plausibility check (on the decisive frame).
If the decisive frame shows visually implausible anatomy/geometry, override the answer with "no (visually implausible)".
Plausibility red flags include (not exhaustive):
Human anatomy anomalies: duplicated/missing hands/feet/arms, impossible joint bends, detached limbs.
Object/body fusion/splitting artifacts within a bbox, or severe distortions that break physical continuity.
Self-intersection or impossible penetration (e.g., hand passes through a solid object) that invalidates the observation.
Purpose: reject interactions that "occur" but are visually nonsensical.

Output (JSON)
Return an array of objects:
[
{ "question_id": 1, "answer": "yes", "frame_index": 12 },
{ "question_id": 2, "answer": "no (visually implausible)", "frame_index": 7 },
{ "question_id": 3, "answer": "no", "frame_index": null }
]
answer ∈ {"yes", "no", "no (visually implausible)"}

frame_index is the decisive frame used to judge the answer (or null if none was decisive).

Figure 35: **Prompt design for KISA and SGI in Evaluation Protocol.**

# Role
You are a hallucination detection expert.
Your task is to evaluate a sequence of video frames relative to a fixed anchor frame (frame 0) and determine whether:
- Any known entity class emerged without a bounding box, or
- Any previously tracked instance disappeared in later frames.
---
## Inputs
You are given:
- `frame_0`: the anchor/reference frame
- `frames_k`: a list of frames where k ∈ [1, N]
- Each frame contains bounding boxes, and every bbox is defined by a `(class, color)` identity
- A color-to-class mapping JSON:
```json
{
"entities": ["person", "cup", "paper"],
"colors": ["green", "red", "blue"]
}
```
- The arrays `entities` and `colors` are index-aligned, e.g., `"red"` → `"cup"`
Use this mapping to identify and track instances consistently across all frames.
---
## Detection Rules
### 1. Emergence
Mark `emergence = "yes"` if any frame *k* contains:
- An unboxed object of a class listed in `entities`, and
- That class had no visual instance (boxed or unboxed) in frame 0
This includes cases where:
- The object appears fully unboxed in the background
- The object appears embedded inside another bbox (e.g., a ball inside a person)
Track all frame indices where emergence occurred.
---
### 2. Disappearance
Let the set of `(class, color)` pairs from `frame_0` define the complete instance roster.
For each frame *k*, there must be a bbox with the same (class, color) for every such instance.
If any original instance is missing in frame *k*, mark `disappearance = "yes"` and include:
- The frame index *k*
- A description of which instances were lost (by `(class, color)` pair or class count)
---
## Output Format
Produce a single JSON object that summarizes emergence and disappearance across all frames:
```json
{
"emergence": "yes" | "no",
"emergence_frames": [<frame_idx_1>, <frame_idx_2>, ...],
"emergence_reason": "brief explanation or empty string if no",
"disappearance": "yes" | "no",
"disappearance_frames": [<frame_idx_1>, <frame_idx_2>, ...],
"disappearance_reason": "list missing instances as (class,color) and/or class-level count deltas"
}
```
## Evaluation Notes
- You must compare all frames after frame 0 against the instance roster from frame 0.
- Ignore any objects not listed in the `entities` array.
- Emergence is class-based: a second instance of a class (without a bbox) can be emergent if not present in frame 0.
- If no emergence or disappearance occurs in any frame, all values should default to `"no"`, `[]`, and `""`.

Figure 36: **Prompt design for SPI in Evaluation Protocol.**

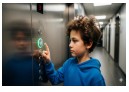 The boy in a blue hoodie with curly hair presses the round elevator button.

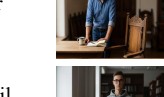 The man in a blue shirt with rolled-up sleeves pushes the wooden chair toward the table.

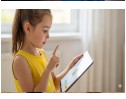 The girl in a yellow dress with a ponytail taps the black tablet screen.

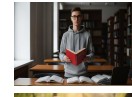 The student in a gray hoodie with glasses places a red book on the desk.

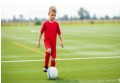 The boy in a red jersey with short blond hair kicks the white soccer ball on the field.

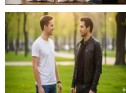 The friend in a white T-shirt hugs his friend in a black jacket in the park.

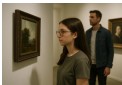 A girl with glasses touches a framed painting in a quiet art gallery with soft lighting and white walls while a man is walking behind.

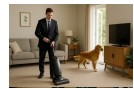 A man in a business suit uses a vacuum cleaner on a beige carpet, and a golden retriever runs toward the open door at the back of the living room.

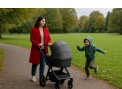 A woman in a red coat pushes a stroller along a park path with fallen leaves scattered around. Nearby, a child in a green hoodie jumps with excitement on the grassy field.

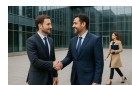 A man in a business suit is shaking hands with another man in front of a glass office building, while a woman nearby is walking across the plaza with a folder in her hand.

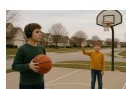 A boy wearing headphones throws a basketball toward a hoop in a quiet neighborhood court, while another boy waves from the sideline.

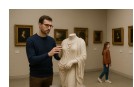 In a museum, a man in glasses touches a sculpture with curiosity, while a young girl walks slowly past a row of paintings on the wall.

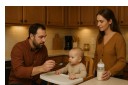 A man feeds a baby in a high chair while a woman holds a baby bottle in a cozy kitchen with warm lighting and wooden cabinets.

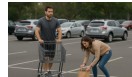 A man in workout clothes pushes a shopping cart in a parking lot, while a woman next to him picks up a grocery bag from the ground.

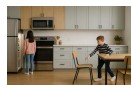 A girl in a pink sweater opens a refrigerator, and her brother pulls a chair toward the kitchen table in a modern home interior.

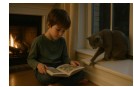 A boy reads a picture book beside a fireplace, while a cat on the windowsill touches a toy mouse with its paw.

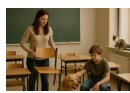 A woman lifts a chair in a classroom, while a boy pats a dog sitting calmly near the desk.

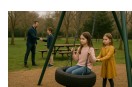 A girl pushes another girl on a tire swing at a park, while a man in the background is shaking hands with a boy near the picnic tables.

Figure 37: **Generated Evaluation Dataset Pairs Example.**

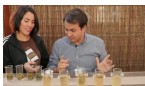 The woman in a black sports jacket hands over the sealed tea packet in front of the woman to the man in a blue shirt.

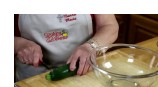 The woman slices the zucchini with the kitchen knife placed on the wooden counter.

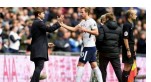 The soccer player exchanges a high five with the coach near the sideline after being substituted.

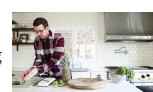 The man in a checkered shirt gently holds a bowl of prepped vegetables, his hands steady as if ready to transfer them into a pan.

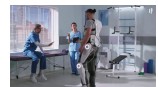 The female nurse taps on the tablet screen to start recording the man's gait pattern.

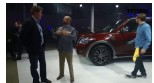 A man walking past with yellow towel wipes the front panel or windshield of the red SUV.

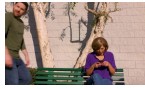 The man in a green shirt walks and sits down on green bench, settling next to the woman.

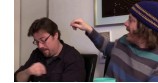 The man in a striped sweater and beanie gently pats the head of the man wearing glasses and a dark shirt.

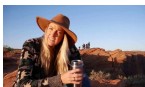 The woman in the wide-brimmed hat raises her silver travel mug to take a sip.

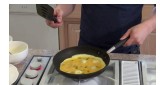 The person tilts the frying pan slightly to spread the egg mixture evenly across the surface.

Figure 38: **Sampled Evaluation Dataset Pairs Example.**

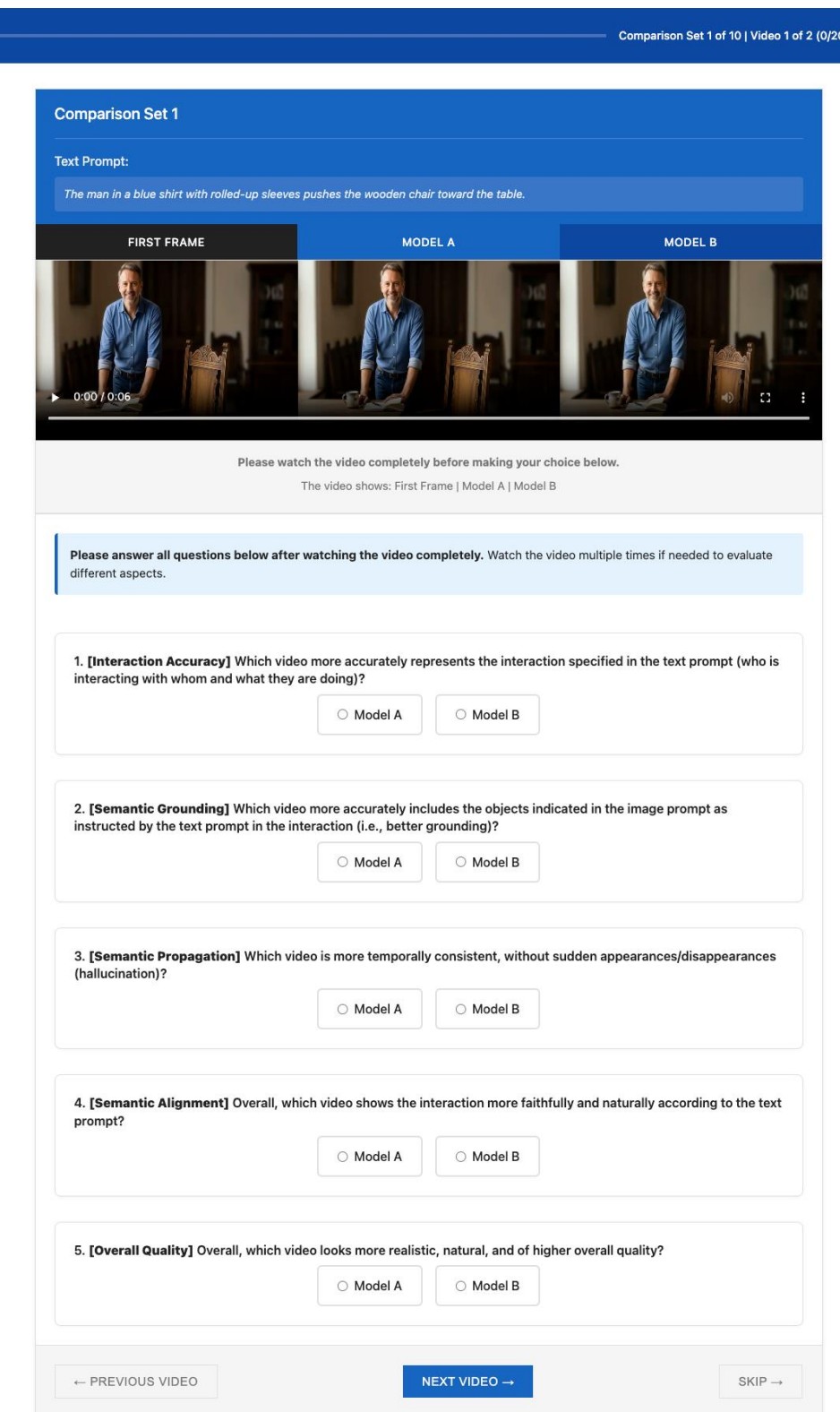

Figure 39: **An example of human evaluation.**

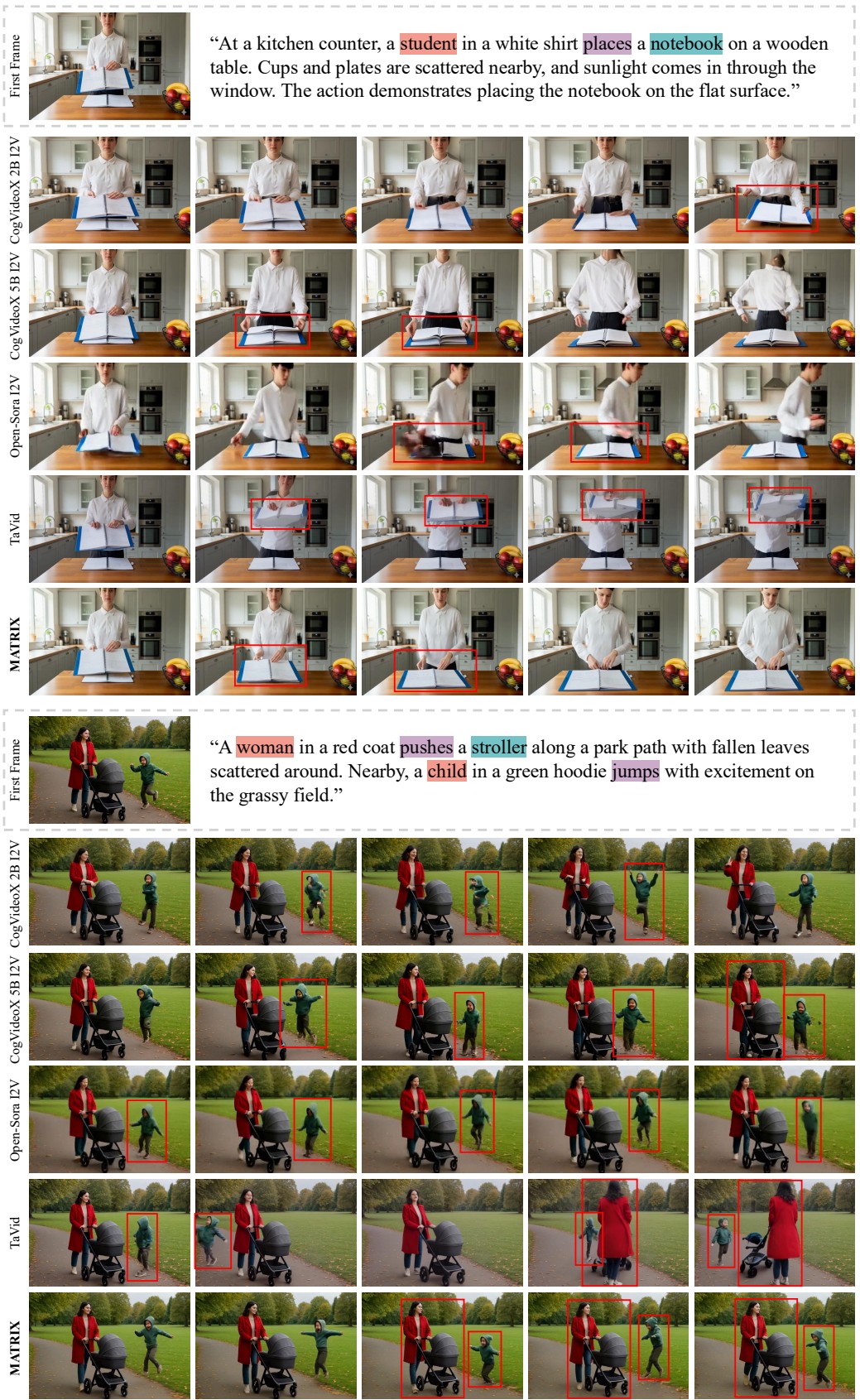

Figure 40: **Additional Qualitative Results.**

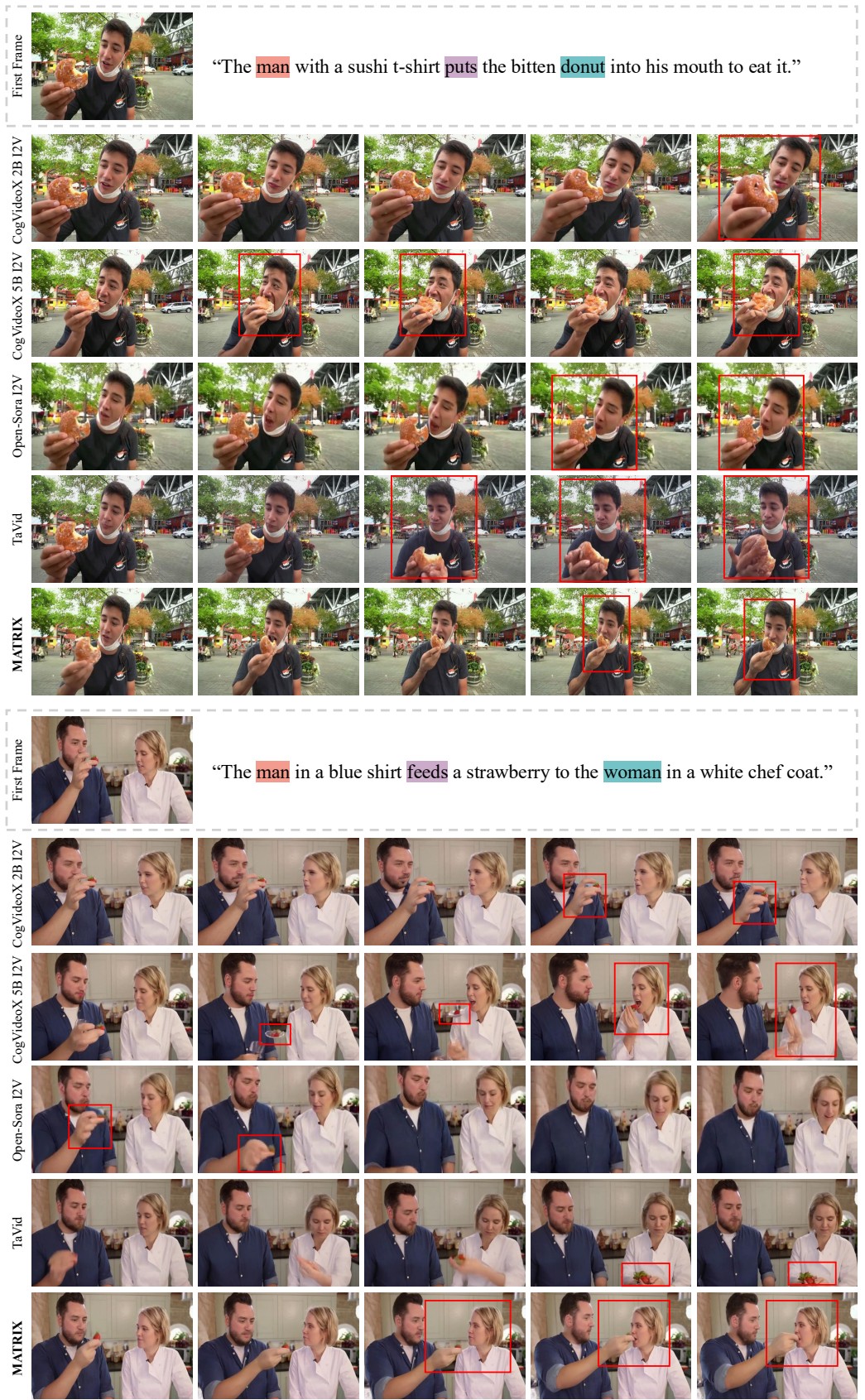

Figure 41: **Additional Qualitative Results.**

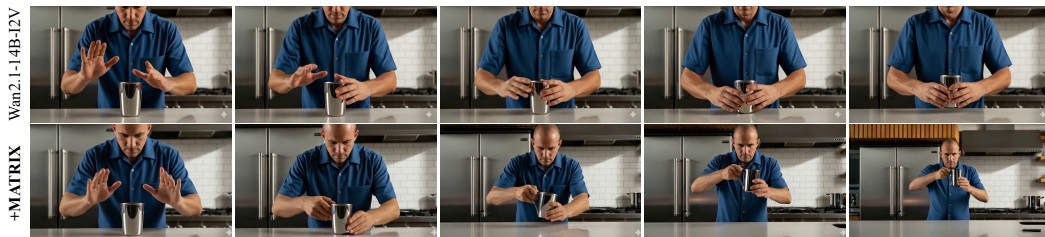

"Inside a cozy kitchen with white walls and wooden shelves, a man wearing a blue shirt stands in front of the fridge near the counter. He carefully holds a silver cup in his right hand while light from the window shines on the tiled floor."

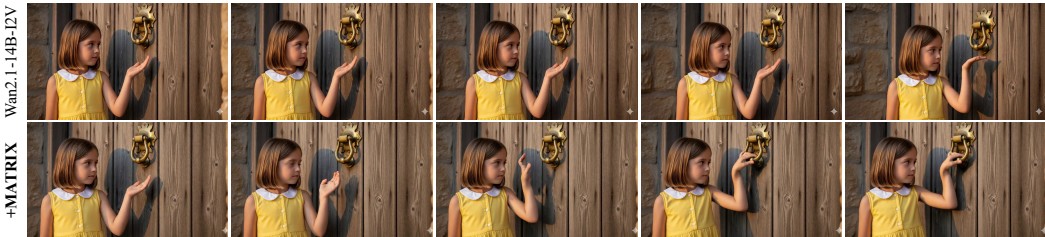

"At the front door of a house, a girl in a yellow dress stands outside near a wooden door. She lifts her hand and taps the metal door knocker lightly, producing a clear sound. The simple action shows her initiating contact with the door."

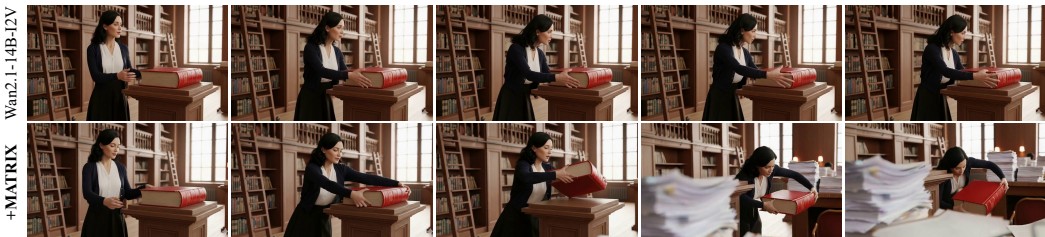

"In a library filled with tall wooden shelves, a woman in a black skirt stands near a study desk under a bright lamp. She bends slightly as she lifts a thick red book with both hands. The woman carefully raises the heavy book from the desk surface, preparing to move it to another spot in the library."

Figure 42: **Qualitative Comparisons between Wan-14B-I2V (Wan et al., 2025) and MATRIX (Ours).**

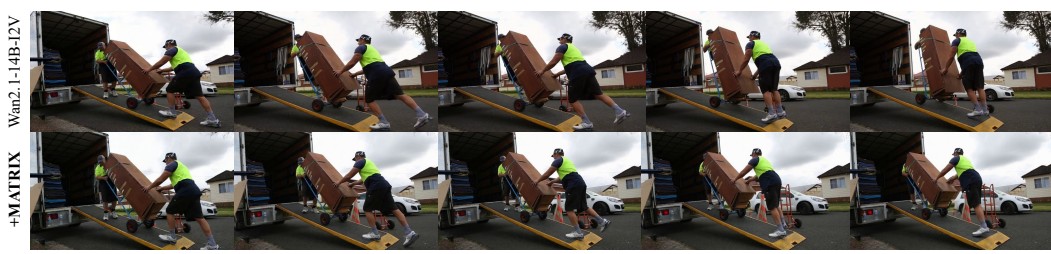

"The man on the ramp pushes the piece of furniture toward the truck while the man inside the truck pulls it inward."

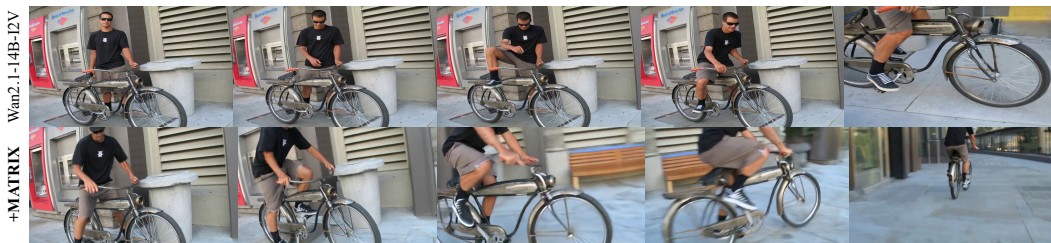

"The man wearing a black t-shirt and shorts gets on the bike and starts riding forward."

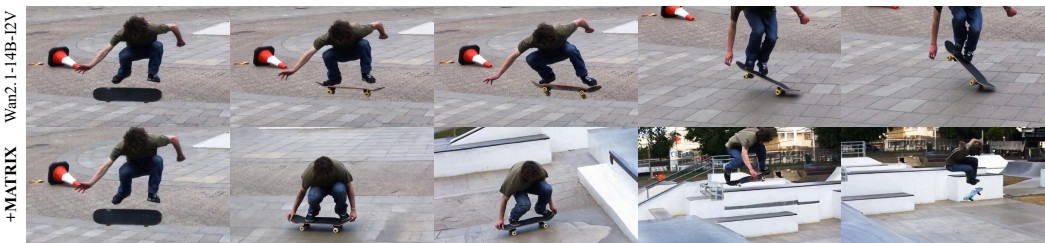

"The man lands on the skateboard and performs a jump."

Figure 43: textbfQualitative Comparisons between Wan-14B-I2V (Wan et al., 2025) and MATRIX (Ours).

