# OpenReview forum: "MATRIX: Mask Track Alignment for Interaction-aware Video Generation"
_ICLR.cc/2026/Conference — ICLR 2026 Poster_

### Official Review · Reviewer_NstU · 2025-10-20

**Soundness:** 2
**Presentation:** 2
**Contribution:** 1
**Rating:** 2
**Confidence:** 4

**Summary:**

The current work investigates how the object interactions are represented inside the video diffusion transformers and proposes a training strategy to refine them. More concretely, it investigates the attention map inside the CogVideoX-5B-I2V model in terms of text/visual tokens interactions, and observe that there is some object grounding (some spatial visual tokens have higher attention scores to the object/subject text prompt) and interaction grounding (the interaction verb attends to both the object and the subject). The curate a dataset of 11K videos and annotate instance-level segmentation mask tracks to quantify this grounding. This allows to locate the layers where grounding is the highest, and also to evaluate the interaction quality later. To improve the interaction modeling, the authors propose extra regularizations on attention maps pushing them closer to the correctly grounded segmentation maps.

**Strengths:**

- The paper is well written and easy to follow, and the illustrations are high-quality
- The proposed dataset could be useful for simply fine-tuning to improve the human-object and human-human interactions
- The included attention visualizations could be useful for future references
- The overall topic is quite important to the community

**Weaknesses:**

- The fact that text tokens are grounded to visual tokens in attention to some extent is a trivial observation and dates back to StableDiffusion-V1 editing pipelines for UNet+Attention architectures. The same observation was made for convolution-free architectures as well (i.e., DiT/UViT). One of the recent references is https://arxiv.org/abs/2505.04320 where it's visualized for Flux. I do not think that it's a surprising observation that similar mechanics happens for video DiTs. I would be more surprised if it would be something opposite. That's why the analysis is not insightful.
- The paper uses outdated baselines (e.g., CogVideoX-5B) instead of the modern ones (e.g., Wan 2.1/2.2)
- The paper does not compare to the baseline of simply fine-tuning on their curated data. I believe it's a critical ablation
- The submission does not include any samples from the dataset, making it hard to assess its quality
- The overall visual quality is far behind modern open-source models (e.g. Wan, Hunyuan, etc).

**Questions:**

- Typo: "suject-object" (the first sentence in the abstract)

---

> ### Author Response · Authors · 2025-11-26
> **Response to Reviewer NstU (1/5)**
>
> ## **W1. Insight of our attention analysis beyond prior studies**
>
> We appreciate the reviewer for raising this point. Our analysis does not aim to rediscover the known fact that text tokens can be grounded to visual tokens. Instead, we begin from this established behavior and ask a more specific question tailored to **interaction-aware video generation**:
>
> **“How do attention patterns behave during complex subject–verb–object interactions in videos, and where do they fail?”**
>
> Addressing this question leads to several differences from prior grounding visualizations, and other reviewer (**payz**) explicitly highlighted this as a novel and useful perspective on interaction-aware model behavior.
>
> **1. Video→text, instance-centric grounding (not text→image grounding)**
> Prior works, including Flux analysis referenced by the reviewer, primarily study text-to-visual effects, i.e., *“for a given text token, which image region does it attend to?”*
>
> In contrast, our semantic grounding is defined from the video token’s perspective, using video-to-text attention. For each spatio-temporal token along an instance mask track, we ask: *“which subject/object/verb token does this video token rely on?”*
>
> We then aggregate this over each tracked instance and role (subject, object, verb). This instance- and role-centric view is what we need to diagnose interaction failures (wrong subject, wrong object, verb not aligned with the actual contact region), and is not captured by conventional text→image attention
>
> **2. Temporal semantic propagation in multi-instance videos.**
>
> Most prior analyses are performed on single images. Our setting is video, where temporal consistency is critical. We study semantic propagation: whether subject/object/verb instances maintain stable attention to the correct text tokens across frames, or whether attention drifts, weakens, or jumps to other entities.
>
> This analysis directly relates to typical interaction failure modes—identity swaps, wrong targets, missing or duplicated interactions—highlighted in Fig. 2. Our question is not merely *“is there grounding once?”* but whether role-consistent grounding remains stable over time, as illustrated in Fig. 4 of the revised version.
>
> This temporal, multi-instance viewpoint makes the problem significantly more challenging and distinguishes our analysis from prior work.
>
> **3. Structured interaction: subject / object / verb, not just noun–object pairs.**
> Many prior visualizations examine simple noun–object correspondences (e.g., the “cat” token attending to the cat region). We instead decompose interactions into subject, object, and verb, where the verb corresponds to the physical interaction region.
>
> We analyze:
>
> (i) whether subject and object tokens align with their respective instance tracks, and
>
> (ii) whether the verb token aligns with the contact region.
>
> This allows us to assess whether the attention pattern encodes a coherent interaction structure, going beyond simple object presence.
>
> **4. From descriptive analysis to dominant layers and method design**
> Our goal is not merely descriptive. We use the analysis to guide where to intervene in the model:
>
> 1. We separate success and failure cases.
> 2. We examine grounding and propagation across layers.
> 3. We identify dominant layers where these differences are largest.
>
> We then selectively apply our interaction-aware objectives to these dominant layers. Ablations show that this targeted layer selection provides the strongest improvements in interaction fidelity.
>
> Thus, the analysis directly informs *where* interaction reasoning breaks down and *how* to design an effective training strategy.

---

> ### Author Response · Authors · 2025-11-26
> **Response to Reviewer NstU (2/5)**
>
> ## **W2. Experiments on other DiT-based baseline**
>
> We appreciate the reviewer for suggesting us to consider more recent DiT-based video models, such as HunyuanVideo and Wan. We emphasize that our method is compatible with any DiT-based video diffusion architecture. Following the reviewer’s suggestion, in the revised version, we explicitly extend our study to HunyuanVideo-I2V and Wan2.1-14B-I2V, reporting the corresponding analyses in Appendix C.
>
> For HunyuanVideo-I2V, which uses 3D full attention, we observe a dominant-layer structure that closely matches our findings on CogVideoX-5B-I2V. For Wan2.1-14B-I2V, whose architecture processes visual tokens via self-attention and textual tokens via separate cross-attention, we analyze the variance of attention mass across prompts, identify and remove near-invariant “dummy” layers, and still obtain a clear dominance pattern. These analyses and visualizations are provided in Appendix C of the revised version.
>
> Based on these observations, we further apply our simple and complementary loss design (SPA and SGA with selected dominant layers) to Wan2.1-14B-I2V, a strong state-of-the-art image-to-video baseline. Despite the already strong baseline performance, MATRIX further improves interaction fidelity, while preserving comparable overall appearance and video quality. Qualitative examples are provided in Fig. 10 of the revised paper, demonstrating the plug-and-play effectiveness of our framework across various DiT-based video models.

---

> ### Author Response · Authors · 2025-11-26
> **Response to Reviewer NstU (3/5)**
>
> ## **W3. Ablation on simple fine-tuning on our curated dataset**
>
> We appreciate the reviewer’s comment and fully agree that a baseline that simply fine-tunes on our curated dataset is an essential ablation. This setting is already included in Tab. 2 of the current submission, but we acknowledge that the descriptions of the variants were not sufficiently clear. In the revised version, we clarify the role of each variant and what each ablation is intended to demonstrate.
>
> Specifically, the key variants in Tab. 2 are organized as follows:
>
> - (I) Naive baseline: the original CogVideoX-5B-I2V model without additional conditioning or fine-tuning.
> - (III) Fine-tuning on curated data (LoRA, no layer selection): starting from (I), we introduce single-mask conditioning and fine-tune the model with LoRA on our curated interaction-aware dataset. This variant does *not* use our proposed layer selection or interaction-aware losses (SPA, SGA).
> - (IV)~ Layer selection with auxiliary losses: starting from (III), we incorporate our layer selection strategy and apply interaction-aware losses.
>
> Thus, (III) corresponds exactly to the baseline requested by the reviewer—namely, *“simply fine-tuning on their curated data.”* Our dataset contains (video, prompt, mask track) triplets, and the only architectural difference from (I) is the minimal modification required to support mask conditioning. No layer selection or interaction-aware objective is used in (III).
>
> Under this setup:
>
> - The performance gain from (I) → (III) reveals the effect of fine-tuning on our curated interaction-aware dataset under minimal changes.
>
> In the revised version, we explicitly clarify these mappings in both the Tab. 2 caption and the ablation discussion in the main text, so that readers can immediately understand that the requested baseline is already included and how each ablation step builds progressively upon the previous one.

---

> ### Author Response · Authors · 2025-11-26
> **Response to Reviewer NstU (4/5)**
>
> ## **W4. Dataset samples and quality**
>
> We thank the reviewer for raising this point and for their interest in our curated dataset. We agree that without any visual examples, it is difficult to fully assess the quality of the curation data and the effectiveness of our curation design. In the supplementary material, specifically, Fig. 30 and Fig. 31 show representative examples of our curated interaction annotations (subject–verb–object triplets) together with the corresponding instance mask tracks. These examples are selected to cover diverse interaction types and scene configurations. In addition, in the revised version, we have added two more qualitative examples (Fig. 32 and Fig 33) from the newly extracted portion of the dataset to further illustrate the variety and quality of the interaction-aware annotations. We explicitly reference these figures from the main paper (Section 3.2) so that readers can easily locate them.

---

> ### Author Response · Authors · 2025-11-26
> **Response to Reviewer NstU (5/5)**
>
> ## **W5. Visual quality compared to modern open-source models.**
>
> We appreciate the reviewer’s comment regarding visual quality. In this work, our main backbone is CogVideoX-5B-I2V, which is indeed significantly smaller than recent open-source models such as Wan 2.1/2.2 (14B) and Hunyuan (13B), both in terms of parameter count and pretraining resources. We therefore agree that, in terms of absolute sharpness and fine-grained detail, our generations can appear behind those very large modern models.
>
> At the same time, the goal of our work is not to propose a new ultra-large backbone, but rather to study ”how a simple, interaction-aware analysis and plug-and-play design can improve interaction fidelity under a fixed model capacity”. From this perspective, we also include a comparison with a larger open-source model, OpenSora-11B. In complex interaction scenarios, we show that CogVideoX-5B-I2V with our loss design often achieves higher interaction fidelity than OpenSora-11B, both qualitatively and quantitatively (see Tab. 1, the main qualitative figures and supplementary video). This highlights that our method provides improvements that are complementary to simply scaling model size: even with a relatively modest backbone, we obtain meaningful gains in interaction quality.
> Moreover, in the revised manuscript we further show that applying our analysis and loss design to the recent Wan2.1-14B-I2V backbone also improves interaction fidelity (see Sec. 6.4 and Supplementary Figures. 42–43, Supplementary Videos), indicating that our findings are not tied to a single model.
>
> **Q1. Typo Check**
>
> We thank the reviewer for pointing out this typo. We have corrected it in the revised version.

---

### Official Review · Reviewer_payz · 2025-10-21

**Soundness:** 3
**Presentation:** 4
**Contribution:** 4
**Rating:** 8
**Confidence:** 4

**Summary:**

This paper analyzes a common problem in video models, incorrect multi-object interactions. It identifies key concepts that allow analyzing two aspects of this problem (semantic grounding and propagation), and uses these concepts and their corresponding metrics to understand how these interactions are represented inside a typical DiT model. Based on this analysis and metrics, the paper introduces 1) a method for regularization (Matrix), 2) an evaluation protocol (InterGenEval), and  3) a dataset with 11k videos (Matrix-11k) with properly labeled interactions and instances.

The method for effective regularization is based on locating the layers that most impact on semantic grounding and propagation, and fine-tuning those using LoRA on the corresponding losses for those two concepts: Semantic Grounding Alignment and Semantic Propagation Alignment.

**Strengths:**

This paper is a strong work in all the important dimensions:

* originality: to the best of my knowledge this is novel work
* quality: the problem is analyzed from an interesting perspective, starting with an analysis that provides a new understanding on how the interactions are represented in particular layers of the network, and the solution (introducing new concepts, a regularization method, a dataset, and an evaluation benchmark) presented properly deals with the problem
* clarity: the paper is well written and easy to follow
* significance: this is where I doubted the most, because while the problem is important for video generation, it was unclear to me that per se it would justify the added complexity in the training system (even if based on something as lightweight as LoRA). However, the use of this method is also beneficial for overall quality and other important metrics, which is intriguing, impactful, and to me brings final justification to this additional complexity.

**Weaknesses:**

* There is no analysis about how this method impacts training time. Especially important would be to understand whether the benefits of fine-tuning are as strong as presented, when compared to a training run with the same amount of flops used (iso-FLOPS comparison)

* The dataset used for eval, based on only 60 synthetic and 60 real pairs appears too small. It would be useful to include analysis on what to expect, in terms of noise, from this dataset size

* While ablations for most important aspects of this work are provided, there's little justification for why Av2t  and Av2v are the best proxies for semantic grounding and propagation. Could one use At2v for the former, or even reuse the Av2t for the latter? What other alternatives were considered? While it's not reasonable to expect a completely exhaustive ablation for all possible alternatives, even a small explanation would improve on this.

* this work focuses on actions that require contact between different objects in the videos, such that the fine-tuning process only includes such actions in the dataset. This is a reasonable assumption for this particular paper (but as a minor improvement, it could be interesting to consider how the work could be expanded to better support contact-less interactions). However it would be useful to understand how this choice impacts other model properties, more specifically:
  * is the fine-tuned model more prone to generate contact actions even when provided with a contactless prompt?
  * minor: it would be interesting to understand, also, whether a fine-tuned model improves on contactless actions as well, even if those aren't included in the expanded loss

* similarly, since the primary source HOIGen is described to be mostly focused on human-based interaction,  it seems useful to ask what's the impact on non-human video generation

* minor: given the strong results using LoRA, the paper could benefit from some discussion about whether using the same method on large scale pretraining

**Questions:**

* What are your insights regarding which layers are the most influential? do you find that they are consistently placed around the same locations in different kinds of networks?

* Have you considered how general this method is; would you expect similar results in non-DiT based networks for example?

---

> ### Author Response · Authors · 2025-11-26
> **Response to Reviewer payz (1/7)**
>
> ## **W1. Regarding training cost**
>
> To address the reviewer’s concern, we compare training costs under a near-iso-FLOPs setting for three variants: the naïve CogVideoX-5B-I2V model, a LoRA-only baseline, and our full method (LoRA selection + SGA + SPA).
>
> All fine-tuning experiments are conducted on a single A6000 GPU at a resolution of 480×720 with the same batch size. Under these controlled conditions, the wall-clock time per epoch serves as a practical proxy for total FLOPs. As summarized in Table 5 in the Appendix, the LoRA-only baseline (“baseline w/o selection”) requires **24.43 hours/epoch**, while our full method (“ours”) requires **31.09 hours/epoch**, corresponding to an additional **+6.7 hours**.
>
> Despite this modest overhead, the gains on interaction-centric metrics are substantially larger. Relative to the naïve CogVideoX-5B-I2V (no LoRA), our method improves:
>
> - **KISA:** +34.5%
> - **SGI:** +30.5%
> - **IF:** +32.1%
>
> Even compared directly against the LoRA-only baseline, our method still yields:
>
> - **KISA:** +22.7%
> - **SGI:** +21.9%
> - **IF:** +22.0%
>
> while holistic appearance metrics (HA/MS/IQ) remain essentially unchanged.
>
> Importantly, this efficiency comparison directly addresses joint-training strategies (e.g., VideoJAM), which typically require training a video backbone from scratch or performing heavy fine-tuning, and therefore incur significantly higher computational cost. In contrast, our approach is a lightweight, *plug-and-play LoRA-based fine-tuning* built on top of an existing CogVideoX-5B-I2V backbone. It achieves **30–35% relative improvements** on core interaction metrics using only a single-GPU setup and with minimal additional training cost.
>
> In other words, an increase of roughly **6.7 hours** per epoch leads to **≈30–35%** relative improvements in interaction awareness, without degrading image quality. This indicates that the gains of our method are not attributable to “training longer” but instead stem from the *interaction-aware fine-tuning strategy* itself.
>
> In the revised version, we make this computational-efficiency analysis explicit to more clearly demonstrate the practicality and effectiveness of our method.

---

> ### Author Response · Authors · 2025-11-26
> **Response to Reviewer payz (2/7)**
>
> ## **W2. Size and reliability of the evaluation dataset**
>
> Our evaluation is tailored to the image-to-video, interaction-centric setting, where each test case must satisfy all of the following requirements:
>
> (i) a single input frame that clearly depicts the initial state of the scene;
>
> (ii) corresponding instance masks for the subject and object in this first frame; and
>
> (iii) an interaction-centric text prompt specifying the expected future interaction within the generated video.
>
> Public benchmarks that simultaneously satisfy all three criteria are, to the best of our knowledge, extremely scarce. We also note that recent evaluations for physics or interaction aware video generation, such as PhyGenBench [1], likewise adopt relatively small but carefully curated test suites (e.g., 160 text-to-video examples), indicating that compact, high-quality evaluations are common in such specialized settings.
>
> Existing image-to-video benchmarks are generally unsuitable for our protocol: their prompts typically emphasize appearance, scene style, or simple object motion rather than explicit, meaningful interactions, and they rarely provide a unique subject–object pair together with first-frame masks. Conversely, HOI-oriented benchmarks contain interaction labels but define them at the *video level*. As a result, the annotated interaction may not occur near the designated starting frame, and in many clips the subject and/or object are not even visible in that frame. Such annotations are acceptable for training, where the entire video and global caption can be used jointly, but they are inappropriate for our inference setting, which must begin from a single initial frame with masks and an interaction prompt that are temporally and semantically aligned with the expected upcoming motion.
>
> For this reason, we manually curated an evaluation set that strictly meets the above criteria, yielding 60 synthetic and 60 real interaction-centric pairs in the initial submission. Following the reviewer’s suggestion, we have additionally collected 30 more cases under the same protocol (initial frame, instance masks, and aligned interaction prompt). Results on this extended evaluation set are included in the revised manuscript, and the performance trends observed in the original 60+60 set remain consistent.
>
> To provide clearer evidence of reliability, we also report, for each metric, the mean and standard deviation across evaluation clips (Fig. 22 in the revision), enabling readers to assess both the average improvements and the variability at this dataset size. Our preliminary analysis shows that the gains of our method over baselines and competing models exceed the observed variance, suggesting that the improvements are not simply due to noise.
>
> Finally, we will release this carefully constructed evaluation set as a benchmark for assessing interaction awareness. Fig. 37 and  Fig. 38 in Appendix illustrate representative examples from the evaluation set. And Fig. 24 in the Appendix presents the results of additionally collected cases, including non-contact and non-human interaction scenarios.
>
> **Citations** :
>
> [1] F. Meng, et al. Towards World Simulator: Crafting Physical Commonsense-Based Benchmark for Video Generation, ICML 2025

---

> ### Author Response · Authors · 2025-11-26
> **Response to Reviewer payz (3/7)**
>
> ## **W3. Choice of v2t and v2v as proxies for semantic grounding and propagation**
>
> **W3-1. Why v2t rather than t2v for semantic grounding?**
> For semantic grounding, our objective is to determine, for each **video token**, which **text token** (subject, object, verb) it depends on. Each row of **v2t attention** can be interpreted as: *“given this video token, how strongly does it attend to each subject/object/verb token?”* This aligns directly with our instance-level grounding analysis, where we evaluate whether video tokens consistently attend to the correct textual role.
>
> By contrast, **t2v attention** answers a different question: *“for a given text token, which video locations does it attend to?”* While meaningful, this does not directly provide the per-video-token grounding signal required in our analysis.
>
> If we were to use t2v attention, we would have to **invert the mapping**—from “word → spatial region” back to “each spatial token → word.” This inversion requires additional normalization and becomes ambiguous, especially when multiple instances of the same class appear (e.g., two men), since several text tokens may attend to overlapping video regions.
>
> For these reasons, we adopt **v2t attention** as the primary grounding signal, as it (i) directly corresponds to the per-token grounding we need, and (ii) directly affects how video tokens are updated during generation.
>
> In the revised version, we clarify this asymmetry and note that t2v attention is complementary but less aligned with our grounding objectives. We also include an additional **t2v-based grounding variant** in Tab.2 and compare its behavior with ours.
>
>
> **W3-2. Why use v2v attention and not reuse v2t for propagation?**
>
> We agree that, in principle, one might consider reusing v2t for both grounding and propagation, since text tokens are shared across frames. However, our formulation of **semantic propagation** concerns physical continuity, whether a point on an object at time $t$ is correctly linked to the corresponding point at time  $t+δt.$
>
> This notion is best captured by **v2v attention**, whose function is precisely: *“for this video location, where should it map to in other frames?”* Thus, v2v attention naturally reflects motion and temporal correspondence, making it the appropriate proxy for propagation.
>
> In contrast, **v2t attention does not capture intra-video continuity**. A video token for the subject at frame $t$ and another at frame  $t+δt$ may both attend strongly to the subject’s text token, but this only indicates shared semantics, not how the physical region has moved over time. v2t answers the question *“which word describes this token?”*, whereas propagation requires answering *“where does this physical point go?”*.
>
> Our ablations (Tab. 2; rows 6 vs. 7) show that adding the v2v-based propagation term further improves interaction metrics, confirming that propagation cannot be addressed by v2t alone. We make this reasoning clearer in the revised manuscript and report comparisons across alternative variants, including **v2t vs. t2v** for grounding (in Tab. 2).

---

> ### Author Response · Authors · 2025-11-26
> **Response to Reviewer payz (4/7)**
>
> ## **W4. Discussion on contact-focused finetuning and contact-less interactions**
>
> We appreciate the reviewer for this thoughtful question. Our primary goal in this work is to address failure modes that arise in complex multi-instance interactions, including:
>
> (1) identity swaps between different instances,
>
> (2) interactions being directed toward the wrong target,
>
> (3) duplicated identities or duplicated interactions, and
>
> (4) interactions failing to occur at all.
>
> In practice, these issues are especially severe in **contact-rich interactions**, where precise spatial alignment and temporal consistency are critical. For this reason, we focused our fine-tuning dataset on actions involving clear physical contact, ensuring that our interaction-aware objectives are trained on the most challenging and informative scenarios.
>
> **W4-1. Does contact-focused finetuning make the model over-generate contact?**
>
> Within the scope of our experiments, we do **not** observe a tendency for the fine-tuned model to hallucinate contact when the prompt explicitly describes a contactless interaction. For prompts such as “A man in a gray hoodie pointing his finger at another man near a street food market” or “A woman points her finger toward the red lamp”, the fine-tuned model does **not** force the agents into contact, as illustrated in Fig. 24 (a). In other words, although the fine-tuning data is biased toward contact-rich interactions, we do not see a collapse in which the model disregards clearly contactless textual instructions and always produces physical contact.
>
> **W4-2. What happens to contactless interactions, even though they are not explicitly targeted by the loss?**
>
> Although our expanded loss is defined on contact-rich interactions, we **qualitatively observe improvements on contactless interactions as presented in Fig. 24 (a)**. In scenarios such as waving, dancing without touching, or running toward another agent, typical failure modes of the base model, such as identity confusion, incorrect target assignment, or the interaction failing to initiate, are also mitigated. We interpret this as a form of *generalization*: by enforcing stronger subject–object separation and temporal consistency on the more demanding contact cases, the model learns more robust semantic bindings that naturally transfer to simpler contactless interactions.

---

> ### Author Response · Authors · 2025-11-26
> **Response to Reviewer payz (5/7)**
>
> ## **W5. Discussion on the impact of human-centric data for non-human video generation.**
>
> We thank the reviewer for raising this important point. While HOIGen is indeed primarily focused on human-based interactions, our training data also includes PE-Video, which contains a broader variety of general (including non-human) interaction scenarios. More importantly, our formulation of “interaction” is not restricted to humans. We define interactions purely through a **subject–verb–object** structure, so non-human cases (e.g., *“a cat reaching for a toy”*) are handled in exactly the same way as human–object interactions. We note that our evaluation set includes non-human interactions. Alongside human-object examples, we include cases involving animals and other non-human agents, ensuring that these scenarios are part of the test distribution.
>
> To address the reviewer’s concern more directly (and in line with W4), we further constructed a dedicated evaluation subset consisting solely of non-human or non-contact interactions, as illustrated in Fig. 24 (b). Across this subset, we observe consistent improvements in interaction fidelity after fine-tuning.
>
> We clarify these points in the revised manuscript and explicitly highlight the non-human subset results and qualitative examples to better illustrate our method’s effectiveness in non-human interaction video generation.

---

> ### Author Response · Authors · 2025-11-26
> **Response to Reviewer payz (6/7)**
>
> ## **W6. Large-scale pre-training**
>
> We appreciate the reviewer for this insightful suggestion. In the current paper, we intentionally focus on a resource-efficient LoRA fine-tuning setting, and show that even a lightweight adaptation on top of an existing large video diffusion model substantially improves multi-instance interaction fidelity.
>
> This highlights two key observations:
>
> (i) despite extensive large-scale video pretraining, the base model still exhibits systematic failures **in complex interaction scenarios (identity swaps, incorrect targets, missing, hallucinated, or duplicated interactions),**
>
> (ii) simply “seeing more data” or relying solely on standard pixel-level reconstruction objectives is insufficient for robust interaction awareness.
>
> Our ablations (Tab. 2, Section III in the main paper) further show that performance saturate even when fine-tuning on high-quality, interaction-centric data **unless** we introduce an explicit interaction-aware objective. **This supports our claim that the proposed losses (SGA, SPA) provide complementary supervision beyond data scale alone.**

---

> ### Author Response · Authors · 2025-11-26
> **Response to Reviewer payz (7/7)**
>
> ## **Q1 & Q2. Analysis of dominant layers and generalizability across DiT-based video models**
>
> In response to the reviewer’s question about which layers are most influential and how general the observed dominance patterns are, we extend our analysis beyond CogVideoX-5B-I2V to two additional recent DiT-based video models, **HunyuanVideo-I2V** and **Wan2.1-14B-I2V,** in the revised manuscript Appendix C. Although the specific layer indices differ across architectures (e.g., 3D full-attention blocks vs. self-then-cross-attention designs, and varying numbers of layers), we consistently observe that **a small subset of layers exhibits dominant grounding or propagation behavior**. These results and visualizations are reported in the Appendix C of the revised version.
>
> For non-DiT backbones, such as U-Net–based diffusion models, we have not yet conducted a systematic study. Therefore, we do not claim that the same dominance patterns will automatically transfer, as our current analysis explicitly assumes MM-DiT–style blocks and their associated attention structure. Nonetheless, prior work on U-Net–based diffusion models has indicated that certain decoder stages are primarily responsible for semantic alignment or high-level correspondence, suggesting that analogous “grounding” and “propagation” functional roles may also exist in U-Net-like architectures. This hints that our analysis methodology and interaction-aware loss design could potentially be adapted to these models as well.
>
> We regard extending this investigation to non-DiT backbones as an interesting direction for future work, and we clarify this in the revised manuscript.

---

### Official Review · Reviewer_TLG6 · 2025-11-01

**Soundness:** 4
**Presentation:** 3
**Contribution:** 4
**Rating:** 8
**Confidence:** 3

**Summary:**

The paper investigate why t2v diffusion transformers struggle with multi instance, subject object interactions and proposes a targeted fix. The authors proposed MATRIX-11K, a dataset of ~11K videos with interaction-aware captions and multi instance mask tracks. And analyzed where interaction semantics emerge inside 3D full attention of video DiTs, finding that semantic grounding and semantic propagation concentrate in a small set of layers, and introduced the lightweight regularization that aligns attention in those layers to the mask tracks via SGA and SPA loss. On their benchmark, MATRIX improves interaction fidelity and reduces drift and duplication.

**Strengths:**

1. A deep and comprehensive clear failure analysis, and the MATRIX fixed that.
2. The data pipeline of the seg data generation is well designed, pragmatic and reproducible.
3. Simple, effective objective. The SGA and SPA losses are standard but cleverly applied to attention maps rather than pixels alone, brings lower computational cost and easy to adapt into Dit.

**Weaknesses:**

1. It mainly compared and analyse with CogVideoX, as this work is a 2025 work, it probably need to do more with the more recent models like wan and hunyuan.
2. 11K dataset may not that a good amount for the video generation tasks, especially for those larger and new models, the society may be benifit if the dataset could be larger
3. Like the other related work videojam or udpdiff, they used optical flow, seg, depth to help with the video generation, while in this work only segmentation is used, consider to include discussion on involving others would beneficial.

**Questions:**

see weakness

---

> ### Author Response · Authors · 2025-11-26
> **Response to reviewer TLG6 (1/3)**
>
> ## **W1. Additional analysis and comparison with state-of-the-art models**
>
> We thank the reviewer for pointing us to more recent DiT-based video models, including HunyuanVideo and Wan. We emphasize that our method is compatible with any DiT-based video diffusion architecture. Following the reviewer’s suggestion, we have extended our study to HunyuanVideo-I2V [1] and Wan2.1-14B-I2V [2], and we report the corresponding analyses in Appendix C of the revised manuscript.
>
> For HunyuanVideo-I2V, which adopts 3D full attention, we observe a dominant-layer structure that is consistent with our findings on CogVideoX-5B-I2V. For Wan2.1-14B-I2V, whose architecture processes visual tokens via self-attention and textual tokens via separate cross-attention, we analyze the variance of attention mass across prompts, filter out near-invariant “dummy” layers, and still find a clear dominance pattern. These additional results, together with detailed visualizations, are provided in the revision Appendix C.
>
> Building on this analysis, we also apply our simple and complementary loss design (SPA and SGA) to Wan2.1-14B-I2V, the current state-of-the-art image-to-video model. Despite the already strong baseline performance, MATRIX further improves interaction fidelity while maintaining comparable overall appearance and video quality. The corresponding qualitative results are included in Fig. 10, 42 and Fig. 43 of the revised paper. These analyses and experiments further demonstrate the plug-and-play effectiveness of our framework.
>
> **Citations** :
>
> [1] W. Kong, et al. HunyuanVideo: A Systematic Framework For Large Video Generative Models
>
> [2] A. Wang, et al. Wan: Open and Advanced Large-Scale Video Generative Models

---

> ### Author Response · Authors · 2025-11-26
> **Response to reviewer TLG6 (2/3)**
>
> ## **W2. Increasing dataset size**
>
> We fully agree that the community would benefit from even larger interaction-focused datasets.
>
> In this work, our primary goal is not merely to release a fixed 11K dataset, but to propose a **reproducible and generalizable curation pipeline** capable of constructing large-scale, high-quality, interaction-aware video datasets with interaction-centric captions and instance mask tracks. Importantly, the pipeline is designed to operate on any text-video paired source dataset.
>
> In the current paper, we report an 11K dataset obtained by applying our pipeline to subsets of HOIGen and PE-Video. Specifically, after processing videos from HOIGen and PE-Video, we retain **9,341** and **2,534** high-quality interaction-aware samples, respectively. To address the reviewer's concern about scalability, we further increased the volume of processed source videos and verified that our pipeline naturally scales beyond 11K samples. Processing a larger portion of HOIGen (up to **43,107** videos) yields **10,634** curated samples; processing more of PE-Video (up to **119,961** videos) yields **8,186** curated samples; and applying the same pipeline to a general open-domain dataset, OpenVid-1M (up to **22,062** videos), yields an additional **2,568** interaction-aware samples. **Aggregating these curated samples expands our dataset from 11K to 21K videos.**
>
> These results demonstrate that, given more source text-video pairs, our pipeline can be straightforwardly scaled to substantially larger datasets. In the revision, we clarify that (i) our key contribution lies in the general, reproducible interaction-aware curation pipeline described in Sec. 3, and (ii) the 11K dataset reported in the paper is only a subset-not a hard upper bound, as detailed in Appendix A. We also explicitly include the extended curation results (HOIGen, PE-Video, and OpenVid-1M) to make the scalability and benefits to larger models and the broader community more evident.

---

> ### Author Response · Authors · 2025-11-26
> **Response to reviewer TLG6 (3/3)**
>
> ## **W3. Discussion on involving other modalities**
>
> We sincerely appreciate the reviewer for raising this perspective. Recent works such as VideoJAM and UDPDiff indeed leverage additional modalities (e.g., optical flow, depth, segmentation, point tracking) to enhance video generation quality, whereas our work focuses primarily on mask tracks (instance-level segmentation mask tracks) as the main auxiliary signal.
>
> In our setting, we require a reference modality that supports three key properties essential for analyzing and supervising interaction awareness:
>
> (1) Instance-level semantic precision: the ability to verify whether a specific instance in the scene is correctly grounded.
>
> (2) Temporal continuity: the ability to assess whether this grounding is consistently maintained across frames.
>
> (3) Same-class instance disambiguation: the ability to distinguish multiple instances belonging to the same category (e.g., two people).
>
> Among the candidate modalities including optical flow, depth, point tracking, and segmentation, instance-level segmentation mask track is the only one that naturally satisfies all three conditions:
>
> - Optical flow offers dense motion cues and temporal continuity but lacks instance-level grouping and semantic instance IDs, so it cannot satisfy properties (1) and (3). It also suffers from limited long-term consistency.
> - Depth maps encode geometry but do not provide instance-level grouping or identity, making it difficult to determine “who interacts with what.” Thus they do not satisfy properties (1) and (3).
> - Point tracking provides fine-grained temporal trajectories but only at sparse pixel locations and without instance-level semantics, so it does not meet properties (1) and (3).
> - In contrast, instance mask tracks give us clear instance-level regions, persistent IDs over time, and separation between multiple instances, which is exactly what we need for grounding and propagation.
>
> For these reasons, we focus on *mask tracks* as the core supervision and analysis signal. This choice does not preclude integrating additional modalities. We agree with the reviewer that the listed cues are highly complementary: optical flow could strengthen supervision of fine-grained motion and temporal coherence, while depth could help resolve occlusions and 3D proximity in contact-heavy interactions.
>
> Our formulation is modality-agnostic on the analysis side. The same semantic grounding and propagation framework could be extended to incorporate multi-modal cues by adding flow- or depth-based consistency terms or by conditioning the DiT backbone on these signals during training. A full multi-modal integration is beyond the scope of this submission due to computational constraints, but we view this as a valuable next step and explicitly discuss it in the limitations and future work section.
>
> We further include this discussion in Appendix A.2 of the revised manuscript.

---

### Author Response · Authors · 2025-11-26
**General Responses to Reviewers**

We sincerely appreciate the positive comments on the novelty of our work (**payz**), clarity and readability of our paper (**payz**, **NstU**), and their recognition of the significance of our work for the community (**payz**, **NstU**). We also deeply grateful for their appreciation of our comprehensive analysis and how MATRIX directly addresses these issues (**TLG6**, **payz**), the practicality and reproducibility of our data pipeline (**TLG6**, **NstU**), the effectiveness of our proposed design (**TLG6**, **payz**), and the potential usefulness of our works including dataset, visualization, and method in future work (**payz**, **NstU**).

---

### Author Response · Authors · 2025-12-03
**Summary of our works and discussions**

Dear AC and reviewers,
We would like to sincerely thank the ACs and reviewers for taking the time to thoroughly review our manuscript and providing insightful suggestions that helped us improve this work.
As summarized in our general response, we sincerely appreciate the reviewers’ thoughtful and constructive feedback, as well as their positive assessment of the novelty, clarity, and potential impact of our work. Encouraged by these comments, we have conducted additional experiments and added clarifications to further substantiate our claims and improve the manuscript. The revised version includes the following main updates:

- **Clarified the novelty of our analysis.** We revised the introduction to better articulate the motivation and novelty of our semantic grounding and propagation analyses, and to more clearly position our video-level analysis within the existing literature.

- **Explained the extensibility of our dataset.** We clarified how our dataset can be extended to broader scenarios, and we added qualitative examples in **Figures 30–33** to illustrate these extensions.

- **Improved figure interpretability.** We refined **Figure 7** to make the analysis easier to follow and to better convey the key takeaways.

- **Strengthened the ablation study.** In **Section 6.3 and Table 2**, we added an additional variant to more clearly demonstrate the effectiveness of our analysis and proposed method.

- **Highlighted generalization to diverse scenarios.** We added **Figure 24** to showcase results on non-contact, non-human, and other challenging scenarios, thereby emphasizing the generalization ability of our approach.

- **Demonstrated applicability to various DiT-based backbones.** We extended our experiments in **Section 6.4, Figure 10**, **Supplementary Section C, and Figures 15–17** to show that our framework is compatible with a range of DiT-based video diffusion architectures. We also provide the qualitative comparisons between Wan2.1-14B-I2V and MATRIX in **Figure 42 and Figure 43**, and additionally submitted a supplementary video that visualizes these comparisons.

- **Justified the use of mask tracks as supervision and discussed broader extensions.** We added **Supplementary Section A.2** to justify our choice of mask tracks as a supervision signal. In addition, **Supplementary Section J** discusses how our framework could be combined with other modalities, opening directions for future work.

- **Verified robustness beyond outliers.** We included Supplementary **Section G.4 and Figure 22** to show that our method consistently improves performance over baselines and does not rely on a small number of outlier cases.

- **Analyzed efficiency versus effectiveness.** We added **Supplementary Section H.3 and Table 5** to provide a more detailed analysis of training cost and to highlight the favorable trade-off between computational cost and performance gains offered by our approach.

We hope that these revisions and additional results adequately address the reviewers’ concerns and further clarify the contributions and robustness of our work. We are very grateful for the reviewers’ time and effort, and we respectfully hope that the improved manuscript will be viewed favorably.

Best regards,

Authors of Submission 4661

---

### Meta-Review · Area_Chair_e9vX · 2026-01-06

**Summary:**

The paper addresses the inability of current Video DiTs to correctly model multi-instance and subject-object interactions. The authors introduce MATRIX-11K (a dataset with mask tracks), perform a breakdown analysis of attention layers to identify "grounding" and "propagation" failures, and propose a regularization technique (MATRIX) to align attention in dominant layers with instance masks.

The paper tackles a specific, well-defined problem in video generation: the failure of models to maintain semantic consistency during interactions. The reviewers (TLG6, payz) praised the depth of the analysis (identifying dominant attention layers) and the pragmatic solution. While Reviewer NstU felt the findings were unsurprising, the rebuttal demonstrated that the method is not only effective on the original baseline but transfers to SOTA models like Wan2.1. The AC agrees with the majority that the insights into semantic grounding and propagation in video DiTs are valuable.

**Reviewer Concerns:**

The rebuttal successfully addressed the majority of concerns. Concerns regarding the use of older backbones (TLG6, NstU) were addressed by extending to Wan2.1 and HunyuanVideo. Questions regarding the scalability of the dataset (TLG6) and the training cost/efficiency (payz) were answered with additional scaling experiments and an iso-FLOPs analysis. The request for a specific ablation on "fine-tuning without the proposed method" (NstU) was clarified by pointing to existing results in Table 2. NstU's concern regarding the "triviality" of the findings remains a point of disagreement, though the other two reviewers found the analysis novel and significant.

**Reviewer Scores:**

Reviewers TLG6 and payz were already leaning favorably (score 8).

Reviewer NstU (Score: 2): The authors objectively addressed the factual complaints regarding "outdated baselines" and "missing ablations." While the reviewer might maintain reservations about novelty, the demonstration that the method works on Wan2.1 significantly strengthens the empirical validity. The reviewer would have likely increased score (maybe to a 6)

---

### Decision · Program_Chairs · 2026-01-26

Accept (Poster)